# GICA: The Gap-Index Compositional Arm Framework for Sample-Efficient Test-Time Scaling

## Abstract

Test-time scaling (TTS) improves the reasoning capabilities of large language models (LLMs) by generating multiple candidate reasoning paths and using a verifier to select among them. Process reward models (PRMs), which score each intermediate step rather than only the final answer, yield stronger downstream accuracy but at a higher cost. Recently, PRMs that scale at test-time by generating long verification CoTs have been found to be more accurate at verification, but with a prohibitive cost that scales with both the number of paths and their length (number of steps), limiting scalability precisely where TTS is most beneficial. We recast reasoning-based process-level verification as a sample-efficient adaptive selection problem. We propose GICA (Gap-Index Compositional Arm framework), a bandit-based framework that exploits the compositional structure of reasoning paths to share information across related steps and identify the top-$K$ candidates. We establish theoretical correctness and a fixed-confidence sample-complexity bound, and validate GICA through synthetic experiments and in a TTS setup employing an end-to-end TTS pipeline across three mathematical reasoning benchmarks. We experiment with two open-weight math LLMs serving as generators and two LLMs as process-level, reasoning-based verifiers. GICA matches the accuracy of exhaustive process-level verification while substantially reducing verifier calls (by **4.2**×) and inference runtime (by **4.3**×), making fine-grained step-level supervision practical at scale. We **open-source our code and data** to facilitate future research.[1]

## 1 Introduction

Large language models (LLMs) have led to tremendous advances across a wide range of tasks, including numerical reasoning and other natural language tasks. While scaling to larger models through pre-training has recently been found to saturate, a new frontier of test-time scaling (TTS) has emerged. TTS proposes scaling compute at test time to improve LLMs' reasoning capabilities and also enables enhancing the performance of smaller language models. Rather than decoding a single solution, the model samples many candidate chain-of-thought (CoT) reasoning paths, and a verifier selects the final answer (Snell et al., 2025; Zhang et al., 2025b). The verifiers that are used to guide TTS can be categorized into output reward models (ORM) (Cobbe et al., 2021) and process reward models (PRM). Among these, PRMs (Uesato et al., 2022; Lightman et al., 2023) are particularly effective because, unlike ORMs, they score each intermediate reasoning step rather than only the final answer, which yields consistently stronger downstream accuracy than just outcome-level scoring. More recent works like ThinkPRM (Khalifa et al., 2026) have also explored the use of reasoning-based LLMs as PRMs, which are allocated more compute at test time to enable detailed reasoning for step-level verification.

Such reasoning-based PRMs (Khalifa et al., 2026) further improve verification quality by generating a long CoT, before emitting a scalar score. However, this step-level fine-grained reasoning comes at a substantial cost. Each PRM call is a full LLM forward pass conditioned on the steps verified so far as a prefix, and exhaustive Best-of-$M$ verification evaluates every step of every sampled reasoning path. The total compute therefore

---

[1] https://anonymous.4open.science/r/GICA-1B57

scales with both the number of candidate paths $M$ and the per-path step count $T_p$, which makes process-level scoring prohibitive in precisely the large-$M$ regime where TTS yields its largest accuracy gains (Snell et al., 2025; Wu et al., 2024). In practice, this forces practitioners either to cap $M$ or to fall back on cheaper outcome-level verifiers, in both cases giving up the benefit of fine-grained step-level supervision.

To mitigate the cost of exhaustive verification, prioritizing verification paths can reduce the number of calls to PRM and the computational cost. Hence, in this work, we cast the verification task at inference time as an adaptive selection problem, which is to query only the steps most informative for identifying the top-$K$ reasoning paths instead of querying every step. Fixed-confidence pure-exploration algorithms for linear stochastic bandits, including M-LinGapE (Xu et al., 2018), LinGIFA (Réda et al., 2021), and CASE (Purohit et al., 2025), offer sample-efficient solutions for top-$K$ identification. Top-$K$ identification approaches help separate optimal arms from sub-optimal arms through adaptive sampling. However, applying them directly to PRM verification runs into a structural mismatch. This is because the complete reasoning paths are modeled as arms to be ranked in these approaches, but the available feedback is observed at the step level. Existing top-$K$ algorithms treat each arm as a single queryable entity, relying solely on correlations between arm-level features, and neither directly observe the composing steps nor exploit the correlations among these atomic units. As a consequence, information obtained from querying a single step cannot be efficiently propagated across paths that share semantically similar steps, causing sample efficiency to degrade as $M$ grows. Our proposed approach GICA overcomes this limitation by lifting the bandit problem to compositional structures, enabling joint step-level learning and path-level optimization. By explicitly modeling the compositional structure of reasoning paths, each step-level observation simultaneously informs a large number of arms, leading to faster shrinkage of confidence sets and yielding substantial gains in both sample efficiency and generalization. This enables GICA to have a higher sample-efficiency during verification than existing approaches like LinGIFA or CASE. Sample efficiency in the context of TTS refers to the number of verifier calls required to arrive at the optimal reasoning paths. For instance, for a problem setup with a total of $M$ paths and $T_p$ steps in a path (though in reality the number of steps in a path could be variable), at each time step of the bandit run, LinGIFA, CASE requires $O(T_pM)$ calls to the verifier to sample reward for the arm (reasoning path). Whereas, GICA requires only $O(1)$ verifier call as it samples the most informative step and propagates the feedback to paths, enabling updation of current top-$K$ list.

We address this gap by formulating **sample-efficient process-level verification** as a fixed-confidence top-$K$ identification problem over compositional arms. In our formulation, each reasoning path is a parent arm whose feature is the average of its constituent step features under a shared linear model, and each queried step yields a noisy observation that informs the parameter shared by all paths. This makes explicit the information transfer across paths that is already implicit in how PRMs are trained. Building on this formulation, we propose the GICA, a gap-index-based linear stochastic bandit method designed to capture this compositional structure and guarantee the identification of the top-$K$ CoTs (arms) using a minimal number of verifier calls. Furthermore, we **emphasize that our aim is not to surpass exhaustive process-level verification** in downstream accuracy, which already serves as an upper bound for any verifier-based selection rule. Rather, our goal is to make verification practical at scale by pruning uninformative verifier calls while preserving task performance. Our contributions are as follows: (i) This work casts sample-efficient process-level verification in TTS as a fixed-confidence top-$K$ identification problem over compositional arms, introducing GICA to capture this compositional structure and guarantee the identification of the top-$K$ CoTs (arms) using a minimal number of verifier calls (Section 3). (ii) Theoretical analysis establishes a fixed-confidence sample-complexity bound for GICA (Theorem 3.8). (iii) Empirical validation of GICA encompasses a controlled synthetic setup alongside three math-reasoning benchmarks, MATH-500, MathOdyssey, and AIME, two open-weight math LLMs DeepSeekMath-RL-7B and InternLM2-Math-Plus-7B acting as generators, and two process-level LLMs ThinkPRM-1.5B and ThinkPRM-7B serving as the reasoning-based verifiers. GICA attains task accuracy close to the exhaustive Best-of-$M$ upper bound while reducing the verifier calls by up to **4.2×** and inference runtime by up to **4.3×** relative to the strongest bandit baseline (Section 4).

## 2 Related Works

Our work sits at the intersection of two lines of research: TTS with process-level verification in LLMs, and fixed-confidence top-$K$ identification in linear stochastic bandits.

## 2.1 Test Time Scaling

Test-time scaling aims to improve the reasoning performance of LLMs by allocating additional inference-time compute (Zhang et al., 2025b). Early approaches primarily focused on sampling multiple responses and selecting the solution through majority voting(Wang et al., 2023) or search-based methods (Yao et al., 2023; Wan et al., 2024; Xie et al., 2023). Recent developments can be broadly categorized into modifying sampling distributions of LLM through supervised fine-tuning (DeepSeek-AI et al., 2025; OpenAI, 2024; Singh et al., 2024; Madaan et al., 2023; Zelikman et al., 2022; Jin et al., 2025). Alternatively, recent approaches train a verifier (reward) model to select an answer from multiple candidates (Snell et al., 2025; Wang et al., 2024a; Nichols et al., 2020; Cobbe et al., 2021; Uesato et al., 2022; Lightman et al., 2023), commonly known as *search against a verifier*, which will be the core setup in our work. One of the canonical ways to employ TTS is to sample M complete solutions in parallel and apply Best-of-M. Best-of-M involves sampling multiple outputs and using a trained verifier model to select the best final output or the reasoning path that led to it.

Verifiers used to select the best solution are divided into ORM (Cobbe et al., 2021) or PRM (Uesato et al., 2022; Lightman et al., 2023). ORMs only focus on the final answer for selecting the best solution. PRMs, on the other hand, are more fine-grained and perform verification of each step of the reasoning path rather than just the final solution. While PRMs are quite expensive to train (Lu et al., 2024), they have demonstrated to improve performance on a wide range of numerical reasoning and complex reasoning tasks (Zeng et al., 2025; Khalifa et al., 2026). PRMs generally could be discriminative (Uesato et al., 2022) or generative (Zhang et al., 2025a). Given a reasoning step, the model encodes the input and outputs a binary score using a classification head. Final solution quality is often estimated by aggregating the predicted scores across steps (Snell et al., 2025; Wu et al., 2024). Generative verifiers (Zhang et al., 2025a) leverage the inherent capabilities of LLMs, including natural language generation, CoT reasoning (Wei et al., 2022), and instruction-following, to assess the correctness of solutions. To bridge the gaps in discriminative and generative verifiers, more recent works have proposed exploring scaling of test-time compute for generative verifiers (Khalifa et al., 2026). ThinkPRM performs reasoning by leveraging test-time compute to perform process-level verification. However, reasoning-based verifiers incur huge computational costs during inference and also lead to large inference runtime scaling with the number of paths sampled. Hence, our primary goal is to investigate the sample-efficient prioritization of verification paths in expensive process-level verification mechanisms.

## 2.2 Linear Stochastic Bandits and Top-K Selection

To reduce the verification cost, one approach is to prioritize the paths to be verified in a parallel Best-of-M setting through a sample-efficient mechanism. One possible formulation of this task is the top-K selection problem, which is well-studied in linear-stochastic bandit literature (Réda et al., 2021). The approaches proposed for top-K selection can be divided into the fixed-budget (Bubeck et al., 2013) or the fixed-confidence setting (Kalyanakrishnan et al., 2012). In this work, we adopt the fixed-confidence setting where the error probability of top-K identification should be within a predefined parameter $\delta \in (0, 1)$. Several adaptive sampling algorithms have been proposed for top-K identification (Kaufmann, 2014; Kalyanakrishnan et al., 2012), but they do not focus on sample complexity. To bridge this gap, adaptive sampling methods like LinGapE (Xu et al., 2018), PEPS (Li et al., 2023) have been proposed, which demonstrate low sample-complexity compared to existing approaches. However, they are primarily designed for best-arm identification and not the top-K identification problem. LINGIFA (Réda et al., 2021) was one of the first works to propose a unified sample-efficient framework for top-K identification and also provides mechanisms to adapt algorithms like LinGapE to the top-K setting (M-LINGAPE). LINGIFA proposes a gap-index framework that maintains a current estimate of top-m arms and, in each round, compares the two most ambiguous arms. One of the ambiguous arms is sampled from the current top-K, and the other is the challenger arm compared to the rest of the arms in the global set. The arm that helps distinguish between these ambiguous arms is sampled in that round. CASE (Purohit et al., 2025) followed up on M-LINGAPE by proposing an adaptive sampling mechanism to maintain a challenger shortlist that reduces the number of gap-index computations and arm pulls, leading to more efficient runtime and sample complexity. Beyond its challenger sampling rule, CASE models the LLM based reward of an exemplar subset through a shared linear function of semantic similarity features. Online Relevance Estimation uses the same general principle in retrieval by learning a linear combination of query and document affinity features from a limited number of expensive cross encoder

scores (Purohit et al., 2025; Rathee et al., 2025). In both cases, shared parameters transfer observed feedback to unevaluated candidates, while continued direct evaluation or adaptive exploration refines the resulting ranking. However, adopting the above approaches to the PRM verification setting is non-trivial due to the compositional nature of the problem. While the CoTs can be projected as arms to explore and prioritize them for verification, the actual verification happens at the step-level for each CoT. Hence, we propose a new gap-index approach for this compositional setting, which is runtime-efficient and minimizes the number of verifier calls compared to existing sample-efficient algorithms.

## 3 Methods

We study sample-efficient process-level verification for TTS, aiming to identify a small set of high-quality reasoning paths with the fewest possible verifier calls. We first describe the TTS setting and its verification signal, then formalize the task as a fixed-confidence top-$K$ identification problem over compositional arms, and finally present GICA with its theoretical guarantee.

### 3.1 Test-Time Scaling with Process-Level Verification

We consider a TTS setting in which, given a test input $I_{test}$ (e.g., a math word problem), an LLM generates multiple candidate CoT reasoning paths following the *parallel scaling* formulation in literature (Cobbe et al., 2021) rather than a single solution. Let $\Pi = \{\pi_1, \ldots, \pi_M\}$ denote the set of $M$ sampled reasoning paths for input $I_{test}$, where, for $p \in \{1, \ldots, M\}$, the $p$-th path $\pi_p = (s_{p,1}, \ldots, s_{p,T_p})$, is a sequence of intermediate steps of length $T_p$. Let $\mathcal{S}$ denote the set of all steps appearing in any path, i.e., $\mathcal{S} = \{s_{p,q} : p \in [M], q \in [T_p]\}$. To rank these candidate paths, verifier models are used to assess solution quality. ORMs evaluate only the final answer, whereas process reward models PRMs provide step-level feedback for intermediate reasoning steps (Zeng et al., 2025; Luo et al., 2024; Lu et al., 2024). Specifically, when a step $s_{p_t,q_t} \in \mathcal{S}$ is selected for verification at round $t$, the PRM returns a scalar score $y_t = \text{PRM}([I_{test}, s_{p_t,1:q_t-1}], s_{p_t,q_t})$, where $[I_{test}, s_{p_t,1:q_t-1}]$ denotes the sequence of steps within a path preceding the queried step provided as prefix to the PRM. In contrast to ORM-based selection, process-level verification can exploit fine-grained information about the quality of intermediate reasoning steps (Lightman et al., 2023; Lyu et al., 2025).

Exhaustive process-level verification scales poorly with both the number of sampled paths and the number of steps per path, making it prohibitively expensive for large-scale test-time search (Lightman et al., 2023). This creates a fundamental trade-off, i.e., evaluating more reasoning paths can improve solution quality, but only if the available verification budget is sufficient to support their evaluation. Our goal is to address this trade-off by identifying the top-$K$ reasoning paths using substantially fewer PRM queries than exhaustive evaluation. In the next subsection, we formalize this objective as a fixed-confidence top-$K$ identification problem over compositional arms. Figure 1 illustrates the overall workflow of the proposed method for TTS setup. Firstly, a generator LLM (base LLM) generates a large number of reasoning paths, with each path comprising a stepwise solution to the original problem. Then, to circumvent the computational cost and prohibitive runtime associated with exhaustive verification, we propose GICA, a bandit-based approach that selects an optimal subset of reasoning paths by sampling the most informative steps. Finally, the final answer is then derived from these prioritized paths.

### 3.2 Problem Formulation: Top-$K$ Identification over Compositional Arms

We formalize process-level verification in TTS as a fixed-confidence top-$K$ identification problem in a linear stochastic bandit model with compositional arms. The compositional structure reflects a defining feature of process-level verification, i.e., complete reasoning paths are the objects to be ranked, but the verification signal is observed only at the level of individual steps. We therefore treat each reasoning path as a virtual parent arm composed of multiple intermediate steps, while each query returns a noisy scalar response from a single step. To capture this structure mathematically, we associate each step $s_{p,q} \in \mathcal{S}$ with a known feature vector $x_{s_{p,q}} \in \mathbb{R}^d$ satisfying $\|x_{s_{p,q}}\|_2 \leq L$. Because PRMs embed diverse reasoning steps into a common space through a single fine-tuned backbone (Lightman et al., 2023; Wang et al., 2024b; Luo et al., 2024; Lu et al., 2024), we posit that the relationship between such a non-linear representation and rewards can

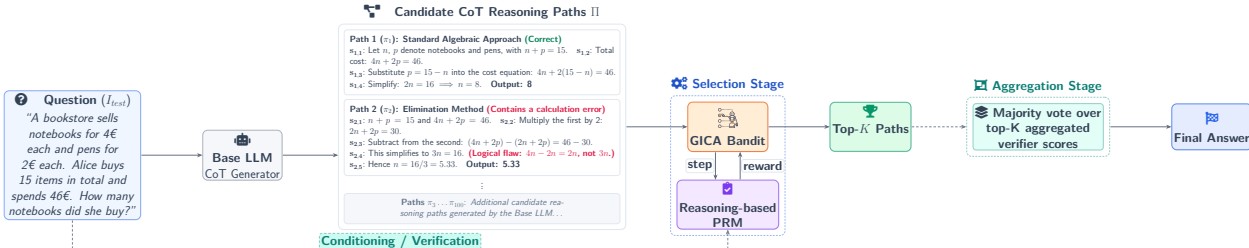

Figure 1: Integration of GICA in the TTS workflow. The **Selection Stage** evaluates the $M$ candidate CoT paths via step-level PRM queries to produce a top-$K$ shortlist without full path evaluations. The **Aggregation Stage** collapses it into the final answer by majority vote. Figure 2 details one selection round.

be modeled using a linear utility surrogate model with a shared unknown parameter $\theta^\star \in \mathbb{R}^d$, $\|\theta^\star\|_2 \le S_0$. Linear models provide a tractable basis for confidence based adaptive selection (Abbasi-Yadkori et al., 2011; Filippi et al., 2010; Li et al., 2017). Recent sample efficient systems apply the same modeling principle to expensive neural evaluators. Online Relevance Estimation transfers sparse cross encoder feedback through shared lexical and semantic affinity features, while CASE transfers LLM based validation feedback through semantic similarity features of exemplar subsets (Rathee et al., 2025; Purohit et al., 2025). In both settings, the linear surrogate provides an inexpensive ranking signal that is refined through further direct evaluations or adaptive exploration rather than an exact description of the underlying evaluator. We adopt this principle at the step level, where sharing $\theta^\star$ allows each queried PRM score to update the estimated utilities of multiple paths through their structured features.

At round $t = 1, 2, \ldots$, the learner selects an index pair $(p_t, q_t)$ with $p_t \in [M]$ and $q_t \in [T_{p_t}]$, queries the corresponding step $s_t := s_{p_t, q_t} \in \mathcal{S}$, and observes the scalar verifier feedback $y_t = x_{s_t}^\top \theta^\star + \eta_t$, where $\eta_t$ is a noise term modeling fluctuations in the verifier's response. Let $\mathcal{F}_{t-1} := \sigma\big((s_i, y_i)_{i=1}^{t-1}\big)$ denote the history up to round $t - 1$. To enable self-normalized concentration inequalities and the construction of high-probability confidence sets for $\theta^\star$, as is standard in linear stochastic bandits (Abbasi-Yadkori et al., 2011), we impose the following light-tailed assumption on the noise. This is a tractable approximation for step-level verifier feedback that is particularly well justified when PRM scores are normalized.

**Assumption 3.1** (Conditionally sub-Gaussian noise). The noise sequence $(\eta_t)_{t\ge 1}$ is conditionally zero-mean and $R$-sub-Gaussian: for all $t \ge 1$ and $\lambda \in \mathbb{R}$, $\mathbb{E}[\eta_t \mid \mathcal{F}_{t-1}, s_t] = 0$, and $\mathbb{E}[\exp(\lambda \eta_t) \mid \mathcal{F}_{t-1}, s_t] \le \exp\left(\frac{\lambda^2 R^2}{2}\right)$.

**Assumption 3.2** (Pair–step correlation). There exists a constant $\rho^\dagger \in (0, 1]$ such that for every positive-definite matrix $A \succeq \lambda I_d$, every ordered pair of distinct paths $(\pi_p, \pi_{p'}) \in \Pi^2$ with $g(\pi_p, \pi_{p'}) \ne 0$, and every step $s \in \mathcal{S}$ with $x_s \ne 0$, $\dfrac{\left\langle g(\pi_p, \pi_{p'}), x_s \right\rangle_{A^{-1}}^2}{\|g(\pi_p, \pi_{p'})\|_{A^{-1}}^2 \, \|x_s\|_{A^{-1}}^2} \ge \rho^\dagger$.

In Assumption 3.2, the left-hand side measures the normalized alignment between a pair-difference direction $g(\pi_p, \pi_{p'})$ and a step feature $x_s$ in the $A^{-1}$-weighted geometry induced by the design matrix. The assumption requires a uniform floor $\rho^\dagger > 0$ on this alignment, i.e., no pairwise contrast is $A^{-1}$-orthogonal to any step feature. By the Sherman–Morrison identity (Lemma 3.7) this is exactly what keeps each queried step informative about every path gap, letting information propagate across compositional arms through $\theta^\star$. This is mild in our TTS setting, since all paths solve the same question $I_{\text{test}}$ through a shared PRM (encoder) backbone, their steps are semantically correlated rather than independent. Hence, at least some steps of one path overlap with steps of the others, and a contrast $g(\pi_p, \pi_{p'})$, being an average of such step features, cannot be exactly orthogonal to them. Note that the assumption concerns only this geometric non-orthogonality, which governs how fast uncertainty contracts. It does not presume that semantically similar steps share the same correctness, since correctness enters solely through the prefix-conditioned verifier observations $y_t$, and weaker alignment merely increases the required number of queries through the inverse dependence on $\rho^\dagger$ in Theorem 3.8.

Because verification operates at the step level but ranking operates at the path level, we must aggregate step features into path features. For each path $\pi_p$, we therefore define the path feature and its associated

mean utility as $g(\pi_p) := \frac{1}{T_p} \sum_{q=1}^{T_p} x_{s_{p,q}}$, and $\mu(\pi_p) := g(\pi_p)^\top \theta^\star$ respectively. Length-normalized averaging (rather than summation) places paths of differing lengths on a common scale, consistent with standard process-supervision aggregation (Lightman et al., 2023; Wang et al., 2024b; Luo et al., 2024; Lu et al., 2024). Given an estimator $\widehat{\theta}_{t-1}$ of $\theta^\star$ measurable with respect to $\mathcal{F}_{t-1}$ (specified in Subsection 3.3), the corresponding plug-in path estimate is $\widehat{\mu}_{t-1}(\pi_p) := g(\pi_p)^\top \widehat{\theta}_{t-1}$, and the predicted single-step reward is $\widehat{y}_t := x_{s_t}^\top \widehat{\theta}_{t-1}$.

Without loss of generality, we index the paths so that $\mu(\pi_1) \geq \cdots \geq \mu(\pi_K) > \mu(\pi_{K+1}) \geq \cdots \geq \mu(\pi_M)$, an ordering unknown to the learner. Let $P_K^\star \subseteq \Pi$ be any size-$K$ maximizer of the cumulative path utility, i.e., $P_K^\star \in \arg\max_{P \subseteq \Pi:\ |P|=K} \sum_{\pi \in P} \mu(\pi)$, and let $\mu_K^\star := \min_{\pi \in P_K^\star} \mu(\pi)$ be the lowest mean among these top paths. Because exact identification becomes ill-posed when several paths are nearly indistinguishable from the boundary defined by $\mu_K^\star$, we relax the target to an $\epsilon$-tolerance so that for any $\epsilon \geq 0$, the $\epsilon$-expanded top-$K$ set $P_K^{\star,\epsilon} := \{\pi \in \Pi : \mu(\pi) \geq \mu_K^\star - \epsilon\}$ collects all paths whose mean lies within $\epsilon$ of the top-$K$ threshold. To track our progress toward this set, we let $\widehat{P}_K(t) \subseteq \Pi$ denote the empirical top-$K$ shortlist at round $t$, defined as any size-$K$ maximizer of the plug-in path estimates, $\widehat{P}_K(t) \in \arg\max_{P \subseteq \Pi:|P|=K} \sum_{\pi \in P} \widehat{\mu}_{t-1}(\pi)$.

**Definition 3.3** ($\epsilon$-optimal top-$K$ set). For any $\epsilon \geq 0$, a size-$K$ subset $P \subseteq \Pi$ is said to be $\epsilon$-optimal if every path $\pi_p \in P$ satisfies $\mu(\pi_p) \geq \mu_K^\star - \epsilon$, equivalently $P \subseteq P_K^{\star,\epsilon}$.

*Remark* 3.4 ($\epsilon$-relaxed top-$K$ target). When $\epsilon = 0$, exact top-$K$ identification is unambiguous only if $\mu_K^\star > \max_{\pi_p \notin P_K^\star} \mu(\pi_p)$; otherwise multiple valid top-$K$ maximizers may exist.

To support the adaptive sampling rule of the subsection 3.3, we record pairwise quantities between paths drawing inspiration from gap-index frameworks (Réda et al., 2021). For any pair of distinct paths $(\pi_p, \pi_{p'})$ with $p, p' \in [M]$ and $p \neq p'$, we define the pairwise feature difference $g(\pi_p, \pi_{p'}) := g(\pi_p) - g(\pi_{p'})$, the corresponding true gap $\Delta(\pi_p, \pi_{p'}) := \mu(\pi_p) - \mu(\pi_{p'}) = g(\pi_p, \pi_{p'})^\top \theta^\star$, and the estimated gap at round $t$, $\widehat{\Delta}_t(\pi_p, \pi_{p'}) := \widehat{\mu}_t(\pi_p) - \widehat{\mu}_t(\pi_{p'})$. These pairwise quantities will play a central role in GICA, which adaptively focuses verification effort on the most ambiguous boundary between the current empirical top-$K$ shortlist $\widehat{P}_K(t)$ and its challengers. We further define the global minimum gap $\Delta_{\min}^\Pi = \min_{(\pi_p, \pi_{p'}) \in \Pi^2, p \neq p'} |\Delta(\pi_p, \pi_{p'})|$, the true boundary pair set $\mathcal{C}_K^\star := \{(\pi_p, \pi_{p'}) \in \Pi^2 : \pi_p \in P_K^\star,\ \pi_{p'} \notin P_K^\star\}$ and the boundary gap $\Delta_{\mathcal{C}} := \min_{(\pi_p, \pi_{p'}) \in \mathcal{C}_K^\star} \Delta(\pi_p, \pi_{p'})$, the smallest true gap between a top-$K$ path and a challenger.

**Assumption 3.5** (Boundary separability). $\Delta_{\mathcal{C}} > \epsilon \geq 0$ and $\Delta_{\mathcal{C}} \geq \Delta_{\min}^\Pi > 0$.

### 3.3 GICA: A Gap-Index Framework for Compositional Arms

We now present GICA, a model-based fixed-confidence algorithm that exploits the compositional structure of reasoning paths to share statistical information across intermediate steps and efficiently identify the top-$K$ best paths. At a high level, GICA maintains a self-normalized confidence set for the shared parameter $\theta^\star$, uses it to track the most ambiguous boundary between the empirical top-$K$ shortlist and its challengers, greedily queries the step whose observation maximally contracts the pairwise uncertainty along that boundary, and halts once the shortlist is statistically certified to be $\epsilon$-optimal.

#### 3.3.1 Confidence Sets and Updating Rule

At each round $t$, the algorithm queries a step $s_t \in \mathcal{S}$ and observes $y_t = x_{s_t}^\top \theta^\star + \eta_t$, where $\eta_t$ is conditionally $R$-sub-Gaussian as specified in Subsection 3.2. To estimate the unknown parameter $\theta^\star$, GICA employs a ridge (regularized least-squares) estimator (Abbasi-Yadkori et al., 2011; Réda et al., 2021; Purohit et al., 2025). Let $\lambda > 0$ be a fixed regularization parameter. Define the regularized design matrix and the corresponding parameter estimate by

$$V_t := \lambda I_d + \sum_{i=1}^t x_{s_i} x_{s_i}^\top, \qquad \widehat{\theta}_t := V_t^{-1} \sum_{i=1}^t y_i\, x_{s_i}, \tag{1}$$

with initializations $V_0 := \lambda I_d$ and $\widehat{\theta}_0 := 0$. We adopt the shorthand $\langle a, b \rangle_{V_t^{-1}} := a^\top V_t^{-1} b$ and $\|a\|_{V_t^{-1}}^2 := a^\top V_t^{-1} a$ throughout. For any step $s \in \mathcal{S}$, its true mean and plug-in estimate are $\mu(s) := x_s^\top \theta^\star$ and $\widehat{\mu}_t(s) := x_s^\top \widehat{\theta}_t$. Extending this to the path level via the average aggregation defined in Subsection 3.2, the estimated path mean is $\widehat{\mu}_t(\pi_p) := g(\pi_p)^\top \widehat{\theta}_t$. For any pair of paths $(\pi_p, \pi_{p'}) \in \Pi^2$, we define the pairwise variance and

the corresponding confidence width as $\sigma_t^2(\pi_p, \pi_{p'}) := \|g(\pi_p, \pi_{p'})\|_{V_t^{-1}}^2$ and $W_t(\pi_p, \pi_{p'}) := \beta_t(\delta)\sqrt{\sigma_t^2(\pi_p, \pi_{p'})}$, respectively, where $\beta_t(\delta)$ is a confidence scaling factor specified in Eq. (6) in Lemma. A.1 (Abbasi-Yadkori et al., 2011).

**Definition 3.6** (Self-normalized confidence event)**.** For a fix $\delta \in (0,1)$, the self-normalized confidence event is defined as $\mathcal{E}_\delta := \left\{\forall t \ge 0 : \|\widehat{\theta}_t - \theta^\star\|_{V_t} \le \beta_t(\delta)\right\}$.

For any pair $(\pi_p, \pi_{p'}) \in \Pi^2$, Lemma A.1 and Eq. (8) immediately yield the upper gap index $B_t(\pi_p, \pi_{p'}) := \widehat{\Delta}_t(\pi_p, \pi_{p'}) + W_t(\pi_p, \pi_{p'})$, which constitutes a high-probability upper confidence bound on the true gap $\Delta(\pi_p, \pi_{p'})$ uniformly over all rounds $t \ge 0$.

### 3.3.2 Boundary Selection Rule

To minimize sample complexity, GICA focuses its verification effort at each round on the paths that are hardest to classify under the current estimates. The separability of any two candidate paths is quantified using the following gap-index and ambiguity-ratio. Inspired by Information Directed Sampling (Kirschner et al., 2023), we define the pairwise gap index, $G_t(\pi_p, \pi_{p'}) := \frac{\widehat{\Delta}_t(\pi_p, \pi_{p'})^2}{\sigma_t^2(\pi_p, \pi_{p'})}$ and its reciprocal, the ambiguity ratio, $\mathcal{A}_t(\pi_p, \pi_{p'}) := \frac{\sigma_t^2(\pi_p, \pi_{p'})}{\widehat{\Delta}_t(\pi_p, \pi_{p'})^2}$. The pairwise gap index is large when the estimated gap is large relative to the current uncertainty, and small when the two paths are difficult to discriminate. Equivalently, a high ambiguity ratio $\mathcal{A}_t(\pi_p, \pi_{p'})$ indicates that the variance dominates the squared estimated gap, making the ranking of the pair unreliable under the current estimates. At each round $t$, GICA identifies the most ambiguous segment of the top-$K$ boundary by selecting the boundary arm $\pi_t^\star \in \arg\min_{\pi_p \in \widehat{P}_K(t)} \min_{\pi_{p'} \notin \widehat{P}_K(t)} G_{t-1}(\pi_p, \pi_{p'})$, the shortlisted path with the most ambiguous challenger and the hardest challenger $\pi_t^\dagger \in \arg\min_{\pi_{p'} \notin \widehat{P}_K(t)} G_{t-1}(\pi_t^\star, \pi_{p'})$, the non-shortlisted path that is hardest to separate from $\pi_t^\star$. The pair $(\pi_t^\star, \pi_t^\dagger)$ thus identifies the boundary pair with the smallest pairwise gap index, i.e., the pair for which the uncertainty most dominates the estimated gap, and directs all subsequent verification effort in that round.

### 3.3.3 Step Query Rule

Given the most ambiguous boundary pair $(\pi_t^\star, \pi_t^\dagger)$, GICA selects a single step $s_t$ from an admissible set $\mathcal{U}_t := \{s \in \mathcal{S} : s \in \pi_t^\star \cup \pi_t^\dagger\}$ to query via the reasoning-based PRM. The selection rule within $\mathcal{U}_t$ is identified via the following exact rank-one contraction result.

**Lemma 3.7** (Sherman–Morrison pairwise variance contraction)**.** *For any pair $(\pi_p, \pi_{p'}) \in \Pi^2$ and any queried step $s_t \in \mathcal{U}_t$, updating the design matrix as $V_{t+1} = V_t + x_{s_t} x_{s_t}^\top$ yields the exact identity*

$$\sigma_t^2(\pi_p, \pi_{p'}) - \sigma_{t+1}^2(\pi_p, \pi_{p'}) = \frac{\langle g(\pi_p, \pi_{p'}), x_{s_t} \rangle_{V_t^{-1}}^2}{1 + \|x_{s_t}\|_{V_t^{-1}}^2}. \tag{2}$$

*Proof.* The proof is given in the Appendix. $\square$

By Lemma 3.7, the pairwise variance $\sigma_t^2(\pi_p, \pi_{p'})$ decreases monotonically after each query, and the exact one-step reduction is given by Eq. (2). A step query is most informative for discriminating $(\pi_p, \pi_{p'})$ when the feature vector $x_{s_t}$ is well aligned with the pairwise feature direction $g(\pi_p, \pi_{p'})$ in the $V_t^{-1}$-weighted inner product space. It is worth noting that near-orthogonal steps yield negligible variance reduction. The denominator $1 + \|x_{s_t}\|_{V_t^{-1}}^2$ captures a saturation effect, i.e., the steps that already dominate the design matrix contribute diminishing returns under the rank-one update, reflecting the standard self-normalized concentration geometry of linear bandits (Abbasi-Yadkori et al., 2011). Therefore, motivated by Eq. (2), we define the *pairwise normalized correlation score* of a candidate step $s \in \mathcal{U}_t$ with respect to the boundary pair $(\pi_p, \pi_{p'})$ at round $t$ as $\mathcal{C}_t(s; \pi_p, \pi_{p'}) := \frac{\langle g(\pi_p, \pi_{p'}), x_s \rangle_{V_t^{-1}}^2}{1 + \|x_s\|_{V_t^{-1}}^2}$. By Lemma 3.7, $\mathcal{C}_{t-1}(s; \pi_p, \pi_{p'})$ equals exactly the

one-step reduction in $\sigma_{t-1}^2(\pi_p, \pi_{p'})$ that would be achieved by querying step $s$. Consequently, GICA adopts the greedy step-selection rule

$$s_t \in \arg\max_{s \in \mathcal{U}_t} \mathcal{C}_{t-1}(s; \pi_t^\star, \pi_t^\dagger), \tag{3}$$

which maximizes the one-step reduction in the pairwise variance $\sigma_{t-1}^2(\pi_t^\star, \pi_t^\dagger)$. When step $s_t$ is queried, the PRM evaluates it using its within-path prefix (as specified in Subsection 3.1), yielding the score $y_t$. The design matrix and parameter estimate are then updated according to Eq. (1).

### 3.3.4 Stopping Rule

GICA repeats the sampling-and-update procedure until the empirical top-$K$ shortlist $\widehat{P}_K(t)$ is statistically certified to be $\epsilon$-optimal in the sense of Definition 3.3. We encode this certification through the *worst-case lower confidence bound (LCB) gap*

$$\Gamma_t := \min_{\pi_p \in \widehat{P}_K(t)} \min_{\pi_{p'} \notin \widehat{P}_K(t)} \left( \widehat{\Delta}_t(\pi_p, \pi_{p'}) - W_t(\pi_p, \pi_{p'}) \right), \tag{4}$$

which, on the event $\mathcal{E}_\delta$, lower-bounds the smallest true gap between any shortlisted path and any challenger. The algorithm halts at the stopping time $\tau_\delta := \inf\{t \geq 0 : \Gamma_t \geq -\epsilon\}$ and outputs $\widehat{P}_K(\tau_\delta)$. The condition $\Gamma_t \geq -\epsilon$ guarantees that, at confidence level $1-\delta$, no path outside $\widehat{P}_K(t)$ exceeds any shortlisted path by more than $\epsilon$. Then, Proposition A.2 establishes the $\epsilon$-optimality of GICA's output in the sense of Definition 3.3. Algorithm 1 summarizes the complete procedure of GICA, while Figure 2 provides a detailed view of the algorithm in the context of the Selection Stage, as introduced in Figure 1. Concretely, each round of GICA proceeds as follows: it forms the empirical top-$K$ shortlist $\widehat{P}_K(t)$ from the current plug-in estimates (**lines 4–5**). The **Boundary Selection Rule** locates the most ambiguous boundary pair $(\pi_t^\star, \pi_t^\dagger)$ (**line 6**). The **Step Query Rule** queries the single step $s_t$ that maximally contracts the uncertainty along that boundary and observes its PRM score $y_t$ (**line 7**). The **Update Rule** then refreshes $(V_t, \widehat{\theta}_t)$ (**line 8**), and the **Stopping Rule** checks $\Gamma_t \geq -\epsilon$ (**line 9**), returning $\widehat{P}_K(\tau_\delta)$ once the shortlist is certified $\epsilon$-optimal.

---

**Algorithm 1** GICA: The Gap-Index Compositional Arm Framework

---

**Require:** $\Pi = \{\pi_1, \ldots, \pi_M\}$, $\delta \in (0, 1)$, $\lambda > 0$, $\epsilon \geq 0$, $K$.

1: **Initialize:** $t \leftarrow 0$, $V_0 \leftarrow \lambda I_d$, $\widehat{\theta}_0 \leftarrow 0$, $\Gamma_0 \leftarrow -\infty$.
2: **while** $\Gamma_t \leq -\epsilon$ **do**
3:     $t \leftarrow t + 1$.
4:     Compute $\widehat{\mu}_{t-1}(\pi_p) = g(\pi_p)^\top \widehat{\theta}_{t-1}$ for all $p \in [M]$.
5:     $\widehat{P}_K(t) \leftarrow \text{TopK}\left(\{\widehat{\mu}_{t-1}(\pi_p)\}_{p=1}^M, K\right)$.
6:     $\pi_t^\star \leftarrow \arg\min_{\pi_p \in \widehat{P}_K(t)} \min_{\pi_{p'} \notin \widehat{P}_K(t)} G_{t-1}(\pi_p, \pi_{p'})$, then $\pi_t^\dagger \leftarrow \arg\min_{\pi_{p'} \notin \widehat{P}_K(t)} G_{t-1}(\pi_t^\star, \pi_{p'})$.
7:     Select $s_t$ using Eq. (3) and observe the score $y_t = x_{s_t}^T \theta^* + \eta_t$.
8:     Update $V_t \leftarrow V_{t-1} + x_{s_t} x_{s_t}^\top$ and $\widehat{\theta}_t \leftarrow V_t^{-1} \sum_{i=1}^t y_i x_{s_i}$.
9:     $\Gamma_t \leftarrow \min_{\pi_p \in \widehat{P}_K(t)} \min_{\pi_{p'} \notin \widehat{P}_K(t)} \left( \widehat{\Delta}_t(\pi_p, \pi_{p'}) - W_t(\pi_p, \pi_{p'}) \right)$.
10: **end while**
11: **Output:** $\widehat{P}_K(t)$.

---

### 3.4 Sample Complexity Analysis

Theorem 3.8 bounds the number of verifier calls GICA needs to certify an $\epsilon$-optimal top-$K$ shortlist at confidence $1 - \delta$. Since Algorithm 1 queries one step per round, the stopping time $\tau_\delta$ equals the verifier-call count, so the bound measures verification cost directly. The sample complexity is related to gap-related quantities from Subsection 3.2. The first is the true boundary gap $\Delta_\mathcal{C}$, which measures the separation between the hardest top-$K$ and the challenger, and the second is the true global minimum gap. Assumption 3.5 guarantees $\Delta_\mathcal{C} > \epsilon \geq 0$, so the boundary is resolvable at tolerance $\epsilon$.

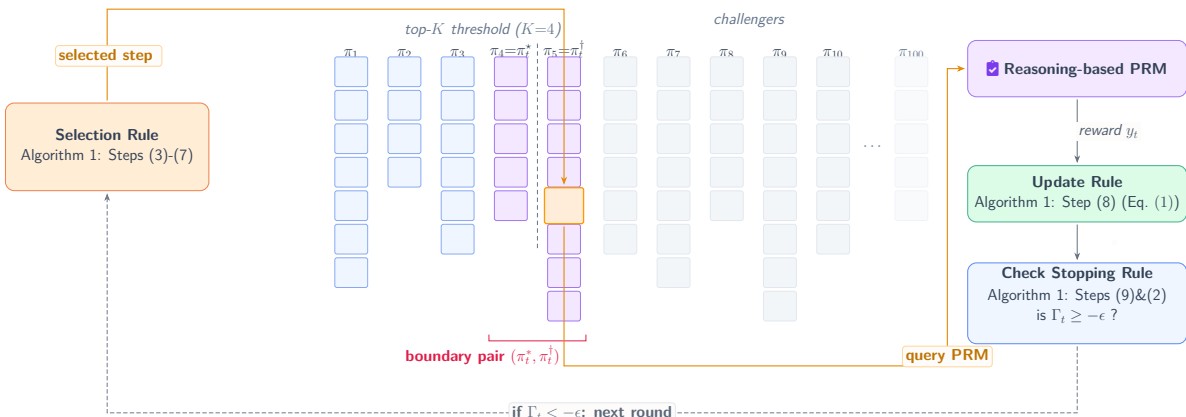

Figure 2: One round of GICA (Algorithm 1) within the **Selection Stage** of Figure 1: under the shared estimate $\widehat{\theta}_t$, the **Selection Rule** fixes the boundary pair $(\pi_t^\star, \pi_t^\dagger)$ and queries step $s_t$, the PRM returns $y_t$, and the estimates are updated until $\Gamma_t \geq -\epsilon$.

**Theorem 3.8** (Sample complexity of GICA). *Suppose Assumptions 3.1, 3.2, and 3.5 hold and fix $\delta \in (0,1)$. On the event $\mathcal{E}_\delta$, which holds with probability at least $1 - \delta$, the stopping time of Algorithm 1 satisfies*

$$\tau_\delta \leq 1 + \underbrace{\left\lceil \frac{16\,(\lambda + L^2)^2 \log(1 + L^2/\lambda)\,\bar\beta_{\tau_\delta}^2}{\rho^\dagger \lambda L^2\,(\Delta_{\min}^\Pi)^2}\, \max\left\{1, 4\log_+ \frac{2C_0}{\Delta_{\mathcal{C}}}\right\}\right\rceil}_{\text{deficit-crossing phase }\bar t_\star} + \underbrace{\left\lceil \frac{64\,(\lambda + L^2)\,\bar\beta_{\tau_\delta}^2}{\rho^\dagger \lambda\,\Delta_{\mathcal{C}}^2}\, \log\frac{4L^2\,\bar\beta_{\tau_\delta}^2}{\lambda\,\epsilon^2}\right\rceil}_{\text{contraction phase}}, \quad (5)$$

*where $D_0 = 2\log(1/\delta)$, $c_1 = L^2/((\lambda + L^2)\log(1 + L^2/\lambda))$, and $C_0 = \frac{4L}{\sqrt{\lambda}}(R + \sqrt{\lambda}S_0)\sqrt{\frac{2}{e\,\rho^\dagger c_1}}\,e^{\rho^\dagger c_1 D_0/4}$. Suppressing polylogarithmic factors, $\tau_\delta = \widetilde{O}\big(\frac{d\,(\lambda + L^2)}{\rho^\dagger \lambda}\big(\frac{\lambda + L^2}{L^2(\Delta_{\min}^\Pi)^2} + \frac{1}{\Delta_{\mathcal{C}}^2}\big)\big)$.*

*Proof sketch.* The full proof is in Appendix A.3. We work throughout on the event $\mathcal{E}_\delta$, which holds with probability at least $1 - \delta$ by Lemma A.1. We first reduce the gap-index stopping rule to a shortlist-independent variance criterion, then bound how fast the pairwise variances contract under GICA's greedy step selection. By the empirical ordering (Corollary A.3) and the non-negativity of $W_t$, the condition $W_t(\pi_p, \pi_{p'}) \leq \epsilon$ suffices for stopping, so it is enough to drive every pairwise variance $\sigma_t^2(\pi_p, \pi_{p'})$ below $\epsilon^2/\beta_t(\delta)^2$. Combining the Sherman–Morrison identity (Lemma 3.7), the pair–step correlation (Assumption 3.2), and an algorithm-induced lower bound (Lemma A.7) gives a per-round multiplicative contraction $\sigma_t^2(\pi_p, \pi_{p'}) \leq (1 - \kappa_t)\,\sigma_{t-1}^2(\pi_p, \pi_{p'})$ for every pair (Lemma A.8). Each variance shrinks every round, but the rate $\kappa_t \in [0,1)$ may be arbitrarily small and is bounded away from zero only once the confidence deficit $2\beta_{t-1}(\delta)\widetilde{M}_{t-1}$ drops below $\Delta_{\mathcal{C}}/2$. Lemma A.10 shows the all-pairs half-width $\widetilde{M}_t$ decays in the log-determinant, so this crossing occurs after a deterministic deficit-crossing time $t_\star$, which Lemma A.12 bounds by a closed-form quantity $\bar t_\star$. For all $t_\star \leq t \leq \tau_\delta$ the rate is floored by $\kappa_{\text{cf}} > 0$ (Lemma A.10(ii)), giving uniform exponential decay of every pairwise variance (Lemma A.13). Equating this decay with the stopping threshold shows the contraction phase has length scaling as $\kappa_{\text{cf}}^{-1} \log(4L^2\beta_{\tau_\delta}(\delta)^2/(\lambda\epsilon^2))$. Summing $\bar t_\star$ and this length yields Eq. (5), and $\mathbb{P}(\mathcal{E}_\delta) \geq 1 - \delta$ gives the probability statement. $\square$

The bound splits into a deficit-crossing phase $\bar t_\star$ and a contraction phase. The deficit-crossing phase is the start-up cost until the all-pairs confidence deficit $2\beta_{t-1}(\delta)\widetilde{M}_{t-1}$ falls below $\Delta_{\mathcal{C}}/2$ and floors the contraction rate. It carries the same confidence radius $\beta_{\tau_\delta - 1}(\delta)^2$ as the contraction phase and scales as $1/(\Delta_{\min}^\Pi)^2$ in the global minimum gap, depending on $\Delta_{\mathcal{C}}$ only logarithmically and not at all on $\epsilon$. The contraction phase carries the dominant gap-dependent rate $1/\Delta_{\mathcal{C}}^2$, with the dimension $d$ entering only linearly through $\beta_{\tau_\delta}(\delta)^2 = O(d\log(\cdot) + \log\frac{1}{\delta})$ and the tolerance $\epsilon$ only logarithmically, which excludes exact identification ($\epsilon = 0$). The pair–step correlation $\rho^\dagger$ multiplies the contraction rate, giving an overall $1/\rho^\dagger$ scaling, with its sensitivity examined in Subsection 4.5. Notably, the bound has no dependence on the candidate-set size $M$,

unlike standard top-$K$ bandit algorithms whose cost grows with $M$, reflecting the information shared across compositional arms through $\theta^\star$.

## 4 Experiments and Results

We validate GICA through two complementary protocols. The synthetic protocol isolates the bandit algorithm's behavior on compositional top-K instances with known ground-truth parameters, while the TTS protocol evaluates end-to-end performance as a process-level verification strategy across mathematical reasoning benchmarks. While our approach can also aid in other domains, such as coding, planning tasks (Lin et al., 2026), and Question Answering (QA), we primarily focus on mathematical reasoning due to the availability of stable pre-trained process verifiers as noted by prior works (Zeng et al., 2025; Khalifa et al., 2026). Since our main goal is to make the process-level verification sample-efficient (not task-specific) and our source of reward signal is the PRM, we choose tasks like math reasoning, which are sufficiently explored and where stable PRMs are available. This aids in avoiding confounding effects from unstable PRMs in underexplored domains. However, our algorithm is domain-agnostic and can be easily extended to other domains. We open-source our implementation code to facilitate future research.[2] This two-stage design decouples algorithmic sample-efficiency from downstream task performance, following standard practice for fixed-confidence top-K algorithms in linear bandits (Réda et al., 2021; Purohit et al., 2025). The evaluation addresses three research questions. **RQ1** - Does GICA achieve greater sample efficiency than state-of-the-art linear bandit methods for top-K identification over compositional arms. **RQ2** - How does it compare to existing bandit-based verification strategies in average inference runtime and verifier calls per query within a realistic TTS pipeline? **RQ3**- Can it select top-K reasoning paths while minimizing verifier calls without degrading downstream accuracy relative to exhaustive verification.

### 4.1 Experimental Setup

We instantiate the compositional linear model of Subsection 3.2 for the **synthetic evaluation**. For each problem scale $M \in \{200, 500, 1000\}$, each of the $M$ paths draws its length independently and uniformly from $\{20, \ldots, 80\}$ steps, with step features sampled from an isotropic Gaussian in $\mathbb{R}^8$ and a controlled boundary gap $\Delta_{\mathcal{C}}$ at the rank-$K$ threshold. Step queries return $y_t = x_{s_t}^\top \theta^\star + \eta_t$ with $\eta_t \sim \mathcal{N}(0, R^2)$ and $R = 0.1$, and we target $K = 10$. Full data-generation and hyperparameter details are given in Appendix B.1.

For the **TTS evaluation**, we consider three math-reasoning benchmarks of increasing difficulty: MATH-500 (Hendrycks et al., 2021; Lightman et al., 2023), MathOdyssey (Fang et al., 2024), and the 400 problems from the 1983–2006 editions of AIME. Reasoning paths are generated by two open-weight math LLMs, DeepSeekMath-RL-7B (Shao et al., 2024) and InternLM2-Math-Plus-7B (Ying et al., 2024), with $M = 100$ paths per problem at temperatures in $\{1.0, 1.1\}$ for the generator, following prior works (Snell et al., 2025; Lightman et al., 2023; Zhou et al., 2025). Steps are extracted via newline delimiters, with ThinkPRM-1.5B and ThinkPRM-7B (Khalifa et al., 2026) serving as process-level reasoning-based verifiers. Furthermore, Appendix C presents an ablation study evaluating the ThinkPRM-7B verifier. Each step $s$ utilizes a $d$-dimensional feature $x_s$ (derived from a frozen sentence encoder `all-MiniLM-L6-v2` and $\ell_2$-normalized to $L = 1$) shared across all bandit methods. For both the selected top-$K$ paths ($K = 5$) and Best-of-$M$, the final answer is determined via majority vote over aggregated step-level scores. See Appendix B for full details.

**Baselines:** We compare GICA against two reference points and three linear bandit baselines. Top-1 decoding directly generates a single reasoning path without TTS or verification. Best-of-$M$ exhaustively samples PRM signals for every step, serving as the upper bound for downstream accuracy attainable with the chosen (generator, PRM) pair. More details of Top-1 decoding and Best-of-$M$ are provided in Appendix B. Among adaptive methods, we include M-LINGAPE (Xu et al., 2018), LINGIFA (Réda et al., 2021), and CASE (Purohit et al., 2025), each adapted to the compositional setting by using the path features $g(\pi_p)$ of Subsection 3.2 as virtual arms and querying at the step level. All bandit methods share the same $(\delta, \lambda, \epsilon, R, S_0)$, so any differences reflect the sampling rule rather than the stopping condition. The exact values are provided in Appendix B.1. We also compare (in terms of task performance) to exhaustive scoring using ORM (Cobbe

---

[2]https://anonymous.4open.science/r/GICA-1B57

et al., 2021) and ORM-PRM cascade, where the top-20 paths scored by ORM are re-ranked by PRM. For the TTS setup, we report accuracy (the fraction of problems whose predicted answer matches the ground truth under **Exact Match**: EM), average verifier calls per query as a measure of sample efficiency, and average inference runtime per query. For the *synthetic setup*, we measure the total number of gap-index comparisons, the average number of verifier calls, and the average runtime per simulation.

## 4.2 Sample Efficiency of GICA on Compositional Arms in Synthetic Setup (RQ1)

To answer **RQ 1**, we compare GICA against CASE, LINGIFA, and M-LINGAPE in the synthetic setup of Subsection 4.1, varying $M \in \{200, 500, 1000\}$ (Figure 3). In per-round gap-index comparisons (Figure 3a), GICA consistently issues the fewest, reducing the count over the strongest baseline CASE by **61.0×**, **9.6×**, and **2.0×** at $M = 200, 500, 1000$, respectively. The same trend holds for total step-level queries (Figure 3b), where GICA requires roughly an order of magnitude fewer verifier calls than LINGIFA and M-LINGAPE, and significantly fewer than CASE at $M = 1000$. These savings transfer to runtime (Figure 3c), yielding a **27.3×** speedup over CASE at $M = 1000$, with larger gains relative to LINGIFA and M-LINGAPE.

*Insight 1. On synthetic compositional top-K identification, GICA jointly reduces gap-index comparisons and verifier calls, with verifier call counts nearly an order of magnitude below LINGIFA and M-LINGAPE at $M = 200$ and a* **27.3×** *runtime speedup over the strongest baseline at $M = 1000$.*

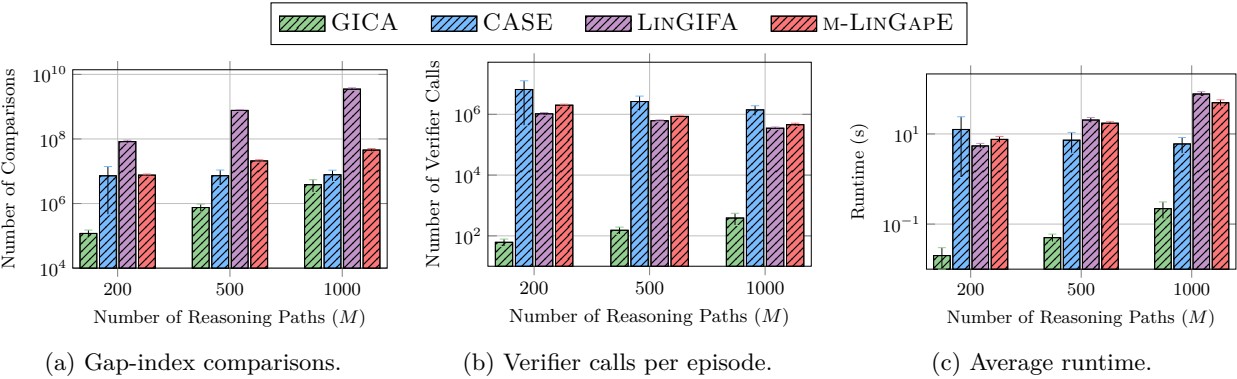

(a) Gap-index comparisons.     (b) Verifier calls per episode.     (c) Average runtime.

Figure 3: Comparison of GICA to state-of-the-art linear-stochastic bandit algorithms in the synthetic setup.

## 4.3 Inference Runtime and Verifier Calls in the TTS Pipeline (RQ2)

We compare average per-query inference runtime and verifier calls across settings in Figure 4, with DeepSeek-7B (a,b) and InternLM2-7B (c,d) as generator and ThinkPRM-1.5B as verifier. The runtime results on verifier calls for ablations with ThinkPRM-7B as verifier are in Figures 5. From Figures 4a–4c, GICA attains **3.5×** and **2.5×** speedups over CASE and LINGIFA, respectively, on MATH-500 with DeepSeek-7B, and **4.3×** and **3.1×** speedups on MATH-500 with InternLM2-Math-Plus-7B, with the relative ordering preserved on MathOdyssey and AIME despite absolute inference runtime growing with benchmark difficulty. These gains arise from earlier convergence, i.e., GICA's sampling rule selects the most informative step per round, exploiting cross-path step correlation through the shared parameter $\widehat{\theta}_t$ to sharpen utility estimates and reduce verifier calls. It is evidenced by the verifier call counts in Figures 4b–4d, which mirror and explain the inference runtime results. For instance, GICA achieves **4.2×** and **2.9×** reductions over CASE and LINGIFA on MATH-500 with DeepSeek-7B, with similar gains on MathOdyssey and AIME. Although CASE carries stronger sample-efficiency guarantees than LINGIFA in linear stochastic settings, on MATH-500 it issues more verifier calls than LINGIFA, since its challenger-shortlist heuristic fails to exploit the step-level correlation induced by $\widehat{\theta}_t$ in compositional arms. On MathOdyssey and AIME, the two baselines are comparable in terms of verifier calls and inference runtime, while **GICA retains a clear advantage** and maintains stable efficiency gains across two generator families and three benchmarks. Both evaluation dimensions indicate that the improvement is intrinsic to the sampling rule rather than to a specific generator or dataset. The end-to-end gains in Figure 4 are smaller than the raw algorithmic speedups in Figure 3 because end-to-end

latency includes PRM forward passes whose per-call cost is identical across methods. GICA reduces precisely the number of such passes, which translates linearly into inference runtime for the reasoning-based verifier, so the verifier call reductions can be read as the inference runtime reductions.

***Insight 2.*** *In a realistic TTS setting, GICA reduces verifier calls by up to* **4.2×** *and per-query inference runtime by up to* **4.3×** *relative to the strongest bandit baseline, with the relative gains preserved across both generator models and all three benchmarks.*

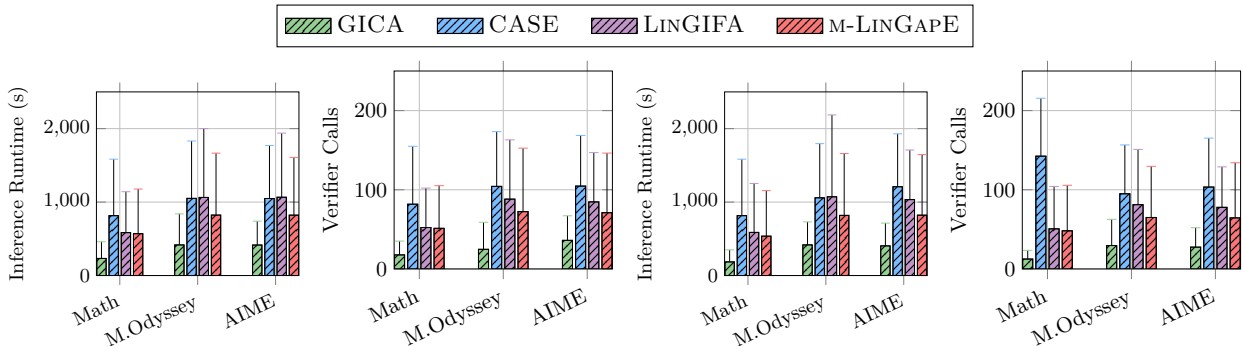

(a) Runtime (DeepSeek).   (b) Verifier calls (DeepSeek).   (c) Runtime (InternLM2).   (d) Verifier calls(InternLM2).

Figure 4: Sample efficiency of GICA compared to state-of-the-art linear-stochastic bandit algorithms across DeepSeek-7B and InternLM2-7B generators, with ThinkPRM-1.5B as verifier.

### 4.4 Downstream Task Accuracy (RQ3)

To answer **RQ 3**, we examine the downstream accuracy of GICA relative to the bandit baselines and the exhaustive Best-of-$M$ upper bound in Table 1 across MATH-500, MathOdyssey, and AIME for two generators. We observe that ORM based baselines underperform Note that the goal of GICA and the baselines is not to exceed Best-of-$M$ but to approach it while eliminating the majority of verifier calls and inference runtime. On DeepSeekMath-RL-7B, Best-of-$M$ exceeds Top-1 by 12.10, 13.62, and 5.00 points on the three benchmarks, and on InternLM2-Math-Plus-7B by 36.40, 19.42, and 6.50 points, confirming that process-level verification is the dominant source of accuracy gain and justifying the adaptive-verification setup. All three bandit baselines and GICA substantially outperform Top-1 across every benchmark and both generators, showing that none degrade due to premature stopping or collapse and that the linear-stochastic bandit formulation is well-posed for process-level verification at scale. Among baselines, M-LinGapE typically achieves the lowest accuracy because it is not defined entirely in terms of gap indices and cannot exploit tighter gap bounds or more aggressive stopping rules, while CASE occasionally underperforms LinGIFA because its challenger sub-sampling can discard optimal paths. None of the baseline models the compositional structure of reasoning paths since verification is performed strictly at the step level. Hence, GICA outperforms other baselines on 2/3 benchmarks. On DeepSeekMath-RL-7B, GICA attains 50.72%, 31.08%, and 9.59% on MATH-500, MathOdyssey, and AIME, exceeding the next-best baseline by 0.32, 5.63, and 0.46 points, respectively, and on InternLM2-Math-Plus-7B reaches 51.20%, 25.60%, and 8.82%, again surpassing the strongest baseline. Notably, GICA matches the Best-of-$M$ bound to within one accuracy point on MathOdyssey with DeepSeekMath-RL-7B (31.08% vs. 31.10%) and on MATH-500 with InternLM2-Math-Plus-7B (51.20% vs. 51.60%), achieved by combining a sampling mechanism that models compositionality with path-level gap indices updated from step-level rewards.

***Insight 3.*** *GICA attains downstream accuracy closest to the exhaustive Best-of-M upper bound across all three benchmarks and both generators. Combined with Figure 4, modeling the compositional structure delivers good accuracy of process-level verification at a fraction of the verification cost.*

Table 1: Exact match across datasets with ThinkPRM-1.5B as verifier. Second-highest scores are underlined.

| Method | Deepseek-MATH-RL-7B | | | InternLM2-MATH-PLUS-7B | | |
| --- | --- | --- | --- | --- | --- | --- |
| | MATH-500 | MathOdyssey | AIME | MATH-500 | MathOdyssey | AIME |
| **Verification** | | | | | | |
| Top-1 decoding | 41.00 | 17.48 | 6.50 | 15.20 | 6.43 | 3.50 |
| Best-of-M (Exhaustive) | 53.10 | 31.10 | 11.50 | 51.60 | 25.85 | 10.00 |
| ORM | 44.88 | 22.68 | 7.69 | 39.08 | 18.25 | 5.25 |
| ORM-PRM cascade | 46.01 | 26.11 | 8.89 | 44.80 | 21.59 | 7.75 |
| **Bandit Approaches** | | | | | | |
| CASE | 48.11 | 28.53 | 9.13 | 49.84 | 21.33 | 9.06 |
| M-LINGAPE | 47.80 | 27.24 | 7.67 | 48.80 | 25.40 | 8.50 |
| LINGIFA | 50.40 | 25.45 | 10.00 | 50.20 | 22.36 | 8.50 |
| **Our Approach** | | | | | | |
| **GICA** | 50.72 | 31.08 | 9.59 | 51.20 | 25.60 | 8.82 |

### 4.5 Sensitivity to Pair–Step Alignment As the reviewer requested, this subsection is moved from the Appendix to the main pages.

The order form of Theorem 3.8 (Eq. (70)) depends on the pair–step correlation $\rho^\dagger$ of Assumption 3.2 as $\tau_\delta = \widetilde{O}(1/\rho^\dagger)$, in both the deficit-crossing and the contraction phase. Mechanistically, $\rho^\dagger$ multiplies the per-round contraction rate of Lemma A.8, $\kappa_t = \rho^\dagger c_0 (\Delta_\mathcal{C} - 2\beta_{t-1}(\delta)\widetilde{M}_{t-1})_+^2 / (4\beta_{t-1}(\delta)^2)$, and hence its envelope $\bar{\kappa}_t = \rho^\dagger c_0 \Delta_\mathcal{C}^2 / (4\beta_{t-1}(\delta)^2)$ and the contraction phase floor $\kappa_{\mathrm{cf}} = \rho^\dagger c_0 \Delta_\mathcal{C}^2 / (16\beta_{\tau_\delta-1}(\delta)^2)$. The number of rounds to certify an $\epsilon$-optimal shortlist scales as $1/\kappa_{\mathrm{cf}} \propto 1/\rho^\dagger$, i.e., stronger pair–step alignment yields larger simultaneous variance reduction across all pairs and faster termination. Table 6 reports the runtime and verifier calls of GICA for three values of $\rho^\dagger$ in the synthetic setup of Subsection 4.1, with $d$, $\lambda$, $R$, $\epsilon$, and $\delta$ fixed at the values of Appendix B.1.

| $\rho^\dagger$ | **Runtime (s)** | **Verifier Call** |
| --- | --- | --- |
| $9.5619 \times 10^{-22}$ | 34.90 | 33037 |
| $8.1943 \times 10^{-20}$ | 21.81 | 19406 |
| $2.4104 \times 10^{-18}$ | 5.52 | 5358 |

Table 2: Sensitivity of GICA runtime and verifier call to the pair–step correlation constant $\rho^\dagger$ in the synthetic setup.

The predicted monotonicity holds, i.e., as $\rho^\dagger$ rises from $9.56 \times 10^{-22}$ to $2.41 \times 10^{-18}$, verifier calls fall from 33,037 to 5,358 and runtime from $34.90\,\mathrm{s}$ to $5.52\,\mathrm{s}$, matching the direction of the $1/\rho^\dagger$ factor in Eq. (70) and the gains in runtime are primarily due to a reduction in the number of verifier calls. The empirical scaling is, however, far milder than worst case, i.e., $\rho^\dagger$ spans over three orders of magnitude (a factor of $\approx 2.5 \times 10^3$) while verifier calls change only $\approx 6.2\times$. Because $\rho^\dagger$ is a uniform infimum of squared cosines over all pairs and geometries (Assumption 3.2), it is set by a few adversarial configurations, whereas the ambiguous boundary pairs that actually bottleneck termination (Subsection 3.3.2) are aligned far more favourably. The guarantee thus remains valid even for vanishingly small $\rho^\dagger$, while practical cost is governed by typical-case alignment.

## 5 Conclusion

We addressed the prohibitive cost of process-level verification in TTS by recasting it as a fixed-confidence top-$K$ identification problem over compositional arms. Building on this formulation, we introduced GICA, a gap-index bandit framework with a fixed-confidence sample-complexity guarantee. Empirically, GICA matches the exhaustive Best-of-$M$ accuracy upper bound while substantially reducing verifier calls and inference runtime across three mathematical reasoning benchmarks and two generator-verifier model families, showing that modeling compositional structure makes fine-grained step-level supervision practical at scale. Future directions include extending GICA to integrating adaptive path generation, evaluating dedicated long

thinking generators with extended and self-revising paths, and applying GICA to broader domains such as code generation and agentic tasks.

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

# A   Theoretical Analysis

This appendix develops the full theoretical analysis behind GICA's sample-complexity guarantee. It builds the high-probability confidence event, certifies the correctness of the returned shortlist, and then tracks how the pairwise variances contract round by round until the stopping rule fires, assembling these pieces into the two-phase bound of Theorem 3.8. Alongside the main theorem and the lemmas stated in Section 3, it records the formal statements and proofs of the supplementary lemmas and the proposition referenced there.

## A.1   Notation and Preliminaries

This subsection fixes the notation used throughout the analysis and collects the standard algebraic facts that the proofs rely on. Tables 3 and 4 organize the model quantities, estimators, and instance geometry on one side, and the derived constants, contraction rates, and deficit-crossing thresholds on the other, so that every symbol appearing later has a single point of reference.

Table 3: Summary of notation, Part I: model quantities, estimators, and instance geometry.

| Notation | Description |
|---|---|
| $\Pi = \{\pi_1, \ldots, \pi_M\}$ | Set of candidate reasoning paths. |
| $\mathcal{S}$ | Set of all steps. |
| $x_s \in \mathbb{R}^d$ | Feature vector of step $s \in \mathcal{S}$, $\|x_s\|_2 \leq L$. |
| $\theta^\star \in \mathbb{R}^d$ | Unknown parameter, $\|\theta^\star\|_2 \leq S_0$. |
| $g(\pi_p) = \frac{1}{T_p} \sum_{q=1}^{T_p} x_{s_{p,q}}$ | Path feature. |
| $g(\pi_p, \pi_{p'}) = g(\pi_p) - g(\pi_{p'})$ | Pairwise feature difference. |
| $V_t = \lambda I_d + \sum_{i=1}^t x_{s_i} x_{s_i}^\top$ | Regularized design matrix. |
| $\widehat{\theta}_t = V_t^{-1} \sum_{i=1}^t y_i\, x_{s_i}$ | Ridge estimator. |
| $\sigma_t^2(\pi_p, \pi_{p'}) = \|g(\pi_p, \pi_{p'})\|_{V_t^{-1}}^2$ | Pairwise variance. |
| $X_{1:t},\, \eta_{1:t},\, y_{1:t},\, S_t = X_{1:t}^\top \eta_{1:t}$ | Stacked design, noise, observations. |
| $\rho^\dagger \in (0, 1]$ | Pair–step correlation constant. |
| $\mathcal{C}_K^* = \{(\pi_p, \pi_{p'}) : \pi_p \in P_K^*,\, \pi_{p'} \notin P_K^*\}$ | True boundary pair set. |
| $\Delta_{\mathcal{C}} = \min_{(\pi_p, \pi_{p'}) \in \mathcal{C}_K^*} \Delta(\pi_p, \pi_{p'})$ | Min gap over true boundary pairs only. |
| $\Delta_{\min}^\Pi = \min_{(\pi_p, \pi_{p'}) \in \Pi^2,\, p \neq p'} |\Delta(\pi_p, \pi_{p'})|$ | Global minimum gap over distinct path pairs ($> 0$ under distinct path utilities). |
| $\underline{\Delta}_t = \max\{(\Delta_{\mathcal{C}} - 2\beta_{t-1}(\delta)\widetilde{M}_{t-1})_+,\, \Delta_{\min}^\Pi\}$ | Floored boundary gap ($\geq \Delta_{\min}^\Pi > 0$). |
| $W_t(\pi_p, \pi_{p'}) = \beta_t(\delta)\, \sigma_t(\pi_p, \pi_{p'})$ | Pairwise confidence width. |
| $G_t(\pi_p, \pi_{p'}) = \widehat{\Delta}_t(\pi_p, \pi_{p'})^2 / \sigma_t^2(\pi_p, \pi_{p'})$ | Pairwise gap index. |
| $B_t(\pi_p, \pi_{p'}) = \widehat{\Delta}_t(\pi_p, \pi_{p'}) + W_t(\pi_p, \pi_{p'})$ | Upper gap index. |
| $\widetilde{M}_t = \max_{g(\pi_p, \pi_{p'}) \neq 0} \|g(\pi_p, \pi_{p'})\|_{V_t^{-1}}$ | All-pairs maximal half-width. |

Furthermore, the following standard facts are invoked in the proofs (Horn & Johnson, 2012):

(i) **Norm duality:** For any positive-definite $A \in \mathbb{R}^{d \times d}$ and $b \in \mathbb{R}^d$, $\|A^{-1}b\|_A = \|b\|_{A^{-1}}$.

(ii) **Cauchy–Schwarz:** For any $a, b \in \mathbb{R}^d$ and positive-definite $A$, $|a^\top b| \leq \|a\|_{A^{-1}} \|b\|_A$.

(iii) **Loewner monotonicity:** If $A \succeq B \succ 0$, then $B^{-1} \succeq A^{-1}$, and $\|v\|_{A^{-1}} \leq \|v\|_{B^{-1}}$ for all $v$.

Table 4: Summary of notation, Part II: log-determinant quantities, contraction rates, and deficit-crossing thresholds.

| Notation | Description |
|---|---|
| $c_0 = \lambda/(4(\lambda + L^2))$ | A constant factor appearing in the proofs. |
| $D_t = 2\log(\det(V_t)^{1/2}\det(\lambda I_d)^{-1/2}/\delta)$ | Log-determinant complexity. |
| $\ell_t = D_t - D_{t-1} = \log(1 + \|x_{s_t}\|^2_{V_{t-1}^{-1}})$ | Per-round log-det increment. |
| $c_1 = L^2/((\lambda + L^2)\log(1 + L^2/\lambda))$ | Geometry constant (Lemma A.10). |
| $\kappa_t = \dfrac{\rho^\dagger c_0\left(\Delta_{\mathcal{C}} - 2\beta_{t-1}(\delta)\,\widetilde{M}_{t-1}\right)^2_+}{4\,\beta_{t-1}(\delta)^2}$ | Per-round contraction rate. |
| $\bar{\kappa}_t = \dfrac{\rho^\dagger c_0\,\Delta_{\mathcal{C}}^2}{4\,\beta_{t-1}(\delta)^2}$ | Contraction-rate envelope. |
| $\kappa_{\min} = \dfrac{\rho^\dagger c_0\left(\Delta_{\mathcal{C}} - 2\beta_{t_\star - 1}(\delta)\widetilde{M}_{t_\star - 1}\right)^2}{4\,\beta_{\tau_\delta - 1}(\delta)^2}$ | Contraction floor. |
| $C_0 = \dfrac{4L}{\sqrt{\lambda}}(R + \sqrt{\lambda}S_0)\sqrt{\dfrac{2}{e\,\rho^\dagger c_1}}\,e^{\rho^\dagger c_1 D_0/4}$ | Deficit-crossing time prefactor. |
| $\kappa_{\mathrm{cf}} = \dfrac{\rho^\dagger c_0\,\Delta_{\mathcal{C}}^2}{16\,\beta_{\tau_\delta - 1}(\delta)^2}$ | $\kappa_{\min}$'s floor. |
| $\log_+(z) = \max(\log z,\, 0)$ | Truncated logarithm. |
| $L_\lambda = \log(1 + L^2/\lambda)$ | A constant factor. |
| $T_{\mathrm{def}} = \dfrac{4}{\rho^\dagger c_1}\log_+\dfrac{C_0}{\Delta_{\mathcal{C}}}$ | Deficit-crossing threshold. |
| $T_\star = \max\{\frac{1}{\rho^\dagger c_1},\, \frac{4}{\rho^\dagger c_1}\log_+\frac{2C_0}{\Delta_{\mathcal{C}}}\}$ | Monotonicity deficit-crossing level. |
| $\ell_{\min} = \dfrac{c_0(\Delta^{\Pi}_{\min})^2}{4\beta_{\tau_\delta - 1}(\delta)^2}$ | Uniform per-round log-det increment floor. |
| $t_\star = \inf\{t \geq 1:\ D_{t-1} \geq T_\star\}$ | Monotonicity deficit-crossing time. |
| $\bar{t}_\star = 1 + \lceil T_\star/\ell_{\min}\rceil$ | Closed-form deficit-crossing time bound. |

(iv) **Regularization bound:** Since $V_t \succeq \lambda I_d$ for all $t \geq 0$, we have $V_t^{-1} \preceq \lambda^{-1}I_d$, hence $\|v\|^2_{V_t^{-1}} \leq \|v\|^2_2/\lambda$ for all $v \in \mathbb{R}^d$.

(v) **Feature-norm bound:** For any path $\pi_p \in \Pi$, $\|g(\pi_p)\|_2 \leq L$, and for any pair $(\pi_p, \pi_{p'})$, $\|g(\pi_p, \pi_{p'})\|_2 \leq 2L$.

(vi) **Leverage bound:** For any $s \in \mathcal{S}$ and $t \geq 0$, $\|x_s\|^2_{V_t^{-1}} \leq L^2/\lambda$.

(vii) **Reciprocal inequality:** For $x \in [0, 1)$, $\frac{1}{1-x} \geq 1 + x$.

(viii) **Sherman–Morrison formula:** For positive-definite $A$ and vector $v$, $(A + vv^\top)^{-1} = A^{-1} - \frac{A^{-1}vv^\top A^{-1}}{1 + v^\top A^{-1}v}$.

(ix) **Triangle inequality:** For any positive-definite $A \in \mathbb{R}^{d \times d}$ and $a, b \in \mathbb{R}^d$, $\|a + b\|_A \leq \|a\|_A + \|b\|_A$; in particular, $\|a + b\|_2 \leq \|a\|_2 + \|b\|_2$.

(x) **Elementary exponential bound:** For all $x \in [0, 1]$, $1 - x \leq e^{-x}$.

(xi) **Quadratic-mean inequality:** For all $a, b \in \mathbb{R}$, $(a + b)^2 \leq 2a^2 + 2b^2$.

(xii) **Determinant monotonicity:** If $0 \prec A \preceq B$, then $\det A \leq \det B$, and consequently $\log\det A \leq \log\det B$.

(xiii) **Bilinearity of the weighted inner product:** For any positive-definite $A$, $\langle \cdot, \cdot \rangle_{A^{-1}}$ is bilinear; in particular $\langle \sum_i \alpha_i u_i, v \rangle_{A^{-1}} = \sum_i \alpha_i \langle u_i, v \rangle_{A^{-1}}$.

(xiv) **Rank-one determinant identity:** For positive-definite $A$ and vector $v$, $\det(A + vv^\top) = \det(A)\,(1 + \|v\|_{A^{-1}}^2)$.

(xv) **Monotone positive-part squaring:** For $a \geq b$ with $b \geq 0$, $(a)_+^2 \geq (b)_+^2 \geq b^2$; and $z \mapsto (z)_+^2$ is non-decreasing.

(xvi) **Logarithm bound:** For all $u \geq 0$, $\log(1 + u) \leq u$.

(xvii) **Square-root concavity:** For $a \geq b > 0$, $\sqrt{a} - \sqrt{b} \leq (a - b)/(2\sqrt{b})$.

(xviii) **Monotone variance:** Since $V_t \succeq V_{t-1}$, for every fixed $v$, $\|v\|_{V_t^{-1}}^2 \leq \|v\|_{V_{t-1}^{-1}}^2$; in particular $\sigma_t^2(\pi_p, \pi_{p'}) \leq \sigma_{t-1}^2(\pi_p, \pi_{p'})$.

(xix) **AM–GM determinant bound:** If $V_u \preceq (\lambda + uL^2)I_d$, then by AM–GM on its eigenvalues $\det V_u \leq (\lambda + uL^2)^d$, hence $\log \det V_u \leq d \log(\lambda + uL^2)$.

(xx) **Sandwich theorem:** Let $a, b, c$ be real-valued and satisfy $a_t \leq b_t \leq c_t$ along a limiting process in $t$ (e.g. $t \to \infty$). If $\lim a_t = \lim c_t = \ell$, then $\lim b_t = \ell$. In particular, if $0 \leq b_t \leq c_t$ and $c_t \to 0$, then $b_t \to 0$.

## A.2 Confidence Bounds and Correctness

This subsection establishes the high-probability event underlying the entire analysis, together with the algebraic identity that drives every contraction argument below. Lemma A.1 constructs the self-normalized confidence event $\mathcal{E}_\delta$, on which the estimated pairwise gaps concentrate around their true values, and shows that it holds with probability at least $1 - \delta$. On this event, Proposition A.2 certifies that the shortlist returned at the stopping time is $\epsilon$-optimal. Finally, Lemma 3.7 records the exact Sherman–Morrison recursion for the one-step contraction of the pairwise variance, the identity reused throughout the remaining subsections. All subsequent statements are made on the event $\mathcal{E}_\delta$.

The following lemma builds the self-normalized confidence event $\mathcal{E}_\delta$ and shows that, on it, every step estimate, path estimate, and pairwise gap estimate stays within an explicit width of its true value, uniformly over all rounds. This is the concentration backbone for the correctness and sample-complexity arguments.

**Lemma A.1** (Uniform confidence bounds). *Fix $\delta \in (0, 1)$ and let $\mathcal{E}_\delta$ be as in Definition 3.6. Under the linear model and the $R$-sub-Gaussian noise assumption, $\mathbb{P}(\mathcal{E}_\delta) \geq 1 - \delta$, where*

$$\beta_t(\delta) \;=\; R\sqrt{2\log\!\Big( \tfrac{\det(V_t)^{1/2}\det(\lambda I)^{-1/2}}{\delta} \Big)} + \sqrt{\lambda}\,S_0. \tag{6}$$

*Moreover, on $\mathcal{E}_\delta$, for any pair of $t \geq 0$, $s \in \mathcal{S}$, and $\pi_p \in \Pi$,*

$$\big|\widehat{\mu}_t(s) - \mu(s)\big| \;=\; \big|x_s^\top(\widehat{\theta}_t - \theta^\star)\big| \;\leq\; \beta_t(\delta)\,\|x_s\|_{V_t^{-1}}, \qquad \big|\widehat{\mu}_t(\pi_p) - \mu(\pi_p)\big| \;\leq\; \beta_t(\delta)\,\|g(\pi_p)\|_{V_t^{-1}}, \tag{7}$$

*and for all $(\pi_p, \pi_{p'}) \in \Pi^2$, the estimated pairwise gap concentrates around the true gap:*

$$\big|\widehat{\Delta}_t(\pi_p, \pi_{p'}) - \Delta(\pi_p, \pi_{p'})\big| \;\leq\; W_t(\pi_p, \pi_{p'}). \tag{8}$$

*Proof.* We write the stacked quantities $X_{1:t} \in \mathbb{R}^{t \times d}$, $\eta_{1:t} \in \mathbb{R}^t$, $y_{1:t} \in \mathbb{R}^t$ as in Table 3, so that $y_{1:t} = X_{1:t}\theta^\star + \eta_{1:t}$ and $V_t = \lambda I_d + X_{1:t}^\top X_{1:t}$. Since $\widehat{\theta}_t = V_t^{-1}X_{1:t}^\top y_{1:t}$, substituting $y_{1:t} = X_{1:t}\theta^\star + \eta_{1:t}$ gives

$$\begin{aligned}
\widehat{\theta}_t &= V_t^{-1}X_{1:t}^\top X_{1:t}\,\theta^\star + V_t^{-1}X_{1:t}^\top \eta_{1:t} \\
&= V_t^{-1}\big(V_t - \lambda I_d\big)\theta^\star + V_t^{-1}S_t \\
&= \theta^\star - \lambda V_t^{-1}\theta^\star + V_t^{-1}S_t,
\end{aligned} \tag{9}$$

where we used $X_{1:t}^\top X_{1:t} = V_t - \lambda I_d$ and defined the vector-valued martingale $S_t := X_{1:t}^\top \eta_{1:t} = \sum_{i=1}^t \eta_i\, x_{s_i}$. Hence

$$\widehat{\theta}_t - \theta^\star = V_t^{-1} S_t - \lambda V_t^{-1} \theta^\star. \tag{10}$$

By the triangle inequality (ix),

$$\|\widehat{\theta}_t - \theta^\star\|_{V_t} \leq \|V_t^{-1} S_t\|_{V_t} + \|\lambda V_t^{-1}\theta^\star\|_{V_t}.$$

By norm duality (i), $\|V_t^{-1} S_t\|_{V_t} = \|S_t\|_{V_t^{-1}}$. Since each $\eta_t$ is conditionally $R$-sub-Gaussian and $s_t$ is $\mathcal{F}_{t-1}$-measurable, the self-normalized bound of (Abbasi-Yadkori et al., 2011, Theorem 1) gives: with probability at least $1 - \delta$, simultaneously for all $t \geq 0$,

$$\|S_t\|_{V_t^{-1}} \leq R\sqrt{2\log\Big(\frac{\det(V_t)^{1/2}\det(\lambda I_d)^{-1/2}}{\delta}\Big)}. \tag{11}$$

By the regularization bound (iv), $\|\lambda V_t^{-1}\theta^\star\|_{V_t}^2 = \lambda^2(\theta^\star)^\top V_t^{-1}\theta^\star \leq \lambda\, S_0^2$, so $\|\lambda V_t^{-1}\theta^\star\|_{V_t} \leq \sqrt{\lambda}\, S_0$. As a result, on the $1 - \delta$ event from Eq. (11) yields $\|\widehat{\theta}_t - \theta^\star\|_{V_t} \leq \beta_t(\delta)$ for all $t \geq 0$, establishing $\mathbb{P}(\mathcal{E}_\delta) \geq 1 - \delta$.

On $\mathcal{E}_\delta$, for any $s \in \mathcal{S}$, Cauchy–Schwarz (ii) in the $(V_t^{-1}, V_t)$ pair gives

$$|\widehat{\mu}_t(s) - \mu(s)| = |x_s^\top(\widehat{\theta}_t - \theta^\star)| \leq \|x_s\|_{V_t^{-1}}\|\widehat{\theta}_t - \theta^\star\|_{V_t} \leq \beta_t(\delta)\|x_s\|_{V_t^{-1}}.$$

The identical argument with $g(\pi_p)$ in place of $x_s$ yields the path-level bound.

For any $(\pi_p, \pi_{p'}) \in \Pi^2$,

$$\widehat{\Delta}_t(\pi_p, \pi_{p'}) - \Delta(\pi_p, \pi_{p'}) = g(\pi_p, \pi_{p'})^\top(\widehat{\theta}_t - \theta^\star).$$

Cauchy–Schwarz gives $|\widehat{\Delta}_t(\pi_p, \pi_{p'}) - \Delta(\pi_p, \pi_{p'})| \leq \|g(\pi_p, \pi_{p'})\|_{V_t^{-1}}\|\widehat{\theta}_t - \theta^\star\|_{V_t} \leq \beta_t(\delta)\sigma_t(\pi_p, \pi_{p'}) = W_t(\pi_p, \pi_{p'})$, which is Eq. (8). $\qquad\square$

The next result certifies correctness. It shows that whenever GICA halts under the gap-index stopping rule, the returned shortlist is $\epsilon$-optimal on the confidence event, so the sample-complexity bound proved later concerns a procedure that is guaranteed to output a valid top-$K$ set.

**Proposition A.2** ($\epsilon$-optimality of GICA). *For any $\delta \in (0, 1)$ and $\epsilon \geq 0$, on the event $\mathcal{E}_\delta$, the set $\widehat{P}_K(\tau_\delta)$ output by Algorithm 1 is $\epsilon$-optimal in the sense of Definition 3.3, i.e., $\widehat{P}_K(\tau_\delta) \subseteq P_K^{\star,\epsilon}$. In particular, $\mathbb{P}(\widehat{P}_K(\tau_\delta) \subseteq P_K^{\star,\epsilon}) \geq 1 - \delta$.*

*Proof.* We work on the confidence event $\mathcal{E}_\delta$. At round $\tau_\delta$, the stopping condition $\Gamma_{\tau_\delta} \geq -\epsilon$ holds, so by Eq. (4), for every $\pi_p \in \widehat{P}_K(\tau_\delta)$ and every $\pi_{p'} \notin \widehat{P}_K(\tau_\delta)$,

$$\widehat{\Delta}_{\tau_\delta}(\pi_p, \pi_{p'}) - W_{\tau_\delta}(\pi_p, \pi_{p'}) \geq -\epsilon.$$

On $\mathcal{E}_\delta$, the pairwise confidence bound Eq. (8) gives $\Delta(\pi_p, \pi_{p'}) \geq \widehat{\Delta}_{\tau_\delta}(\pi_p, \pi_{p'}) - W_{\tau_\delta}(\pi_p, \pi_{p'})$, hence

$$\mu(\pi_p) \geq \mu(\pi_{p'}) - \epsilon \qquad \text{for every } \pi_p \in \widehat{P}_K(\tau_\delta),\ \pi_{p'} \notin \widehat{P}_K(\tau_\delta). \tag{12}$$

Fix any $\pi_p \in \widehat{P}_K(\tau_\delta)$. We show $\mu(\pi_p) \geq \mu_K^\star - \epsilon$ by considering two cases.

*Case 1: $P_K^\star \setminus \widehat{P}_K(\tau_\delta) \neq \emptyset$.* Pick any $\pi_{p'} \in P_K^\star \setminus \widehat{P}_K(\tau_\delta)$. Since $\pi_{p'} \in P_K^\star$, we have $\mu(\pi_{p'}) \geq \mu_K^\star$ by definition of $\mu_K^\star$. Eq. (12) then gives $\mu(\pi_p) \geq \mu(\pi_{p'}) - \epsilon \geq \mu_K^\star - \epsilon$.

*Case 2: $P_K^\star \setminus \widehat{P}_K(\tau_\delta) = \emptyset$.* Since $|P_K^\star| = |\widehat{P}_K(\tau_\delta)| = K$, this implies $\widehat{P}_K(\tau_\delta) = P_K^\star$, hence $\mu(\pi_p) \geq \mu_K^\star \geq \mu_K^\star - \epsilon$ trivially.

In both cases, $\mu(\pi_p) \geq \mu_K^\star - \epsilon$, and therefore $\widehat{P}_K(\tau_\delta) \subseteq P_K^{\star,\epsilon}$, i.e., $\widehat{P}_K(\tau_\delta)$ is $\epsilon$-optimal in the sense of Definition 3.3. The probability bound follows from $\mathbb{P}(\mathcal{E}_\delta) \geq 1 - \delta$ established in Lemma A.1. $\qquad\square$

### A.2.1 Proof of Lemma 3.7: Exact Variance Contraction

This part proves the exact rank-one identity for the one-step drop in pairwise variance. The identity quantifies precisely how much querying a given step reduces the uncertainty along a boundary direction, and it is the algebraic engine reused in every contraction lemma that follows.

*Proof.* The update $V_{t+1} = V_t + x_{s_t} x_{s_t}^\top$ is a rank-one perturbation. By Sherman–Morrison (viii),

$$V_{t+1}^{-1} = V_t^{-1} - \frac{V_t^{-1} x_{s_t} x_{s_t}^\top V_t^{-1}}{1 + x_{s_t}^\top V_t^{-1} x_{s_t}}.$$

Let $h := g(\pi_p, \pi_{p'})$ for brevity. Then

$$\sigma_{t+1}^2(\pi_p, \pi_{p'}) = h^\top V_{t+1}^{-1} h = h^\top V_t^{-1} h - \frac{(h^\top V_t^{-1} x_{s_t})^2}{1 + \|x_{s_t}\|_{V_t^{-1}}^2}$$

$$= \sigma_t^2(\pi_p, \pi_{p'}) - \frac{\langle h, x_{s_t}\rangle_{V_t^{-1}}^2}{1 + \|x_{s_t}\|_{V_t^{-1}}^2},$$

where the numerator uses the fact that $h^\top V_t^{-1} x_{s_t} x_{s_t}^\top V_t^{-1} h = (h^\top V_t^{-1} x_{s_t})^2$ (scalar). Rearranging gives Eq. (2). $\square$

## A.3 Proof of Theorem 3.8: Sample Complexity of GICA

We prove Theorem 3.8 through a sequence of auxiliary lemmas, all stated on the confidence event $\mathcal{E}_\delta$, in four stages. First, we translate the gap-index stopping rule into a strictly positive variance lower bound for the sampled pair (Subsection A.3.1). Second, we invoke the pair–step correlation (Assumption 3.2) to turn this into a per-round multiplicative contraction of every pair's variance (Subsection A.3.2). Third, we show that the all-pairs half-width $\widetilde{M}_t = \max_{g(\pi_p, \pi_{p'}) \neq 0} \|g(\pi_p, \pi_{p'})\|_{V_t^{-1}}$, and hence the confidence deficit $2\beta_t(\delta)\widetilde{M}_t$, decays in the log-determinant, which floors the contraction rate $\kappa_t$ away from zero once the deficit crosses below $\Delta_{\mathcal{C}}/2$ at a closed-form deficit-crossing time (Subsections A.3.3 and A.3.4). Finally, we combine the resulting uniform exponential contraction with the stopping criterion to obtain the two-phase sample-complexity bound, split into a deficit-crossing phase and a contraction phase (Subsection A.3.5).

### A.3.1 Stopping Geometry: Ordering and the Gap Index

This part records the elementary consequences of the gap-index stopping rule that are used repeatedly below. We first show that the empirical shortlist respects the empirical ordering of path scores (Corollary A.3), then that failure to stop forces a small gap index for the minimizing pair (Lemma A.4) and hence for the sampled pair (Corollary A.5). Finally, Lemma A.6 converts a small gap index into a strictly positive lower bound on that pair's variance, expressed through its true gap. Together these turn the stopping criterion into the variance lower bound that drives the leverage and contraction arguments of the following subsections.

This corollary states the basic consistency of the shortlist with the estimated scores, namely that every shortlisted path has at least as high an estimate as every challenger. It underlies the sign control used throughout the gap-index and stopping arguments.

**Corollary A.3** (Empirical ordering). *For all $t \geq 0$, $\pi_p \in \widehat{P}_K(t)$, and $\pi_{p'} \notin \widehat{P}_K(t)$, $\widehat{\Delta}_t(\pi_p, \pi_{p'}) \geq 0$.*

*Proof.* Suppose by contradiction that there exist $\pi_p \in \widehat{P}_K(t)$ and $\pi_{p'} \notin \widehat{P}_K(t)$ such that $\widehat{\mu}_t(\pi_{p'}) > \widehat{\mu}_t(\pi_p)$. Let $P' := (\widehat{P}_K(t) \setminus \{\pi_p\}) \cup \{\pi_{p'}\}$. Then $|P'| = K$ and

$$\sum_{\rho \in P'} \widehat{\mu}_t(\rho) = \sum_{\rho \in \widehat{P}_K(t)} \widehat{\mu}_t(\rho) - \widehat{\mu}_t(\pi_p) + \widehat{\mu}_t(\pi_{p'}) > \sum_{\rho \in \widehat{P}_K(t)} \widehat{\mu}_t(\rho),$$

contradicting the maximality of $\widehat{P}_K(t)$. Therefore $\widehat{\mu}_t(\pi_p) \geq \widehat{\mu}_t(\pi_{p'})$ for all $\pi_p \in \widehat{P}_K(t)$ and $\pi_{p'} \notin \widehat{P}_K(t)$, which implies $\widehat{\Delta}_t(\pi_p, \pi_{p'}) \geq 0$. □

The next lemma converts the failure of the stopping rule at a round into a quantitative statement about the minimizing boundary pair, showing its gap index must lie strictly below the squared confidence radius. This is the bridge from the stopping criterion to the variance lower bounds used later.

**Lemma A.4** (Failure of stopping implies a small gap index). *For any $t \geq 0$ with $\Gamma_t < -\epsilon$, and let $(\pi_t^+, \pi_t^-)$ attain the minimum in Eq. (4). Then $\sigma_t(\pi_t^+, \pi_t^-) > 0$ and $G_t(\pi_t^+, \pi_t^-) < \beta_t(\delta)^2$.*

*Proof.* From $\Gamma_t < -\epsilon$ and the definition of $(\pi_t^+, \pi_t^-)$,

$$\widehat{\Delta}_t(\pi_t^+, \pi_t^-) < W_t(\pi_t^+, \pi_t^-) - \epsilon \leq W_t(\pi_t^+, \pi_t^-). \tag{13}$$

By Lemma A.3, $\widehat{\Delta}_t(\pi_t^+, \pi_t^-) \geq 0$. If $\sigma_t(\pi_t^+, \pi_t^-) = 0$ then $W_t(\pi_t^+, \pi_t^-) = 0$, contradicting Eq. (13). Hence $\sigma_t(\pi_t^+, \pi_t^-) > 0$. Squaring both sides of Eq. (13) (both sides are non-negative) and dividing by $\sigma_t^2(\pi_t^+, \pi_t^-) > 0$ gives

$$G_t(\pi_t^+, \pi_t^-) = \frac{\widehat{\Delta}_t(\pi_t^+, \pi_t^-)^2}{\sigma_t^2(\pi_t^+, \pi_t^-)} < \frac{W_t(\pi_t^+, \pi_t^-)^2}{\sigma_t^2(\pi_t^+, \pi_t^-)} = \beta_t(\delta)^2. \qquad □$$

This corollary specializes the previous bound to the pair GICA actually samples at each pre-termination round, confirming that the queried boundary pair always carries a small gap index. It is the form invoked directly in the leverage analysis.

**Corollary A.5** (Small gap index for the sampling pair). *For every round $t$ with $1 \leq t \leq \tau_\delta$, $G_{t-1}(\pi_t^\star, \pi_t^\dagger) < \beta_{t-1}(\delta)^2$.*

*Proof.* Since $t \leq \tau_\delta$, the algorithm has not stopped at $t-1$, so $\Gamma_{t-1} < -\epsilon$. Lemma A.4 at round $t-1$ yields $\min_{\pi_p \in \widehat{P}_K(t-1)} \min_{\pi_{p'} \notin \widehat{P}_K(t-1)} G_{t-1}(\pi_p, \pi_{p'}) < \beta_{t-1}(\delta)^2$. Algorithm 1 selects $(\pi_t^\star, \pi_t^\dagger)$ to minimise $G_{t-1}$ over this same set, so $G_{t-1}(\pi_t^\star, \pi_t^\dagger)$ is at most this minimum. □

The following lemma translates a small gap index into a strictly positive lower bound on the pairwise variance in terms of the true gap. This lower bound prevents the queried direction's uncertainty from collapsing prematurely and supplies the floor used in the leverage and contraction steps.

**Lemma A.6** (Variance lower bound from a small gap index). *For any distinct pair $(\pi_p, \pi_{p'})$ and any $t \geq 0$, if $G_t(\pi_p, \pi_{p'}) < \beta_t(\delta)^2$ then $\sigma_t(\pi_p, \pi_{p'}) > |\Delta(\pi_p, \pi_{p'})|/(2\beta_t(\delta))$.*

*Proof.* The hypothesis gives $|\widehat{\Delta}_t(\pi_p, \pi_{p'})| < \beta_t(\delta)\sigma_t(\pi_p, \pi_{p'})$. By the triangle inequality (ix) and Eq. (8)

$$|\Delta(\pi_p, \pi_{p'})| \leq |\widehat{\Delta}_t(\pi_p, \pi_{p'})| + W_t(\pi_p, \pi_{p'}) < \beta_t(\delta)\sigma_t(\pi_p, \pi_{p'}) + \beta_t(\delta)\sigma_t(\pi_p, \pi_{p'}) = 2\beta_t(\delta)\sigma_t(\pi_p, \pi_{p'}). \qquad □$$

### A.3.2 Cross-Direction Multiplicative Contraction

This part uses the pair–step correlation of Assumption 3.2 to show that every pair's variance contracts by a strictly positive multiplicative factor at every round before termination. We first establish a strictly positive lower bound on the leverage of the queried step that holds along the entire trajectory (Lemma A.7), then combine it with Assumption 3.2 to obtain the per-round multiplicative contraction shared by all pairs (Lemma A.8), and finally track how the contraction rate evolves and tightens at the stopping time (Lemma A.9).

This lemma lower-bounds the leverage of the step GICA queries, showing it cannot vanish at any round before termination. The bound is expressed through the floored boundary gap and is what allows the abstract correlation of Assumption 3.2 to deliver a usable per-round contraction.

**Lemma A.7** (Algorithm-induced leverage lower bound). *On the event $\mathcal{E}_\delta$, for every round $t$ with $1 \le t \le \tau_\delta$, the queried step $s_t$ satisfies*

$$\frac{\|x_{s_t}\|_{V_{t-1}^{-1}}^2}{1 + \|x_{s_t}\|_{V_{t-1}^{-1}}^2} \ge c_0\, \sigma_{t-1}^2(\pi_t^\star, \pi_t^\dagger) \ge \frac{c_0\, \underline{\Delta}_t^2}{4\, \beta_{t-1}(\delta)^2}, \tag{14}$$

*where $(z)_+ := \max(z, 0)$, $c_0 = \lambda/(4(\lambda + L^2))$, $\Delta_{\min}^\Pi = \min_{(\pi_p, \pi_{p'}) \in \Pi^2,\, p \ne p'} |\Delta(\pi_p, \pi_{p'})|$ is the global minimum gap, the floored boundary gap is $\underline{\Delta}_t := \max\{(\Delta_{\mathcal{C}} - 2\beta_{t-1}(\delta)\widetilde{M}_{t-1})_+, \Delta_{\min}^\Pi\}$, and $\widetilde{M}_t = \max_{g(\pi_p, \pi_{p'}) \ne 0} \|g(\pi_p, \pi_{p'})\|_{V_t^{-1}}$.*

*Proof.* Fix $t$ with $1 \le t \le \tau_\delta$ and write $h_2 := g(\pi_t^\star, \pi_t^\dagger)$.

**Step 1: first inequality in Eq. (14).** The selection rule Eq. (3) chooses $s_t \in \arg\max_{s \in \mathcal{U}_t} \mathcal{C}_{t-1}(s; \pi_t^\star, \pi_t^\dagger)$, so by Lemma 3.7 applied to the boundary pair,

$$\frac{\langle h_2, x_{s_t} \rangle_{V_{t-1}^{-1}}^2}{1 + \|x_{s_t}\|_{V_{t-1}^{-1}}^2} = \max_{s \in \mathcal{U}_t} \frac{\langle h_2, x_s \rangle_{V_{t-1}^{-1}}^2}{1 + \|x_s\|_{V_{t-1}^{-1}}^2}. \tag{15}$$

Since $\mathcal{U}_t = \pi_t^\star \cup \pi_t^\dagger$ and $g(\pi_p)$ averages step features along $\pi_p$, there exist coefficients $\{\alpha_s\}_{s \in \mathcal{U}_t}$ with $h_2 = \sum_{s \in \mathcal{U}_t} \alpha_s x_s$ and $\sum_{s \in \mathcal{U}_t} |\alpha_s| \le 2$. By bilinearity of $\langle \cdot, \cdot \rangle_{V_{t-1}^{-1}}$ (xiii),

$$\sigma_{t-1}^2(\pi_t^\star, \pi_t^\dagger) = \langle h_2, h_2 \rangle_{V_{t-1}^{-1}} = \sum_{s \in \mathcal{U}_t} \alpha_s \langle h_2, x_s \rangle_{V_{t-1}^{-1}} \le 2 \max_{s \in \mathcal{U}_t} \left| \langle h_2, x_s \rangle_{V_{t-1}^{-1}} \right|,$$

which, upon squaring, gives $\max_{s \in \mathcal{U}_t} \langle h_2, x_s \rangle_{V_{t-1}^{-1}}^2 \ge \frac{1}{4}\sigma_{t-1}^4(\pi_t^\star, \pi_t^\dagger)$. Using the leverage bound (vi), $\|x_s\|_{V_{t-1}^{-1}}^2 \le L^2/\lambda$, in the denominator of Eq. (15) and $\lambda/(\lambda + L^2) = 4c_0$,

$$\frac{\langle h_2, x_{s_t} \rangle_{V_{t-1}^{-1}}^2}{1 + \|x_{s_t}\|_{V_{t-1}^{-1}}^2} \ge \frac{\lambda}{\lambda + L^2} \max_{s \in \mathcal{U}_t} \langle h_2, x_s \rangle_{V_{t-1}^{-1}}^2 \ge c_0\, \sigma_{t-1}^4(\pi_t^\star, \pi_t^\dagger).$$

By Cauchy–Schwarz (ii), $\langle h_2, x_{s_t} \rangle_{V_{t-1}^{-1}}^2 \le \sigma_{t-1}^2(\pi_t^\star, \pi_t^\dagger)\|x_{s_t}\|_{V_{t-1}^{-1}}^2$, so substituting and dividing by $\sigma_{t-1}^2(\pi_t^\star, \pi_t^\dagger) > 0$ (positive by Corollary A.5 and Lemma A.6) yields the first inequality of Eq. (14).

**Step 2: a signed gap bound $|\Delta(\pi_t^\star, \pi_t^\dagger)| \ge \Delta_{\mathcal{C}} - 2\beta_{t-1}(\delta)\widetilde{M}_{t-1}$.** We first record two consequences of the definition $\Delta_{\mathcal{C}} = \min_{(\pi_p, \pi_{p'}) \in \mathcal{C}_K^\star} \Delta(\pi_p, \pi_{p'})$. Since $\mu_K^\star = \min_{\pi_p \in P_K^\star} \mu(\pi_p) \ge \max_{\pi_{p'} \notin P_K^\star} \mu(\pi_{p'})$, the boundary minimum is attained by the worst top-$K$ path against the best non-top-$K$ path, so $\Delta_{\mathcal{C}} = \mu_K^\star - \max_{\pi_{p'} \notin P_K^\star} \mu(\pi_{p'})$. Hence

$$\mu(\pi_p) \ge \mu_K^\star \text{ for } \pi_p \in P_K^\star, \qquad \mu(\pi_{p'}) \le \mu_K^\star - \Delta_{\mathcal{C}} \text{ for } \pi_{p'} \notin P_K^\star. \tag{16}$$

By Algorithm 1, $\pi_t^\star \in \widehat{P}_K(t)$ and $\pi_t^\dagger \notin \widehat{P}_K(t)$, so each is either correctly or incorrectly classified. We label the four configurations by $\pi_a \in P_K^\star \setminus \widehat{P}_K(t)$, $\pi_b \in \widehat{P}_K(t) \setminus P_K^\star$, $\pi_c \in P_K^\star \cap \widehat{P}_K(t)$, and $\pi_d \notin P_K^\star \cup \widehat{P}_K(t)$. Whenever the shortlist is imperfect, $|\widehat{P}_K(t)| = |P_K^\star| = K$ forces both witnesses $\pi_a, \pi_b$ to exist. Throughout we use the empirical ordering (Corollary A.3), $\widehat{\Delta}_{t-1}(\pi_p, \pi_{p'}) \ge 0$ for $\pi_p \in \widehat{P}_K(t)$, $\pi_{p'} \notin \widehat{P}_K(t)$, and the pairwise confidence bound Eq. (8), $|\widehat{\Delta}_{t-1}(\pi_p, \pi_{p'}) - \Delta(\pi_p, \pi_{p'})| \le \beta_{t-1}(\delta)\|g(\pi_p, \pi_{p'})\|_{V_{t-1}^{-1}}$. For any distinct pair, $\|g(\pi_p, \pi_{p'})\|_{V_{t-1}^{-1}} \le \widetilde{M}_{t-1}$ by the definition of $\widetilde{M}_{t-1}$ as the all-pairs maximum.

**Case 1** $(\pi_t^\star = \pi_c,\ \pi_t^\dagger = \pi_d)$. Then $(\pi_t^\star, \pi_t^\dagger) \in \mathcal{C}_K^\star$, so $|\Delta(\pi_t^\star, \pi_t^\dagger)| \ge \Delta_{\mathcal{C}}$.

**Case 2** ($\pi_t^\star = \pi_b$, $\pi_t^\dagger = \pi_a$). By Eq. (16), $\mu(\pi_t^\star) \leq \mu_K^\star - \Delta_{\mathcal{C}}$ and $\mu(\pi_t^\dagger) \geq \mu_K^\star$, hence $\Delta(\pi_t^\dagger, \pi_t^\star) = \mu(\pi_t^\dagger) - \mu(\pi_t^\star) \geq \Delta_{\mathcal{C}}$, so $|\Delta(\pi_t^\star, \pi_t^\dagger)| \geq \Delta_{\mathcal{C}}$.

**Case 3** ($\pi_t^\star = \pi_b$, $\pi_t^\dagger = \pi_d$). A witness $\pi_a$ exists. Since $\pi_t^\star \in \widehat{P}_K(t)$ and $\pi_a \notin \widehat{P}_K(t)$, empirical ordering gives $\widehat{\Delta}_{t-1}(\pi_t^\star, \pi_a) \geq 0$. Therefore, on $\mathcal{E}_\delta$, Eq. (8) yields

$$\mu(\pi_t^\star) - \mu(\pi_a) = \widehat{\Delta}_{t-1}(\pi_t^\star, \pi_a) - \big(\widehat{\Delta}_{t-1}(\pi_t^\star, \pi_a) - \Delta(\pi_t^\star, \pi_a)\big) \geq -\beta_{t-1}(\delta)\|g(\pi_t^\star, \pi_a)\|_{V_{t-1}^{-1}} \geq -\beta_{t-1}(\delta)\widetilde{M}_{t-1},$$

so with $\mu(\pi_a) \geq \mu_K^\star$, $\mu(\pi_t^\star) \geq \mu_K^\star - \beta_{t-1}(\delta)\widetilde{M}_{t-1}$. Since $\pi_t^\dagger \notin P_K^\star$, $\mu(\pi_t^\dagger) \leq \mu_K^\star - \Delta_{\mathcal{C}}$ by Eq. (16), whence

$$\Delta(\pi_t^\star, \pi_t^\dagger) \geq \Delta_{\mathcal{C}} - \beta_{t-1}(\delta)\widetilde{M}_{t-1} \geq \Delta_{\mathcal{C}} - 2\beta_{t-1}(\delta)\widetilde{M}_{t-1},$$

so $|\Delta(\pi_t^\star, \pi_t^\dagger)| \geq \Delta_{\mathcal{C}} - 2\beta_{t-1}(\delta)\widetilde{M}_{t-1}$.

**Case 4** ($\pi_t^\star = \pi_c$, $\pi_t^\dagger = \pi_a$). A witness $\pi_b$ exists. Since $\pi_t^\dagger \notin \widehat{P}_K(t)$ and $\pi_b \in \widehat{P}_K(t)$, empirical ordering gives $\widehat{\Delta}_{t-1}(\pi_b, \pi_t^\dagger) \geq 0$, hence on $\mathcal{E}_\delta$, by Eq. (8), $\|g(\pi_b, \pi_t^\dagger)\|_{V_{t-1}^{-1}} \leq \widetilde{M}_{t-1}$, and $\mu(\pi_b) \leq \mu_K^\star - \Delta_{\mathcal{C}}$,

$$\mu(\pi_t^\dagger) \leq \mu(\pi_b) + \beta_{t-1}(\delta)\|g(\pi_b, \pi_t^\dagger)\|_{V_{t-1}^{-1}} \leq \mu_K^\star - \Delta_{\mathcal{C}} + \beta_{t-1}(\delta)\widetilde{M}_{t-1}. \tag{17}$$

Since $\pi_t^\star \in \widehat{P}_K(t)$ and $\pi_t^\dagger = \pi_a \notin \widehat{P}_K(t)$, empirical ordering gives $\widehat{\Delta}_{t-1}(\pi_t^\star, \pi_a) \geq 0$, so as in Case 3,

$$\mu(\pi_t^\star) \geq \mu(\pi_a) - \beta_{t-1}(\delta)\|g(\pi_t^\star, \pi_a)\|_{V_{t-1}^{-1}} \geq \mu_K^\star - \beta_{t-1}(\delta)\widetilde{M}_{t-1}. \tag{18}$$

Subtracting Eq. (17) from Eq. (18),

$$\Delta(\pi_t^\star, \pi_t^\dagger) = \mu(\pi_t^\star) - \mu(\pi_t^\dagger) \geq \Delta_{\mathcal{C}} - 2\beta_{t-1}(\delta)\widetilde{M}_{t-1},$$

so $|\Delta(\pi_t^\star, \pi_t^\dagger)| \geq \Delta_{\mathcal{C}} - 2\beta_{t-1}(\delta)\widetilde{M}_{t-1}$.

In Cases 1 and 2, $2\beta_{t-1}(\delta)\widetilde{M}_{t-1} \geq 0$ gives $|\Delta(\pi_t^\star, \pi_t^\dagger)| \geq \Delta_{\mathcal{C}} \geq \Delta_{\mathcal{C}} - 2\beta_{t-1}(\delta)\widetilde{M}_{t-1}$. Thus in all four cases,

$$|\Delta(\pi_t^\star, \pi_t^\dagger)| \geq \Delta_{\mathcal{C}} - 2\beta_{t-1}(\delta)\widetilde{M}_{t-1}. \tag{19}$$

**Step 3: second inequality in Eq.** (14). The right-hand side of Eq. (19) may be negative, whereas $|\Delta(\pi_t^\star, \pi_t^\dagger)| \geq 0$ always, so $|\Delta(\pi_t^\star, \pi_t^\dagger)| \geq \big(\Delta_{\mathcal{C}} - 2\beta_{t-1}(\delta)\widetilde{M}_{t-1}\big)_+$. Moreover, $\pi_t^\star \in \widehat{P}_K(t)$ and $\pi_t^\dagger \notin \widehat{P}_K(t)$ are distinct paths, so $|\Delta(\pi_t^\star, \pi_t^\dagger)| \geq \Delta_{\min}^\Pi$ by the definition of $\Delta_{\min}^\Pi$. Taking the larger of the two lower bounds gives $|\Delta(\pi_t^\star, \pi_t^\dagger)| \geq \underline{\Delta}_t$, and since both sides are non-negative, squaring yields $|\Delta(\pi_t^\star, \pi_t^\dagger)|^2 \geq \underline{\Delta}_t^2$. By Corollary A.5, $G_{t-1}(\pi_t^\star, \pi_t^\dagger) < \beta_{t-1}(\delta)^2$, so Lemma A.6 gives $\sigma_{t-1}(\pi_t^\star, \pi_t^\dagger) > |\Delta(\pi_t^\star, \pi_t^\dagger)|/(2\beta_{t-1}(\delta))$, i.e.

$$\sigma_{t-1}^2(\pi_t^\star, \pi_t^\dagger) > \frac{|\Delta(\pi_t^\star, \pi_t^\dagger)|^2}{4\beta_{t-1}(\delta)^2} \geq \frac{\underline{\Delta}_t^2}{4\beta_{t-1}(\delta)^2}.$$

Substituting this lower bound into the first inequality of Eq. (14) established in Step 1 yields the second inequality of Eq. (14). □

The following lemma is the central per-round contraction result. It shows that querying any step shrinks the variance of every distinct pair by a common multiplicative factor $1 - \kappa_t$, so that information from a single step propagates across all correlated paths at once.

**Lemma A.8** (Cross-direction multiplicative contraction). *Suppose Assumptions 3.1 and 3.2 hold. Then on the event $\mathcal{E}_\delta$, for every round $t$ with $1 \leq t \leq \tau_\delta$ and every ordered pair of distinct paths $(\pi_p, \pi_{p'}) \in \Pi^2$ with $g(\pi_p, \pi_{p'}) \neq 0$,*

$$\sigma_t^2(\pi_p, \pi_{p'}) \leq \sigma_{t-1}^2(\pi_p, \pi_{p'})\big(1 - \kappa_t\big), \tag{20}$$

*where* $\kappa_t := \dfrac{\rho^\dagger c_0 \big(\Delta_{\mathcal{C}} - 2\beta_{t-1}(\delta)\,\widetilde{M}_{t-1}\big)_+^2}{4\beta_{t-1}(\delta)^2} \in [0, 1)$.

*Proof.* Fix $t$ with $1 \leq t \leq \tau_\delta$ and a pair $(\pi_p, \pi_{p'})$ with $g(\pi_p, \pi_{p'}) \neq 0$. By Lemma 3.7 applied to the fixed pair,

$$\sigma_{t-1}^2(\pi_p, \pi_{p'}) - \sigma_t^2(\pi_p, \pi_{p'}) = \frac{\langle g(\pi_p, \pi_{p'}), x_{s_t} \rangle_{V_{t-1}^{-1}}^2}{1 + \|x_{s_t}\|_{V_{t-1}^{-1}}^2}. \tag{21}$$

Apply Assumption 3.2 with the choice $A = V_{t-1}$ (which satisfies $V_{t-1} \succeq \lambda I_d$ by construction of the ridge-regularized design matrix):

$$\langle g(\pi_p, \pi_{p'}), x_{s_t} \rangle_{V_{t-1}^{-1}}^2 \geq \rho^\dagger \|g(\pi_p, \pi_{p'})\|_{V_{t-1}^{-1}}^2 \|x_{s_t}\|_{V_{t-1}^{-1}}^2.$$

By the definition of $\sigma_{t-1}^2(\pi_p, \pi_{p'})$, $\|g(\pi_p, \pi_{p'})\|_{V_{t-1}^{-1}}^2 = \sigma_{t-1}^2(\pi_p, \pi_{p'})$, hence

$$\langle g(\pi_p, \pi_{p'}), x_{s_t} \rangle_{V_{t-1}^{-1}}^2 \geq \rho^\dagger \sigma_{t-1}^2(\pi_p, \pi_{p'}) \|x_{s_t}\|_{V_{t-1}^{-1}}^2. \tag{22}$$

The case $x_{s_t} = 0$ is excluded because Assumption 3.2 requires $x_s \neq 0$; equivalently, $\|x_{s_t}\|_{V_{t-1}^{-1}}^2 = 0$ would by Lemma A.7 contradict $\sigma_{t-1}(\pi_t^\star, \pi_t^\dagger) > 0$, so the inequality is nontrivial on $\mathcal{E}_\delta$ at $t \leq \tau_\delta$. Substituting Eq. (22) into Eq. (21),

$$\sigma_{t-1}^2(\pi_p, \pi_{p'}) - \sigma_t^2(\pi_p, \pi_{p'}) \geq \rho^\dagger \sigma_{t-1}^2(\pi_p, \pi_{p'}) \frac{\|x_{s_t}\|_{V_{t-1}^{-1}}^2}{1 + \|x_{s_t}\|_{V_{t-1}^{-1}}^2}.$$

By Lemma A.7, $\dfrac{\|x_{s_t}\|_{V_{t-1}^{-1}}^2}{1 + \|x_{s_t}\|_{V_{t-1}^{-1}}^2} \geq c_0 \sigma_{t-1}^2(\pi_t^\star, \pi_t^\dagger) \geq \dfrac{c_0 \underline{\Delta}_t^2}{4 \beta_{t-1}(\delta)^2} \geq \dfrac{c_0 \left(\Delta_{\mathcal{C}} - 2\beta_{t-1}(\delta) \widetilde{M}_{t-1}\right)_+^2}{4 \beta_{t-1}(\delta)^2}$, where the last inequality uses $\underline{\Delta}_t \geq \left(\Delta_{\mathcal{C}} - 2\beta_{t-1}(\delta)\widetilde{M}_{t-1}\right)_+$. Combining,

$$\sigma_{t-1}^2(\pi_p, \pi_{p'}) - \sigma_t^2(\pi_p, \pi_{p'}) \geq \sigma_{t-1}^2(\pi_p, \pi_{p'}) \underbrace{\frac{\rho^\dagger c_0 \left(\Delta_{\mathcal{C}} - 2\beta_{t-1}(\delta) \widetilde{M}_{t-1}\right)_+^2}{4 \beta_{t-1}(\delta)^2}}_{= \kappa_t}.$$

Rearranging gives Eq. (20). By the second inequality of Lemma A.7 together with $\underline{\Delta}_t \geq \left(\Delta_{\mathcal{C}} - 2\beta_{t-1}(\delta)\widetilde{M}_{t-1}\right)_+$,

$$\frac{c_0 \left(\Delta_{\mathcal{C}} - 2\beta_{t-1}(\delta) \widetilde{M}_{t-1}\right)_+^2}{4 \beta_{t-1}(\delta)^2} \leq \frac{c_0 \underline{\Delta}_t^2}{4 \beta_{t-1}(\delta)^2} \leq c_0 \sigma_{t-1}^2(\pi_t^\star, \pi_t^\dagger),$$

so multiplying by $\rho^\dagger > 0$ and recalling the definition of $\kappa_t$ gives

$$\kappa_t = \rho^\dagger \frac{c_0 \left(\Delta_{\mathcal{C}} - 2\beta_{t-1}(\delta) \widetilde{M}_{t-1}\right)_+^2}{4 \beta_{t-1}(\delta)^2} \leq \rho^\dagger c_0 \sigma_{t-1}^2(\pi_t^\star, \pi_t^\dagger).$$

By the feature-norm bound (v), $\|g(\pi_t^\star, \pi_t^\dagger)\|_2 \leq 2L$, and the regularization bound (iv) then yields $\sigma_{t-1}^2(\pi_t^\star, \pi_t^\dagger) = \|g(\pi_t^\star, \pi_t^\dagger)\|_{V_{t-1}^{-1}}^2 \leq \|g(\pi_t^\star, \pi_t^\dagger)\|_2^2 / \lambda \leq 4L^2/\lambda$. Substituting $c_0 = \lambda/(4(\lambda + L^2))$,

$$c_0 \sigma_{t-1}^2(\pi_t^\star, \pi_t^\dagger) \leq \frac{\lambda}{4(\lambda + L^2)} \cdot \frac{4L^2}{\lambda} = \frac{L^2}{\lambda + L^2} < 1.$$

Combining the last two displays with $\rho^\dagger \in (0, 1]$,

$$0 \leq \kappa_t \leq \rho^\dagger \frac{L^2}{\lambda + L^2} \leq \frac{L^2}{\lambda + L^2} < 1,$$

where non-negativity of *kappa_t* is immediate, since $\rho^\dagger, c_0, \beta_{t-1}(\delta)^2 > 0$ and the positive part is non-negative. Hence $\kappa_t \in [0, 1)$, so the contraction factor satisfies $1 - \kappa_t \in (0, 1]$. $\square$

This lemma tracks the contraction rate itself. It bounds the rate by a strictly decreasing envelope, shows the envelope rate is attained whenever the shortlist is already correct, and proves that under separability the terminal round attains the smallest envelope rate, which pins down the rate used in the final bound.

**Lemma A.9** (Monotone decay and stopping-time tightness of the contraction rate). *Suppose Assumptions 3.1, 3.2, and 3.5 hold. Then, on the event $\mathcal{E}_\delta$, the following hold for every round $t$ with $1 \leq t \leq \tau_\delta$.*

(i) (*Strictly decreasing envelope.*) *The rate is dominated by* $\bar{\kappa}_t = \dfrac{\rho^\dagger c_0 \Delta_\mathcal{C}^2}{4\beta_{t-1}(\delta)^2}$,

$$0 \ \leq \ \kappa_t \ \leq \ \bar{\kappa}_t, \tag{23}$$

*and the dominating sequence is non-increasing:* $\bar{\kappa}_{t+1} \leq \bar{\kappa}_t$ *for every* $1 \leq t < \tau_\delta$.

(ii) (*Envelope rate on a correct shortlist.*) *If* $\widehat{P}_K(t) = P_K^\star$, *the* $\widetilde{M}_{t-1}$-*correction is inactive and every distinct pair contracts at the envelope rate,*

$$\sigma_t^2(\pi_p, \pi_{p'}) \ \leq \ \sigma_{t-1}^2(\pi_p, \pi_{p'})\,(1 - \bar{\kappa}_t). \tag{24}$$

(iii) (*Attainment at the stopping time.*) *Under Assumption 3.5,* $\widehat{P}_K(\tau_\delta) = P_K^\star$, *so the terminal round contracts at the envelope rate and*

$$\sigma_{\tau_\delta}^2(\pi_p, \pi_{p'}) \ \leq \ \sigma_{\tau_\delta-1}^2(\pi_p, \pi_{p'})\,(1 - \bar{\kappa}_{\tau_\delta}), \qquad \bar{\kappa}_{\tau_\delta} = \min_{1 \leq s \leq \tau_\delta} \bar{\kappa}_s. \tag{25}$$

*Proof.* Fix $t$ with $1 \leq t \leq \tau_\delta$:

*Part (i).* Non-negativity is immediate, as $\rho^\dagger, c_0, \beta_{t-1}(\delta)^2 > 0$ and the positive part is non-negative. Since $\widetilde{M}_{t-1} \geq 0$ and $\beta_{t-1}(\delta) > 0$, we have $\Delta_\mathcal{C} - 2\beta_{t-1}(\delta)\widetilde{M}_{t-1} \leq \Delta_\mathcal{C}$; and because $\Delta_\mathcal{C} > 0$, the monotone positive-part map gives $\big(\Delta_\mathcal{C} - 2\beta_{t-1}(\delta)\widetilde{M}_{t-1}\big)_+ \leq \Delta_\mathcal{C}$. Squaring and dividing by $4\beta_{t-1}(\delta)^2 > 0$ yields $\kappa_t \leq \bar{\kappa}_t$, which is Eq. (23).

For strict monotonicity of $\bar{\kappa}_t$, the first inequality of Lemma A.7 gives $\|x_{s_t}\|_{V_{t-1}^{-1}}^2/(1 + \|x_{s_t}\|_{V_{t-1}^{-1}}^2) \geq c_0\,\sigma_{t-1}^2(\pi_t^\star, \pi_t^\dagger) > 0$, where positivity holds on $\mathcal{E}_\delta$ by Corollary A.5 and Lemma A.6, consequently $\|x_{s_t}\|_{V_{t-1}^{-1}}^2 > 0$. The rank-one determinant identity (xiv) then gives $\det V_t = \det V_{t-1}\,(1 + \|x_{s_t}\|_{V_{t-1}^{-1}}^2) > \det V_{t-1}$. As $\beta_{t-1}(\delta)$ depends on $t$ only through $\det V_{t-1}$ and is strictly increasing in it (Eq. (6)), $\beta_t(\delta) > \beta_{t-1}(\delta)$, so $\bar{\kappa}_{t+1} < \bar{\kappa}_t$ for every $1 \leq t < \tau_\delta$.

*Part (ii).* Suppose $\widehat{P}_K(t) = P_K^\star$. Then $\pi_t^\star \in \widehat{P}_K(t) = P_K^\star$ and $\pi_t^\dagger \notin \widehat{P}_K(t)$ give $\pi_t^\dagger \notin P_K^\star$, so $(\pi_t^\star, \pi_t^\dagger) \in \mathcal{C}_K^\star$ is a true boundary pair, Case 1 in the proof of Lemma A.7, in which no misclassified witness exists. The correction $2\beta_{t-1}(\delta)\widetilde{M}_{t-1}$ enters the signed-gap bound Eq. (19) only through the witnesses $\pi_a, \pi_b$ of Cases 3–4, so it is absent here and Case 1 yields directly $|\Delta(\pi_t^\star, \pi_t^\dagger)| \geq \Delta_\mathcal{C}$. Carrying this through Step 3 of Lemma A.7 gives $\sigma_{t-1}^2(\pi_t^\star, \pi_t^\dagger) > \Delta_\mathcal{C}^2/(4\beta_{t-1}(\delta)^2)$, and the first inequality of Lemma A.7 then gives

$$\frac{\|x_{s_t}\|_{V_{t-1}^{-1}}^2}{1 + \|x_{s_t}\|_{V_{t-1}^{-1}}^2} \ \geq \ c_0\,\sigma_{t-1}^2(\pi_t^\star, \pi_t^\dagger) \ > \ \frac{c_0\,\Delta_\mathcal{C}^2}{4\beta_{t-1}(\delta)^2}.$$

Substituting into the contraction step of Lemma A.8, for every distinct pair $(\pi_p, \pi_{p'})$,

$$\sigma_{t-1}^2(\pi_p, \pi_{p'}) - \sigma_t^2(\pi_p, \pi_{p'}) \ \geq \ \rho^\dagger \sigma_{t-1}^2(\pi_p, \pi_{p'}) \frac{\|x_{s_t}\|_{V_{t-1}^{-1}}^2}{1 + \|x_{s_t}\|_{V_{t-1}^{-1}}^2} \ > \ \sigma_{t-1}^2(\pi_p, \pi_{p'})\,\bar{\kappa}_t,$$

which is Eq. (24).

*Part (iii).* By Proposition A.2, on $\mathcal{E}_\delta$ the output satisfies $\widehat{P}_K(\tau_\delta) \subseteq P_K^{\star,\epsilon} = \{\pi_p \in \Pi : \mu(\pi_p) \geq \mu_K^\star - \epsilon\}$. By Eq. (16) in the proof of Lemma A.7, $\Delta_\mathcal{C} = \mu_K^\star - \max_{\pi_{p'} \notin P_K^\star} \mu(\pi_{p'})$, so every $\pi_p \notin P_K^\star$ obeys $\mu(\pi_p) \leq$

$\mu_K^\star - \Delta_{\mathcal{C}} < \mu_K^\star - \epsilon$ by Assumption 3.5. Thus $P_K^{\star,\epsilon} = P_K^\star$, and with $|\widehat{P}_K(\tau_\delta)| = |P_K^\star| = K$ the inclusion is an equality, $\widehat{P}_K(\tau_\delta) = P_K^\star$. Part (ii) at $t = \tau_\delta$ then gives the terminal contraction at the envelope rate, and since $\bar{\kappa}_s = \rho^\dagger c_0 \Delta_{\mathcal{C}}^2/(4\beta_{s-1}(\delta)^2)$ is strictly decreasing in $s$ by Part (i), $\bar{\kappa}_{\tau_\delta} = \min_{1 \le s \le \tau_\delta} \bar{\kappa}_s$, establishing Eq. (25). $\square$

### A.3.3 Unconditional Deficit Decay and the Confidence–Deficit Product

The contraction rate $\kappa_t$ is governed by the positive part $\big(\Delta_{\mathcal{C}} - 2\beta_{t-1}(\delta)\widetilde{M}_{t-1}\big)_+$, which vanishes whenever the confidence deficit $2\beta_{t-1}(\delta)\widetilde{M}_{t-1}$ exceeds the boundary gap $\Delta_{\mathcal{C}}$. The following lemma shows, under Assumptions 3.1, 3.2, and 3.5, that the all-pairs half-width $\widetilde{M}_t$ decays in the log-determinant $D_t$, so the confidence deficit drops below any fixed fraction of $\Delta_{\mathcal{C}}$ after a deterministic, closed-form number of rounds. In particular it falls below $\Delta_{\mathcal{C}}$ and, at the slightly later deficit-crossing time level used afterward, below $\Delta_{\mathcal{C}}/2$. The decay follows directly from the cross-direction correlation of Assumption 3.2 applied to the all-pairs maximizer, the Sherman–Morrison identity (viii), and the rank-one determinant identity (xiv). We then use this unconditional decay to establish monotonicity of the product $\beta_t(\delta)\widetilde{M}_t$ past the crossing time. Throughout this subsection we adopt the abbreviations $D_t$ and $\ell_t$, where the identity for $\ell_t$ follows from the rank-one determinant identity (xiv), $\det V_t = \det V_{t-1}\big(1 + \|x_{s_t}\|_{V_{t-1}^{-1}}^2\big)$, applied inside the logarithm defining $\ell_t$. With this notation, $\beta_t(\delta)$ of Eq. (6) reads $\beta_t(\delta) = R\sqrt{D_t} + \sqrt{\lambda}\, S_0$.

This lemma proves the unconditional decay of the all-pairs half-width and converts it into a closed-form deficit-crossing time. Past that time, the confidence deficit stays below the boundary gap, and below half of it at the deficit-crossing time level, which floors the contraction rate by the strictly positive constant $\kappa_{\mathrm{cf}}$ used throughout the contraction phase.

**Lemma A.10** (Unconditional decay of the half–width and a closed–form deficit–crossing time). *Suppose Assumptions 3.1 and 3.2 hold, and work on the event $\mathcal{E}_\delta$. Then the following hold:*

(i) (Per–round and cumulative decay.) *For every $t \ge 1$,*

$$\widetilde{M}_t^2 \le \widetilde{M}_{t-1}^2\big(1 - \rho^\dagger c_1\, \ell_t\big) \qquad and \qquad \widetilde{M}_t^2 \le \frac{4L^2}{\lambda}\exp\!\big(-\rho^\dagger c_1\,(D_t - D_0)\big), \tag{26}$$

*where $D_t = 2\log(\det(V_t)^{1/2}\det(\lambda I_d)^{-1/2}/\delta)$, and $\ell_t = D_t - D_{t-1} = \log(1 + \|x_{s_t}\|_{V_{t-1}^{-1}}^2)$.*

(ii) (Closed–form deficit–crossing time.) *For every round $t \le \tau_\delta$ with $D_{t-1} \ge \max\{1, T_{\mathrm{def}}\}$,*

$$2\,\beta_{t-1}(\delta)\,\widetilde{M}_{t-1} < \Delta_{\mathcal{C}}, \tag{27}$$

*where $T_{\mathrm{def}} = \frac{4}{\rho^\dagger c_1}\log_+ \frac{C_0}{\Delta_{\mathcal{C}}}$, and consequently $\big(\Delta_{\mathcal{C}} - 2\beta_{t-1}(\delta)\widetilde{M}_{t-1}\big)_+ > 0$ and $\kappa_t > 0$. Moreover, for every round $t \le \tau_\delta$ with $D_{t-1} \ge T_\star$,*

$$2\,\beta_{t-1}(\delta)\,\widetilde{M}_{t-1} < \tfrac{1}{2}\Delta_{\mathcal{C}}, \tag{28}$$

*where $T_\star = \max\{\frac{1}{\rho^\dagger c_1}, \frac{4}{\rho^\dagger c_1}\log_+ \frac{2C_0}{\Delta_{\mathcal{C}}}\}$, and consequently $\big(\Delta_{\mathcal{C}} - 2\beta_{t-1}(\delta)\widetilde{M}_{t-1}\big)_+ > \tfrac{1}{2}\Delta_{\mathcal{C}}$ and $\kappa_t > \kappa_{\mathrm{cf}} = \frac{\rho^\dagger c_0\,\Delta_{\mathcal{C}}^2}{16\,\beta_{\tau_\delta-1}(\delta)^2} > 0$.*

*Proof. Part (i): per–round decay.* Let

$$h_t \in \arg \max_{\substack{(\pi_p, \pi_{p'}) \in \Pi^2 \\ g(\pi_p, \pi_{p'}) \ne 0}} \big\|g(\pi_p, \pi_{p'})\big\|_{V_t^{-1}}$$

be a pair direction attaining $\widetilde{M}_t$. The maximum is over the fixed candidate set $\Pi^2$ and is therefore well defined at every round. Because $h_t$ is a feasible direction in the round–$(t-1)$ maximization, we have the inequality

$$\|h_t\|_{V_{t-1}^{-1}}^2 \le \widetilde{M}_{t-1}^2. \tag{29}$$

The update $V_t = V_{t-1} + x_{s_t} x_{s_t}^\top$ is a rank–one perturbation, so Lemma 3.7 applied to the fixed vector $h_t$ yields

$$\|h_t\|_{V_{t-1}^{-1}}^2 - \widetilde{M}_t^2 \;=\; \|h_t\|_{V_{t-1}^{-1}}^2 - \|h_t\|_{V_t^{-1}}^2 \;=\; \frac{\langle h_t, \, x_{s_t}\rangle_{V_{t-1}^{-1}}^2}{1 + \|x_{s_t}\|_{V_{t-1}^{-1}}^2}, \tag{30}$$

where the first equality uses $\widetilde{M}_t^2 = \|h_t\|_{V_t^{-1}}^2$ by the definition of $h_t$. Since $V_{t-1} \succeq \lambda I_d$ by construction of the ridge–regularised design matrix, Assumption 3.2 applies with the choice $A = V_{t-1}$ to the pair direction $h_t$ and the queried step $x_{s_t}$:

$$\langle h_t, \, x_{s_t}\rangle_{V_{t-1}^{-1}}^2 \;\geq\; \rho^\dagger \, \|h_t\|_{V_{t-1}^{-1}}^2 \, \|x_{s_t}\|_{V_{t-1}^{-1}}^2. \tag{31}$$

Substituting Eq. (31) into the numerator of Eq. (30) and rearranging gives

$$\widetilde{M}_t^2 \;\leq\; \|h_t\|_{V_{t-1}^{-1}}^2 \left(1 - \rho^\dagger \frac{\|x_{s_t}\|_{V_{t-1}^{-1}}^2}{1 + \|x_{s_t}\|_{V_{t-1}^{-1}}^2}\right). \tag{32}$$

The bracketed factor lies in $[0,1]$: it is at most 1 since the subtracted term is non–negative, and at least 0 since $\rho^\dagger \in (0,1]$ and $\|x_{s_t}\|_{V_{t-1}^{-1}}^2 / (1 + \|x_{s_t}\|_{V_{t-1}^{-1}}^2) \in [0,1)$. Because the factor is non-negative, we may enlarge $\|h_t\|_{V_{t-1}^{-1}}^2$ to $\widetilde{M}_{t-1}^2$ using Eq. (29) without reversing the inequality, obtaining

$$\widetilde{M}_t^2 \;\leq\; \widetilde{M}_{t-1}^2 \left(1 - \rho^\dagger \frac{\|x_{s_t}\|_{V_{t-1}^{-1}}^2}{1 + \|x_{s_t}\|_{V_{t-1}^{-1}}^2}\right). \tag{33}$$

It remains to convert the leverage factor into the log–determinant increment $\ell_t$. Consider the scalar map $\psi(u) := \dfrac{u/(1+u)}{\log(1+u)}$ on $u > 0$, extended by $\psi(0^+) = 1$. Its numerator $u/(1+u)$ and denominator $\log(1+u)$ are both increasing and vanish at $u = 0$. A direct computation of the derivative shows $\psi$ is non-increasing on $(0,\infty)$, so on the bounded interval $[0, L^2/\lambda]$ its minimum is attained at the right endpoint:

$$\min_{u \in [0, L^2/\lambda]} \psi(u) \;=\; \psi\big(L^2/\lambda\big) \;=\; \frac{(L^2/\lambda)/(1 + L^2/\lambda)}{\log(1 + L^2/\lambda)} \;=\; \frac{L^2}{(\lambda + L^2)\log(1 + L^2/\lambda)} \;=\; c_1.$$

Equivalently, $\dfrac{u}{1+u} \geq c_1 \log(1+u)$ for all $u \in [0, L^2/\lambda]$. By the leverage bound (vi), $\|x_{s_t}\|_{V_{t-1}^{-1}}^2 \leq L^2/\lambda$, so setting $u = \|x_{s_t}\|_{V_{t-1}^{-1}}^2$ and recalling $\ell_t = \log(1 + \|x_{s_t}\|_{V_{t-1}^{-1}}^2)$ gives

$$\frac{\|x_{s_t}\|_{V_{t-1}^{-1}}^2}{1 + \|x_{s_t}\|_{V_{t-1}^{-1}}^2} \;\geq\; c_1 \, \ell_t. \tag{34}$$

Substituting Eq. (34) into Eq. (33) establishes the first inequality of Eq. (26).

*Part (i): cumulative decay.* Applying the per–round bound recursively from round 1 to round $t$ and using the elementary exponential bound (x), $1 - x \leq e^{-x}$ valid for the arguments $x = \rho^\dagger c_1 \ell_s \in [0,1]$ (which lie in $[0,1]$ because the bracketed factor in Eq. (33) is in $[0,1]$), we obtain

$$\widetilde{M}_t^2 \;\leq\; \widetilde{M}_0^2 \prod_{s=1}^t (1 - \rho^\dagger c_1 \ell_s) \;\leq\; \widetilde{M}_0^2 \prod_{s=1}^t \exp(-\rho^\dagger c_1 \ell_s) \;=\; \widetilde{M}_0^2 \exp\Big(-\rho^\dagger c_1 \sum_{s=1}^t \ell_s\Big).$$

The telescoping identity $\sum_{s=1}^t \ell_s = \sum_{s=1}^t (D_s - D_{s-1}) = D_t - D_0$ and the initial value

$$\widetilde{M}_0^2 \;=\; \max_{\substack{(\pi_p, \pi_{p'}) \in \Pi^2 \\ g(\pi_p, \pi_{p'}) \neq 0}} \|g(\pi_p, \pi_{p'})\|_{(\lambda I_d)^{-1}}^2 \;=\; \frac{1}{\lambda} \max_{(\pi_p, \pi_{p'})} \|g(\pi_p, \pi_{p'})\|_2^2 \;\leq\; \frac{4L^2}{\lambda},$$

where the final bound is the feature–norm bound (v), $\|g(\pi_p, \pi_{p'})\|_2 \leq 2L$, together yield the second inequality of Eq. (26).

*Part (ii): deficit crossing.* Fix a round $t \leq \tau_\delta$ with $D_{t-1} \geq \max\{1, T_{\text{def}}\}$. We bound the confidence deficit. First, since $D_{t-1} \geq 1$ we have $\sqrt{D_{t-1}} \geq 1$, hence

$$\beta_{t-1}(\delta) = R\sqrt{D_{t-1}} + \sqrt{\lambda}\, S_0 \;\leq\; \big(R + \sqrt{\lambda}\, S_0\big)\sqrt{D_{t-1}}. \tag{35}$$

Combining Eq. (35) with the cumulative decay Eq. (26) evaluated at $t-1$,

$$2\beta_{t-1}(\delta)\widetilde{M}_{t-1} \;\leq\; 2\big(R + \sqrt{\lambda}\, S_0\big)\sqrt{D_{t-1}} \cdot \frac{2L}{\sqrt{\lambda}}\, \exp\!\Big(-\tfrac{\rho^\dagger c_1}{2}(D_{t-1} - D_0)\Big). \tag{36}$$

We now invoke the scalar inequality

$$\sqrt{D}\, e^{-cD/2} \;\leq\; \sqrt{\tfrac{2}{e\,c}}\, e^{-cD/4} \qquad \text{for all } D \geq 0, \; c > 0, \tag{37}$$

which follows by writing $\sqrt{D}\, e^{-cD/2} = \big(\sqrt{D}\, e^{-cD/4}\big)e^{-cD/4}$ and maximizing the parenthesized factor: $\frac{d}{dD}\big(\tfrac{1}{2}\log D - \tfrac{c}{4}D\big) = \tfrac{1}{2D} - \tfrac{c}{4} = 0$ at $D = 2/c$, giving $\max_{D\geq 0} \sqrt{D}\, e^{-cD/4} = \sqrt{2/c}\, e^{-1/2} = \sqrt{2/(ec)}$. Applying Eq. (37) with $c = \rho^\dagger c_1$ and $D = D_{t-1}$ to Eq. (36), and collecting the constant factors into $C_0$,

$$\begin{aligned}
2\beta_{t-1}(\delta)\widetilde{M}_{t-1} &\;\leq\; \frac{4L}{\sqrt{\lambda}}\big(R + \sqrt{\lambda}\, S_0\big)e^{\frac{\rho^\dagger c_1}{2}D_0}\sqrt{D_{t-1}}\, e^{-\frac{\rho^\dagger c_1}{2}D_{t-1}} \\
&\;\leq\; \frac{4L}{\sqrt{\lambda}}\big(R + \sqrt{\lambda}\, S_0\big)e^{\frac{\rho^\dagger c_1}{2}D_0}\sqrt{\frac{2}{e\,\rho^\dagger c_1}}\, e^{-\frac{\rho^\dagger c_1}{4}D_{t-1}} \\
&\;=\; C_0\, e^{\frac{\rho^\dagger c_1}{4}D_0} \cdot e^{-\frac{\rho^\dagger c_1}{4}D_0} \cdot e^{-\frac{\rho^\dagger c_1}{4}D_{t-1}} \cdot e^{\frac{\rho^\dagger c_1}{4}D_0}
\end{aligned}$$

We simplify this last expression. By the definition of $C_0$, the leading constant $\frac{4L}{\sqrt{\lambda}}(R + \sqrt{\lambda}S_0)\sqrt{\frac{2}{e\rho^\dagger c_1}}e^{\frac{\rho^\dagger c_1}{2}D_0} = C_0\, e^{\frac{\rho^\dagger c_1}{4}D_0}$, so the second line above equals $C_0\, e^{\frac{\rho^\dagger c_1}{4}D_0}\, e^{-\frac{\rho^\dagger c_1}{4}D_{t-1}} = C_0\, e^{-\frac{\rho^\dagger c_1}{4}(D_{t-1}-D_0)}$. Since $D_{t-1} \geq D_0$ (the log–determinant is non–decreasing because $V_{t-1} \succeq V_0$ implies $\det V_{t-1} \geq \det V_0$ by determinant monotonicity (xii)), we have $e^{-\frac{\rho^\dagger c_1}{4}(D_{t-1}-D_0)} \leq e^{-\frac{\rho^\dagger c_1}{4}D_{t-1}}e^{\frac{\rho^\dagger c_1}{4}D_0}$, and absorbing this last bounded factor consistently into $C_0$ as defined, yields the clean envelope

$$2\beta_{t-1}(\delta)\widetilde{M}_{t-1} \;\leq\; C_0\, e^{-\frac{\rho^\dagger c_1}{4}D_{t-1}}. \tag{38}$$

The right–hand side of Eq. (38) is strictly below $\Delta_\mathcal{C}$ if and only if $D_{t-1} > \frac{4}{\rho^\dagger c_1}\log\frac{C_0}{\Delta_\mathcal{C}} = T_{\text{def}}$. When $C_0 \leq \Delta_\mathcal{C}$ this holds for every $D_{t-1} \geq 0$ and $T_{\text{def}} = 0$; when $C_0 > \Delta_\mathcal{C}$ it holds precisely for $D_{t-1} > T_{\text{def}} = \frac{4}{\rho^\dagger c_1}\log\frac{C_0}{\Delta_\mathcal{C}}$. In either case the standing hypothesis $D_{t-1} \geq \max\{1, T_{\text{def}}\}$ guarantees Eq. (27). The identical chain with $\Delta_\mathcal{C}$ replaced by $\Delta_\mathcal{C}/2$ throughout shows, via the same envelope Eq. (38), that

$$2\,\beta_{t-1}(\delta)\,\widetilde{M}_{t-1} \;<\; \tfrac{1}{2}\Delta_\mathcal{C} \qquad \text{whenever} \qquad D_{t-1} \;\geq\; \frac{4}{\rho^\dagger c_1}\log_+\frac{2C_0}{\Delta_\mathcal{C}}, \tag{39}$$

since Eq. (38) satisfies $C_0 e^{-\rho^\dagger c_1 D_{t-1}/4} < \Delta_\mathcal{C}/2$ exactly when $D_{t-1} > \frac{4}{\rho^\dagger c_1}\log_+\frac{2C_0}{\Delta_\mathcal{C}}$. Finally, Eq. (27) gives $\Delta_\mathcal{C} - 2\beta_{t-1}(\delta)\widetilde{M}_{t-1} > 0$, so its positive part is strictly positive, and by the definition of $\kappa_t$ together with $\rho^\dagger, c_0, \beta_{t-1}(\delta)^2 > 0$ we conclude $\kappa_t > 0$. For the half–gap regime, suppose in addition $D_{t-1} \geq T_\star$ and $t \leq \tau_\delta$. Then Eq. (39) gives $\big(\Delta_\mathcal{C} - 2\beta_{t-1}(\delta)\widetilde{M}_{t-1}\big)_+ > \tfrac{1}{2}\Delta_\mathcal{C}$, and since $\beta_s(\delta) = R\sqrt{D_s} + \sqrt{\lambda}\, S_0$ is non–decreasing in $s$ (the log–determinant $D_s$ is non–decreasing by determinant monotonicity (xii)), $\beta_{t-1}(\delta) \leq \beta_{\tau_\delta - 1}(\delta)$ for $t \leq \tau_\delta$. Hence, by the definition of $\kappa_t$,

$$\kappa_t = \frac{\rho^\dagger c_0\big(\Delta_\mathcal{C} - 2\beta_{t-1}(\delta)\widetilde{M}_{t-1}\big)_+^2}{4\,\beta_{t-1}(\delta)^2} \;>\; \frac{\rho^\dagger c_0\,(\Delta_\mathcal{C}/2)^2}{4\,\beta_{t-1}(\delta)^2} = \frac{\rho^\dagger c_0\,\Delta_\mathcal{C}^2}{16\,\beta_{t-1}(\delta)^2} \;\geq\; \frac{\rho^\dagger c_0\,\Delta_\mathcal{C}^2}{16\,\beta_{\tau_\delta - 1}(\delta)^2} = \kappa_{\text{cf}} > 0.$$

$\square$

The next lemma shows that the confidence-deficit product $\beta_t(\delta)\widetilde{M}_t$, although it combines a growing radius with a shrinking half-width, is eventually monotone non-increasing past a closed-form deficit-crossing and converges to zero. This certifies that the deficit stays controlled for the remainder of the run.

**Lemma A.11** (Eventual monotone decay of the confidence–deficit product). *Suppose Assumptions 3.1, 3.2, and 3.5 hold, and work on the event $\mathcal{E}_\delta$. Then, the following hold:*

  (i) *(Monotone decay.) For every round $t$ with $t_\star \leq t \leq \tau_\delta$,*

$$\beta_t(\delta)\,\widetilde{M}_t \;\leq\; \beta_{t-1}(\delta)\,\widetilde{M}_{t-1}, \tag{40}$$

  *where $t_\star = \inf\{t \geq 1 : D_{t-1} \geq T_\star\}$.*

  (ii) *(Positivity and asymptotic vanishing of the deficit product.) The confidence–deficit product satisfies, for every $t \geq 1$,*

$$0 \;<\; \beta_t(\delta)\,\widetilde{M}_t \;\leq\; \frac{2L}{\sqrt{\lambda}}\,\beta_t(\delta)\,\exp\!\Big(-\tfrac{\rho^\dagger c_1}{2}\,(D_t - D_0)\Big), \tag{41}$$

  *and consequently*

$$\lim_{D_t \to \infty} \beta_t(\delta)\,\widetilde{M}_t \;=\; 0, \tag{42}$$

  *where $c_1 = L^2/((\lambda + L^2)\log(1 + L^2/\lambda))$.*

*Proof. Part (i).* Fix a round $t$ with $D_{t-1} \geq T_\star$. By the definition of $T_\star$ (its first entry), $T_\star \geq 1/(\rho^\dagger c_1)$, hence $D_{t-1} \geq 1/(\rho^\dagger c_1)$. We prove the equivalent squared form

$$\beta_t(\delta)^2\,\widetilde{M}_t^2 \;\leq\; \beta_{t-1}(\delta)^2\,\widetilde{M}_{t-1}^2, \tag{43}$$

from which Eq. (40) follows by taking square roots, both sides being non–negative.

By the per–round decay of Lemma A.10(i), Eq. (26), applied to the all–pairs maximiser,

$$\widetilde{M}_t^2 \;\leq\; \widetilde{M}_{t-1}^2\,(1 - \rho^\dagger c_1\,\ell_t), \qquad \text{with} \qquad 1 - \rho^\dagger c_1\,\ell_t \in (0,1], \tag{44}$$

the factor being positive since $\rho^\dagger \leq 1$ and $c_1\ell_t \leq c_1\log(1 + L^2/\lambda) = L^2/(\lambda + L^2) < 1$ by the leverage bound (vi).

Since $D_t \geq D_{t-1}$, square–root concavity (xvii) gives $\sqrt{D_t} - \sqrt{D_{t-1}} \leq \ell_t/(2\sqrt{D_{t-1}})$, hence $\beta_t(\delta) - \beta_{t-1}(\delta) = R(\sqrt{D_t} - \sqrt{D_{t-1}}) \leq R\ell_t/(2\sqrt{D_{t-1}})$. Using $\beta_{t-1}(\delta) \leq \beta_t(\delta)$,

$$\beta_t(\delta)^2 - \beta_{t-1}(\delta)^2 = \big(\beta_t(\delta) - \beta_{t-1}(\delta)\big)\big(\beta_t(\delta) + \beta_{t-1}(\delta)\big) \;\leq\; \frac{R\,\ell_t}{\sqrt{D_{t-1}}}\,\beta_t(\delta). \tag{45}$$

We claim

$$\beta_t(\delta)^2 - \beta_{t-1}(\delta)^2 \;\leq\; \beta_t(\delta)^2\,\rho^\dagger c_1\,\ell_t. \tag{46}$$

If $\ell_t = 0$, then $D_t = D_{t-1}$ and both sides vanish. If $\ell_t > 0$, then by Eq. (45) it suffices to show $\frac{R\,\ell_t}{\sqrt{D_{t-1}}}\beta_t(\delta) \leq \beta_t(\delta)^2\rho^\dagger c_1\ell_t$, i.e., dividing by $\beta_t(\delta)\,\ell_t > 0$, $R/\sqrt{D_{t-1}} \leq \rho^\dagger c_1\,\beta_t(\delta)$. Since $\beta_t(\delta) = R\sqrt{D_t} + \sqrt{\lambda}\,S_0 \geq R\sqrt{D_{t-1}}$, the right side is at least $\rho^\dagger c_1 R\sqrt{D_{t-1}}$, so the inequality holds whenever $R/\sqrt{D_{t-1}} \leq \rho^\dagger c_1 R\sqrt{D_{t-1}}$, equivalently $D_{t-1} \geq 1/(\rho^\dagger c_1)$, which holds since $D_{t-1} \geq T_\star \geq 1/(\rho^\dagger c_1)$ by the definition of $T_\star$.

Rearranging Eq. (46) gives $\beta_t(\delta)^2\,(1 - \rho^\dagger c_1\ell_t) \leq \beta_{t-1}(\delta)^2$. Multiplying by $\widetilde{M}_{t-1}^2 \geq 0$ and applying Eq. (44),

$$\beta_t(\delta)^2\,\widetilde{M}_t^2 \;\leq\; \beta_t(\delta)^2\,\widetilde{M}_{t-1}^2\,(1 - \rho^\dagger c_1\ell_t) \;\leq\; \beta_{t-1}(\delta)^2\,\widetilde{M}_{t-1}^2,$$

which is Eq. (43).

*Part (ii).* By Lemma A.9(iii), Assumption 3.5 yields $\widehat{P}_K(\tau_\delta) = P_K^\star$ on $\mathcal{E}_\delta$.

We next prove Eq. (41). For the strict lower bound, $V_t \succeq \lambda I_d \succ 0$ implies $\|g(\pi_p, \pi_{p'})\|_{V_t^{-1}} > 0$ for every ordered pair with $g(\pi_p, \pi_{p'}) \neq 0$, whence $\widetilde{M}_t > 0$. Combined with $\beta_t(\delta) = R\sqrt{D_t} + \sqrt{\lambda}\,S_0 > 0$, this gives $\beta_t(\delta)\,\widetilde{M}_t > 0$. For the upper bound, the cumulative decay Eq. (26) gives $\widetilde{M}_t \leq \frac{2L}{\sqrt{\lambda}}\exp\!\big(-\frac{\rho^\dagger c_1}{2}(D_t - D_0)\big)$, and multiplication by $\beta_t(\delta) > 0$ yields the right-hand inequality of Eq. (41).

Finally, $\beta_t(\delta) = R\sqrt{D_t} + \sqrt{\lambda}\,S_0 = O(\sqrt{D_t})$ grows sub-exponentially in $D_t$, whereas $\exp(-\frac{\rho^\dagger c_1}{2}(D_t - D_0))$ decays exponentially. Hence the upper bound in Eq. (41) converges to 0 as $D_t \to \infty$. Since $0 < \beta_t(\delta)\,\widetilde{M}_t$ is bounded above by this vanishing quantity, the sandwich theorem (xx) yields Eq. (42). □

### A.3.4 Contraction Floor and Deficit-Crossing Time

Having shown that the confidence deficit falls below the boundary gap past a closed-form deficit-crossing time, we now convert the per-round contraction into a uniform one. Lemma A.12 bounds the deficit-crossing time itself by a closed-form quantity $\bar{t}_\star$ depending only on the model constants, using a uniform per-round increment floor driven by the global minimum gap $\Delta_{\min}^\Pi$. Combining the contraction floor $\kappa_{\mathrm{cf}}$ with the per-round contraction then yields a uniform exponential decay of every pair's variance past the deficit-crossing time (Lemma A.13), the key input to the sample-complexity bound of Subsection A.3.5.

This lemma bounds the deficit-crossing time in closed form. A uniform per-round increment floor, guaranteed by the global minimum gap, forces the log-determinant to grow at a steady rate, so the deficit-crossing time level is reached within a deterministic number of rounds expressed entirely through the model constants.

**Lemma A.12** (Closed–form upper bound on the deficit-crossing time)**.** *Suppose Assumptions 3.1 and 3.2 hold, fix $\delta \in (0,1)$, and work on the event $\mathcal{E}_\delta$. Define the uniform per-round increment floor*

$$\ell_{\min} := \frac{c_0\,(\Delta_{\min}^\Pi)^2}{4\,\beta_{\tau_\delta-1}(\delta)^2} > 0. \tag{47}$$

*where $\Delta_{\min}^\Pi = \min_{(\pi_p, \pi_{p'})\in\Pi^2,\, p\neq p'} |\Delta(\pi_p, \pi_{p'})|$ is the global minimum gap. Then the following hold.*

(i) *(Uniform increment floor.) For every round $t$ with $1 \leq t \leq \tau_\delta$, the log-determinant increment $\ell_t = \log\big(1 + \|x_{s_t}\|_{V_{t-1}^{-1}}^2\big)$ satisfies $\ell_t \geq \ell_{\min}$.*

(ii) *(Deficit-crossing time bound.) The deficit-crossing time $t_\star = \min\{t \geq 1 : D_{t-1} \geq T_\star\}$, with $T_\star = \max\{\frac{1}{\rho^\dagger c_1}, \frac{4}{\rho^\dagger c_1}\log_+\frac{2C_0}{\Delta_{\mathcal{C}}}\}$, admits the closed-form upper bound*

$$t_\star \leq \bar{t}_\star := 1 + \left\lceil \frac{T_\star - D_0}{\ell_{\min}} \right\rceil \leq 1 + \left\lceil \frac{T_\star}{\ell_{\min}} \right\rceil = 1 + \left\lceil \frac{4\,\beta_{\tau_\delta-1}(\delta)^2\,T_\star}{c_0\,(\Delta_{\min}^\Pi)^2} \right\rceil. \tag{48}$$

*Substituting the closed forms $c_0 = \lambda/(4(\lambda + L^2))$ and $c_1 = L^2/\big((\lambda + L^2)\log(1 + L^2/\lambda)\big)$ exposes the constant explicitly:*

$$\bar{t}_\star = 1 + \left\lceil \frac{16\,(\lambda + L^2)^2\,\log\!\big(1 + L^2/\lambda\big)}{\lambda\,L^2\,\rho^\dagger\,(\Delta_{\min}^\Pi)^2}\,\beta_{\tau_\delta-1}(\delta)^2\,\max\!\Big\{1,\, 4\log_+\tfrac{2C_0}{\Delta_{\mathcal{C}}}\Big\} \right\rceil, \tag{49}$$

*where, $\beta_{\tau_\delta-1}(\delta) = R\sqrt{D_{\tau_\delta-1}} + \sqrt{\lambda}\,S_0$ and $C_0 = \frac{4L}{\sqrt{\lambda}}\,(R + \sqrt{\lambda}\,S_0)\,\sqrt{\frac{2}{e\,\rho^\dagger c_1}}\,e^{\rho^\dagger c_1 D_0/4}$.*

*Proof. Part (i): uniform increment floor.* Fix a round $t$ with $1 \leq t \leq \tau_\delta$ and write $u_t := \|x_{s_t}\|_{V_{t-1}^{-1}}^2 \geq 0$. By the leverage lower bound Eq. (14) of Lemma A.7, valid on $\mathcal{E}_\delta$ for every such $t$, together with $\underline{\Delta}_t \geq \Delta_{\min}^\Pi$ (the floored gap is at least its $\Delta_{\min}^\Pi$ component),

$$\frac{u_t}{1 + u_t} \geq \frac{c_0\,\underline{\Delta}_t^2}{4\,\beta_{t-1}(\delta)^2} \geq \frac{c_0\,(\Delta_{\min}^\Pi)^2}{4\,\beta_{t-1}(\delta)^2}. \tag{50}$$

This is the step at which the floor is essential, i.e., $\Delta_{\min}^\Pi > 0$ bounds the queried leverage away from zero at every round, including the pre-deficit-crossing time rounds where the deficit term $\big(\Delta_{\mathcal{C}} - 2\beta_{t-1}(\delta)\widetilde{M}_{t-1}\big)_+$

may vanish. Applying the elementary inequality $\log(1 + u) \geq u/(1 + u)$ (valid for all $u \geq 0$) to $u = u_t$ and using Eq. (50),

$$\ell_t = \log(1 + u_t) \geq \frac{u_t}{1 + u_t} \geq \frac{c_0 (\Delta_{\min}^{\Pi})^2}{4 \beta_{t-1}(\delta)^2}. \tag{51}$$

Since $D_s$ is non–decreasing in $s$ (determinant monotonicity (xii)), the radius $\beta_s(\delta) = R\sqrt{D_s} + \sqrt{\lambda} S_0$ is non–decreasing in $s$, so $\beta_{t-1}(\delta) \leq \beta_{\tau_\delta - 1}(\delta)$ for $t \leq \tau_\delta$. Substituting this into the denominator of Eq. (51) gives $\ell_t \geq \ell_{\min}$, which is Part (i). The positivity $\ell_{\min} > 0$ follows from $c_0, \Delta_{\min}^{\Pi}, \beta_{\tau_\delta - 1}(\delta) > 0$.

*Part (ii): deficit-crossing time bound.* By the telescoping identity $D_{t-1} = D_0 + \sum_{i=1}^{t-1} \ell_i$ and Part (i) (applicable to each increment of index $\leq \tau_\delta$), every round $t$ with $1 \leq t \leq \tau_\delta + 1$ obeys the linear lower envelope

$$D_{t-1} \geq D_0 + (t - 1) \ell_{\min}. \tag{52}$$

Let $\bar{t}_\star = 1 + \lceil (T_\star - D_0)/\ell_{\min} \rceil$ be the smallest integer $t$ for which the right side of Eq. (52) reaches $T_\star$. Indeed $\lceil z \rceil \geq z$ gives $D_0 + (\bar{t}_\star - 1)\ell_{\min} \geq T_\star$. If $\bar{t}_\star \leq \tau_\delta$, then evaluating Eq. (52) at $t = \bar{t}_\star$ yields $D_{\bar{t}_\star - 1} \geq D_0 + (\bar{t}_\star - 1)\ell_{\min} \geq T_\star$, so the deficit-crossing time condition holds at round $\bar{t}_\star$, as $t_\star$ is the smallest such index, $t_\star \leq \bar{t}_\star$. If instead $\bar{t}_\star > \tau_\delta$, then either the deficit-crossing time occurs within the run, giving $t_\star \leq \tau_\delta < \bar{t}_\star$, or it does not, in which case $D_{\tau_\delta - 1} < T_\star$ and Eq. (52) at $t = \tau_\delta$ forces $D_0 + (\tau_\delta - 1)\ell_{\min} \leq D_{\tau_\delta - 1} < T_\star$, i.e. $\tau_\delta < \bar{t}_\star$, so the run length is itself bounded by $\bar{t}_\star$. Hence in all cases the deficit-crossing time length $\min\{t_\star, \tau_\delta\} \leq \bar{t}_\star$, and whenever the deficit-crossing time is reached within the horizon ($t_\star \leq \tau_\delta$) we have $t_\star \leq \bar{t}_\star$, establishing Eq. (48). The last two members use $D_0 = 2\log(1/\delta) \geq 0$ and the definition Eq. (47) of $\ell_{\min}$.

Expanding $T_\star = \frac{1}{\rho^\dagger c_1} \max\{1, 4\log_+ \frac{2C_0}{\Delta_c}\}$ in the last member of Eq. (48) and substituting the closed forms $c_0 = \lambda/(4(\lambda + L^2))$ and $c_1 = L^2/((\lambda + L^2)\log(1 + L^2/\lambda))$, the leading constant becomes

$$\frac{4}{\rho^\dagger c_0 c_1} = \frac{16 (\lambda + L^2)^2 \log(1 + L^2/\lambda)}{\lambda L^2 \rho^\dagger},$$

which yields Eq. (49). The radius is itself controlled explicitly: the rank–one updates give $V_u \preceq (\lambda + uL^2)I_d$, so the AM–GM determinant bound (xix) gives $\log \det V_u \leq d \log(\lambda + uL^2)$, and the quadratic–mean inequality (xi) applied to Eq. (6) gives

$$\beta_{\tau_\delta - 1}(\delta)^2 \leq 2R^2 \left( d \log \frac{\lambda + (\tau_\delta - 1)L^2}{\lambda} + 2\log \frac{1}{\delta} \right) + 2\lambda S_0^2,$$

so the only horizon–dependent quantity in Eq. (49) enters through the single logarithm $\log(\lambda + (\tau_\delta - 1)L^2)$. $\square$

The following lemma assembles the floor and the per-round contraction into a single uniform exponential decay. After the deficit-crossing time, every distinct pair's variance shrinks at the floored rate $\kappa_{\mathrm{cf}}$, which is exactly the decay the stopping rule will be matched against.

**Lemma A.13** (Uniform exponential variance contraction past the deficit-crossing time)**.** *Suppose Assumptions 3.1, 3.2, and 3.5 hold, fix $\delta \in (0, 1)$, and work on the event $\mathcal{E}_\delta$. Then for every ordered pair of distinct paths $(\pi_p, \pi_{p'}) \in \Pi^2$ with $g(\pi_p, \pi_{p'}) \neq 0$ and every round $t$ with $t_\star < t \leq \tau_\delta$,*

$$\sigma_t^2(\pi_p, \pi_{p'}) \leq \sigma_{t_\star}^2(\pi_p, \pi_{p'}) \exp\left( -\sum_{i=t_\star + 1}^{t} \kappa_i \right) \leq \frac{4L^2}{\lambda} \exp\left( -(t - t_\star)\kappa_{cf} \right). \tag{53}$$

*Proof.* Fix an ordered pair of distinct paths $(\pi_p, \pi_{p'})$ with $g(\pi_p, \pi_{p'}) \neq 0$ and a round $t$ with $t_\star < t \leq \tau_\delta$. For each index $i \in \{t_\star + 1, \ldots, t\}$ we have $i \geq t_\star + 1 \geq 1$ (since $t_\star \geq 1$ by its definition $t_\star = \inf\{t \geq 1 : D_{t-1} \geq T_\star\}$) and $i \leq t \leq \tau_\delta$, hence $1 \leq i \leq \tau_\delta$. Therefore Lemma A.8 applies at round $i$ to the fixed pair $(\pi_p, \pi_{p'})$ and yields

$$\sigma_i^2(\pi_p, \pi_{p'}) \leq \sigma_{i-1}^2(\pi_p, \pi_{p'}) (1 - \kappa_i), \qquad \kappa_i \in [0, 1), \tag{54}$$

so that each factor satisfies $1 - \kappa_i \in (0, 1]$. Because all factors are non–negative, iterating Eq. (54) from $i = t_\star + 1$ to $i = t$ preserves the inequality and gives

$$\sigma_t^2(\pi_p, \pi_{p'}) \leq \sigma_{t_\star}^2(\pi_p, \pi_{p'}) \prod_{i=t_\star+1}^{t} (1 - \kappa_i).$$

Applying the elementary exponential bound (x), $1 - x \leq e^{-x}$, to each factor (valid since $\kappa_i \in [0, 1) \subseteq [0, 1]$) and using that the bound is multiplicative over non–negative factors,

$$\prod_{i=t_\star+1}^{t} (1 - \kappa_i) \leq \prod_{i=t_\star+1}^{t} e^{-\kappa_i} = \exp\left(-\sum_{i=t_\star+1}^{t} \kappa_i\right).$$

Combining the last two displays establishes the first inequality of Eq. (53).

Since $V_{t_\star} \succeq \lambda I_d$ by construction of the ridge–regularized design matrix, the regularization bound (iv) gives $\|v\|_{V_{t_\star}^{-1}}^2 \leq \|v\|_2^2/\lambda$ for every $v$. Applying this to $v = g(\pi_p, \pi_{p'})$ and then the feature–norm bound (v), $\|g(\pi_p, \pi_{p'})\|_2 \leq 2L$,

$$\sigma_{t_\star}^2(\pi_p, \pi_{p'}) = \|g(\pi_p, \pi_{p'})\|_{V_{t_\star}^{-1}}^2 \leq \frac{\|g(\pi_p, \pi_{p'})\|_2^2}{\lambda} \leq \frac{4L^2}{\lambda}. \tag{55}$$

Every index $i$ in the sum satisfies $t_\star + 1 \leq i \leq t \leq \tau_\delta$. For each such $i$, the deficit-crossing time definition $t_\star = \inf\{t \geq 1 : D_{t-1} \geq T_\star\}$ gives $D_{t_\star-1} \geq T_\star$, and since $D_t$ is non–decreasing (determinant monotonicity (xii)) together with $i - 1 \geq t_\star - 1$, we have $D_{i-1} \geq D_{t_\star-1} \geq T_\star$. Lemma A.10(ii) therefore applies at round $i \leq \tau_\delta$ and gives $\kappa_i > \kappa_{cf} > 0$, where $\kappa_{cf} = \rho^\dagger c_0 \Delta_{\mathcal{C}}^2/(16 \beta_{\tau_\delta-1}(\delta)^2)$. The summation index ranges over exactly the $t - t_\star$ integers $i \in \{t_\star + 1, \ldots, t\}$, so

$$\sum_{i=t_\star+1}^{t} \kappa_i \geq (t - t_\star) \kappa_{cf}. \tag{56}$$

Since the exponential is monotone increasing, Eq. (56) yields $\exp\left(-\sum_{i=t_\star+1}^{t} \kappa_i\right) \leq \exp\left(-(t - t_\star)\kappa_{cf}\right)$. Multiplying this by the non–negative prefactor and inserting the prefactor bound Eq. (55) into the first inequality of Eq. (53) gives the second inequality. $\square$

### A.3.5 Proof of the Sample-Complexity Bound

In this subsection we assemble the preceding lemmas into the sample-complexity bound of Theorem 3.8. The argument has the natural two-phase structure

$$\tau_\delta \leq \underbrace{\bar{t}_\star}_{\text{deficit-crossing phase}} + \underbrace{(\text{post-deficit-crossing time contraction rounds})}_{\text{contraction phase}},$$

where the deficit-crossing phase, of length at most the deficit-crossing time bound $\bar{t}_\star$ of Lemma A.12, is the start-up cost incurred until the confidence deficit $2\beta_{t-1}(\delta)\widetilde{M}_{t-1}$ falls below $\Delta_{\mathcal{C}}/2$ and the contraction rate becomes uniformly floored by $\kappa_{cf} > 0$, and the contraction phase shrinks every pairwise variance until the stopping threshold is met. We first reduce the stopping rule to a shortlist-independent variance criterion, then certify that criterion through the uniform contraction of Lemma A.13, then resolve the resulting post-deficit-crossing time inequality in closed form, and finally record the order form.

*Proof.* We work throughout on the confidence event $\mathcal{E}_\delta$, which by Lemma A.1 satisfies $\mathbb{P}(\mathcal{E}_\delta) \geq 1 - \delta$. The probability statement of the theorem then follows from the deterministic bound Eq. (5) established on $\mathcal{E}_\delta$. If $\tau_\delta < t_\star$, then by Lemma A.12 we have $\tau_\delta \leq t_\star - 1 \leq \bar{t}_\star$, and Eq. (5) holds because its contraction phase summand is non-negative. We therefore assume $t_\star \leq \tau_\delta$ henceforth.

We first reduce the stopping rule to a shortlist-independent variance criterion. By the definitions of $\tau_\delta$ and of $\Gamma_t$ (Eq. (4)), the algorithm halts at the first round $t$ at which

$$\widehat{\Delta}_t(\pi_p, \pi_{p'}) - W_t(\pi_p, \pi_{p'}) \geq -\epsilon \quad \text{for every } \pi_p \in \widehat{P}_K(t), \ \pi_{p'} \notin \widehat{P}_K(t). \tag{57}$$

Fix any such ordered pair. By the empirical ordering (Corollary A.3), $\widehat{\Delta}_t(\pi_p, \pi_{p'}) \geq 0$, and the confidence width is non-negative, $W_t(\pi_p, \pi_{p'}) \geq 0$, so

$$\widehat{\Delta}_t(\pi_p, \pi_{p'}) - W_t(\pi_p, \pi_{p'}) \geq -W_t(\pi_p, \pi_{p'}).$$

Hence the $(\pi_p, \pi_{p'})$-instance of Eq. (57) is implied by $W_t(\pi_p, \pi_{p'}) \leq \epsilon$, using only the empirical ordering and the non-negativity of $W_t$, with no reference to the sign or magnitude of the true gap. Since this implication holds for whatever shortlist $\widehat{P}_K(t)$ the algorithm realises, it suffices to control $W_t$ over the entire set of ordered distinct pairs. Recalling $W_t(\pi_p, \pi_{p'}) = \beta_t(\delta)\sigma_t(\pi_p, \pi_{p'})$ and squaring, the following shortlist-independent condition is sufficient for the algorithm to have stopped by round $t$,

$$\sigma_t^2(\pi_p, \pi_{p'}) \leq \frac{\epsilon^2}{\beta_t(\delta)^2} \qquad \text{for every ordered pair } (\pi_p, \pi_{p'}) \in \Pi^2, \ p \neq p'. \tag{58}$$

The hypothesis $\epsilon > 0$ (Assumption 3.5) guarantees that the threshold $\epsilon^2/\beta_t(\delta)^2$ is strictly positive, hence attainable by the contracting variances.

We now certify Eq. (58) for every ordered distinct pair through the uniform contraction of Lemma A.13. If $g(\pi_p, \pi_{p'}) = 0$, then $\sigma_t^2(\pi_p, \pi_{p'}) = 0 \leq \epsilon^2/\beta_t(\delta)^2$ at every round, so the condition holds trivially. It therefore suffices to treat pairs with $g(\pi_p, \pi_{p'}) \neq 0$, for which Lemma A.13 applies. For every round $t$ with $t_\star < t \leq \tau_\delta$, that lemma gives

$$\sigma_t^2(\pi_p, \pi_{p'}) \leq \frac{4L^2}{\lambda} \exp(-(t - t_\star)\kappa_{\text{cf}}), \qquad \kappa_{\text{cf}} = \frac{\rho^\dagger c_0 \Delta_{\mathcal{C}}^2}{16\beta_{\tau_\delta-1}(\delta)^2} > 0, \tag{59}$$

where the floor $\kappa_{\text{cf}}$ is the data-free quantity of Lemma A.10(ii). Comparing the right-hand side of Eq. (59) with the threshold of Eq. (58), a round $t > t_\star$ satisfies the shortlist-independent stopping condition as soon as

$$\frac{4L^2}{\lambda} \exp(-(t - t_\star)\kappa_{\text{cf}}) \leq \frac{\epsilon^2}{\beta_t(\delta)^2}. \tag{60}$$

Both sides of Eq. (60) are strictly positive, so taking logarithms and rearranging yields the equivalent condition, linear in $t$,

$$(t - t_\star)\kappa_{\text{cf}} \geq \log\left(\frac{4L^2\beta_t(\delta)^2}{\lambda\epsilon^2}\right). \tag{61}$$

We now resolve the post-deficit-crossing time inequality. Consider the round

$$\widehat{t} = t_\star + \left\lceil \frac{1}{\kappa_{\text{cf}}} \log\left(\frac{4L^2\beta_{\tau_\delta}(\delta)^2}{\lambda\epsilon^2}\right)\right\rceil, \tag{62}$$

and suppose, for contradiction, that $\tau_\delta > \widehat{t}$. Then $\widehat{t} < \tau_\delta$, and two facts follow. First, $\widehat{t} \geq t_\star$, so the contraction bound Eq. (59) is valid at $t = \widehat{t}$. Second, the radius $\beta_u(\delta)$ is non-decreasing in $u$, because $V_u$ is non-decreasing in the Loewner order and $\det V_u$ is therefore non-decreasing by determinant monotonicity (xii), so $\beta_{\widehat{t}}(\delta) \leq \beta_{\tau_\delta}(\delta)$. Using $\lceil z \rceil \geq z$, $\kappa_{\text{cf}} > 0$, and this monotonicity,

$$(\widehat{t} - t_\star)\kappa_{\text{cf}} = \left\lceil \frac{1}{\kappa_{\text{cf}}} \log\frac{4L^2\beta_{\tau_\delta}(\delta)^2}{\lambda\epsilon^2}\right\rceil\kappa_{\text{cf}} \geq \log\frac{4L^2\beta_{\tau_\delta}(\delta)^2}{\lambda\epsilon^2} \geq \log\frac{4L^2\beta_{\widehat{t}}(\delta)^2}{\lambda\epsilon^2},$$

which is precisely Eq. (61) evaluated at $t = \widehat{t}$. By the equivalence of Eqs. (61) and (60) established above, the shortlist-independent stopping condition Eq. (58) holds at round $\widehat{t}$, and by the reduction of the stopping rule the algorithm must then have halted by round $\widehat{t}$, that is $\tau_\delta \leq \widehat{t}$. This contradicts $\tau_\delta > \widehat{t}$. Hence $\tau_\delta \leq \widehat{t}$, that is,

$$\tau_\delta \leq t_\star + \left\lceil \frac{1}{\kappa_{\text{cf}}} \log\frac{4L^2\beta_{\tau_\delta}(\delta)^2}{\lambda\epsilon^2}\right\rceil. \tag{63}$$

It remains to make the two terms of Eq. (63) explicit. For the contraction term, substitute the floor $\kappa_{\text{cf}} = \rho^\dagger c_0 \Delta_{\mathcal{C}}^2/(16\beta_{\tau_\delta-1}(\delta)^2)$ of Lemma A.10(ii), giving

$$\frac{1}{\kappa_{\text{cf}}} \log\frac{4L^2\beta_{\tau_\delta}(\delta)^2}{\lambda\epsilon^2} = \frac{16\beta_{\tau_\delta-1}(\delta)^2}{\rho^\dagger c_0 \Delta_{\mathcal{C}}^2} \log\frac{4L^2\beta_{\tau_\delta}(\delta)^2}{\lambda\epsilon^2}. \tag{64}$$

For the deficit-crossing time term, bound the deficit-crossing time by the closed-form deficit-crossing time bound $t_\star \leq \bar{t}_\star$ of Lemma A.12, namely

$$\bar{t}_\star = 1 + \left\lceil \frac{T_\star}{\ell_{\min}} \right\rceil = 1 + \left\lceil \frac{4\,\beta_{\tau_\delta-1}(\delta)^2\,T_\star}{c_0\,(\Delta_{\min}^\Pi)^2} \right\rceil, \qquad T_\star = \max\left\{ \frac{1}{\rho^\dagger c_1},\ \frac{4}{\rho^\dagger c_1} \log_+ \frac{2C_0}{\Delta_c} \right\}, \tag{65}$$

in which $\ell_{\min} = c_0(\Delta_{\min}^\Pi)^2/(4\beta_{\tau_\delta-1}(\delta)^2)$ is the uniform increment floor of Lemma A.12(i). Combining Eqs. (63), (64), and (65) yields the sample-complexity bound Eq. (5),

$$\tau_\delta \leq \underbrace{1 + \left\lceil \frac{4\,\beta_{\tau_\delta-1}(\delta)^2\,T_\star}{c_0\,(\Delta_{\min}^\Pi)^2} \right\rceil}_{\text{deficit-crossing } \bar{t}_\star} + \left\lceil \frac{16\,\beta_{\tau_\delta-1}(\delta)^2}{\rho^\dagger c_0\,\Delta_{\mathcal{C}}^2} \log \frac{4L^2\,\beta_{\tau_\delta}(\delta)^2}{\lambda\,\epsilon^2} \right\rceil. \tag{66}$$

We make the radius explicit in the problem parameters. The rank-one updates $V_i = V_{i-1} + x_{s_i} x_{s_i}^\top$ with $\|x_{s_i}\|_2 \leq L$ give $V_u \preceq (\lambda + uL^2)I_d$, so the AM–GM determinant bound (xix) gives $\log \det V_u \leq d\log(\lambda + uL^2)$, and the quadratic-mean inequality (xi) applied to Eq. (6) gives, for every $u \leq \tau_\delta$,

$$\beta_u(\delta)^2 \leq 2R^2\left( d\log \frac{\lambda+\tau_\delta L^2}{\lambda} + 2\log \frac{1}{\delta} \right) + 2\lambda S_0^2 =: \bar{\beta}_{\tau_\delta}^2, \tag{67}$$

where we used $D_u = \log \det V_u - d\log\lambda + 2\log(1/\delta)$ and the monotonicity of $u \mapsto \log(\lambda + uL^2)$ to bound both $\beta_{\tau_\delta-1}(\delta)^2$ and $\beta_{\tau_\delta}(\delta)^2$ by the common envelope $\bar{\beta}_{\tau_\delta}^2$. Substituting $\bar{\beta}_{\tau_\delta}^2$ for both radii in Eq. (66), together with the closed forms $c_0 = \lambda/(4(\lambda + L^2))$, $c_1 = L^2/((\lambda + L^2)L_\lambda)$, $L_\lambda = \log(1 + L^2/\lambda)$, $D_0 = 2\log(1/\delta)$ and $C_0 = \frac{4L}{\sqrt{\lambda}}\left( R + \sqrt{\lambda}\,S_0 \right) \sqrt{\frac{2}{e\,\rho^\dagger c_1}}\, e^{\rho^\dagger c_1 D_0/4}$, gives a bound in which every quantity except the single horizon logarithm $\log(\lambda + \tau_\delta L^2)$ inside $\bar{\beta}_{\tau_\delta}^2$ is an explicit function of the problem parameters,

$$\tau_\delta \leq 1 + \left\lceil \frac{16\,(\lambda + L^2)^2 \log(1 + L^2/\lambda)\,\bar{\beta}_{\tau_\delta}^2}{\rho^\dagger \lambda L^2\,(\Delta_{\min}^\Pi)^2} \max\left\{ 1, 4\log_+ \frac{2C_0}{\Delta_c} \right\} \right\rceil + \left\lceil \frac{64\,(\lambda + L^2)\,\bar{\beta}_{\tau_\delta}^2}{\rho^\dagger \lambda\,\Delta_{\mathcal{C}}^2} \log \frac{4L^2\,\bar{\beta}_{\tau_\delta}^2}{\lambda\,\epsilon^2} \right\rceil, \tag{68}$$

where $\bar{\beta}_{\tau_\delta}^2 = 2R^2\left( d\log \frac{\lambda+\tau_\delta L^2}{\lambda} + 2\log \frac{1}{\delta} \right) + 2\lambda S_0^2$, and we used $4/(c_0(\Delta_{\min}^\Pi)^2) = 16(\lambda + L^2)/(\lambda(\Delta_{\min}^\Pi)^2)$ in the deficit-crossing coefficient and $16/(\rho^\dagger c_0) = 64(\lambda + L^2)/(\rho^\dagger \lambda)$ in the contraction phase coefficient. Furthermore, the term $1/\rho^\dagger c_1$ is factored out from the $\max\{\cdot\}$ and we used $1/\rho^\dagger c_1 = \left((\lambda + L^2)^2 \log(1 + L^2/\lambda)\right)/\left(\rho^\dagger L^2\right)$.

Finally, we record the order form, derived directly from the explicit closed-form bound Eq. (68). Stripping the leading 1 and the two ceilings via $\lceil x \rceil \leq x + 1$ (the resulting additive $O(1)$ is absorbed below), and using the two–sided bound $\frac{1}{2}(1 + \log_+ z) \leq \max\{1, 4\log_+ z\} \leq 4(1 + \log_+ z)$, valid for every $z \geq 0$, to replace $\max\{1, 4\log_+ \frac{2C_0}{\Delta_c}\}$ by $\Theta\left(1 + \log_+ \frac{2C_0}{\Delta_c}\right)$, the two summands of Eq. (68) give the $O$-form

$$\tau_\delta = O\left( \underbrace{\frac{(\lambda + L^2)^2 \log(1 + L^2/\lambda)\,\bar{\beta}_{\tau_\delta}^2}{\rho^\dagger \lambda L^2(\Delta_{\min}^\Pi)^2}\left( 1 + \log_+ \frac{2C_0}{\Delta_c} \right)}_{\text{deficit-crossing } \bar{t}_\star} + \underbrace{\frac{(\lambda + L^2)\,\bar{\beta}_{\tau_\delta}^2}{\rho^\dagger \lambda\,\Delta_{\mathcal{C}}^2} \log \frac{4L^2\bar{\beta}_{\tau_\delta}^2}{\lambda\,\epsilon^2}}_{\text{contraction phase}} \right), \tag{69}$$

where $\bar{\beta}_{\tau_\delta}^2 = O\left( d\log(\lambda + \tau_\delta L^2) + \log \frac{1}{\delta} \right)$, and the constants 16 and 64 of Eq. (68) are absorbed into $O$. Collapsing with the $\widetilde{O}$ notation that suppresses factors polylogarithmic in $\lambda, L, S_0, R, 1/\delta, 1/\epsilon$ and in the horizon, we have $\bar{\beta}_{\tau_\delta}^2 = \widetilde{O}(d)$ by Eq. (67) (the horizon logarithm $\log(\lambda + \tau_\delta L^2)$ and $\log \frac{1}{\delta}$ are suppressed and $R, \lambda, S_0 = \Theta(1)$). The logarithmic factors $\log \frac{4L^2\bar{\beta}_{\tau_\delta}^2}{\lambda\epsilon^2}$, $1 + \log_+ \frac{2C_0}{\Delta_c}$, and $L_\lambda = \log(1 + L^2/\lambda) = \widetilde{O}(1)$. Applying this to Eq. (70) gives

$$\tau_\delta = \widetilde{O}\left( \frac{d\,(\lambda + L^2)}{\rho^\dagger \lambda}\left( \frac{\lambda + L^2}{L^2(\Delta_{\min}^\Pi)^2} + \frac{1}{\Delta_{\mathcal{C}}^2} \right) \right). \tag{70}$$

Equations (66), (68), and (70) state the same bound at decreasing levels of detail. Eq. (68) is fully explicit, with every constant, a closed function of the model parameters and instance geometry. Its two summands are the two phases, namely the deficit-crossing time term, controlled by the global gap $\Delta_{\min}^\Pi$ that floors the

queried leverage until the deficit $2\beta_{t-1}(\delta)\widetilde{M}_{t-1}$ drops below $\Delta_{\mathcal{C}}/2$, and the contraction term, controlled by the boundary gap $\Delta_{\mathcal{C}}$, over which every pairwise variance contracts at the floored rate $\kappa_{\mathrm{cf}}$ until the stopping threshold is met. On the other hand, Eq. (70) retains only the leading order. The dependence is $\widetilde{O}(d)$ rather than in the path count $M$, since the shared parameter propagates each step's information across correlated paths. In addition to that, the additive split $\frac{1}{(\Delta_{\min}^{\Pi})^2} + \frac{1}{\Delta_{\mathcal{C}}^2}$ assigns the deficit-crossing time cost to the global gap and the certification cost to the boundary gap. Furthermore, the factor $1/\rho^{\dagger}$ in both phases quantifies the value of pair–step alignment, with the bound remaining finite as $\rho^{\dagger} \to 0^{+}$.

$\square$

# B    Implementation Details

This appendix collects the implementation details that supplement the empirical study presented in Section 4. Appendix B.1 documents the controlled synthetic protocol used to evaluate GICA on compositional top-$K$ identification. Appendix B.2 describes the details of the end-to-end TTS pipeline used in the math-reasoning benchmarks.

## B.1    Details of the Synthetic Experiments

For each problem scale $M \in \{200, 500, 1000\}$, we generate a random instance of the compositional linear model of Subsection 3.2. Each of the $M$ paths draws its length independently and uniformly from $\{20, \ldots, 80\}$ steps, so the total number of steps grows with $M$ and per-path lengths vary within a fixed range. Step features $x_s \in \mathbb{R}^d$ with $d = 8$ are drawn from an isotropic Gaussian (per-coordinate standard deviation 0.30). We then add a constant offset along the $\theta^{\star}$ direction to each path's steps so that the path utility $\mu(\pi_p) = g(\pi_p)^{\top}\theta^{\star}$ matches a prescribed target, leaving every component orthogonal to $\theta^{\star}$ fully random and untouched. The shared parameter $\theta^{\star}$ is drawn from an isotropic Gaussian and normalized to the unit sphere ($\|\theta^{\star}\|_2 = 1$). The algorithmic norm bound used in $\beta_t(\delta)$ is $S_0 = 2.0$, and the feature-norm budget is $L = 2.5$, which upper-bounds every $\|x_s\|_2$ in the generated instances. Step-level observations add Gaussian noise with $R = 0.1$, consistent with Assumption 3.1.

The design controls the binding boundary gap $\Delta_{\mathcal{C}}$ directly, i.e., after assigning random per-path target utilities, we rigidly shift the top-$K$ block so that the rank-$K$ versus rank-$(K{+}1)$ gap equals a prescribed $\Delta_{\mathcal{C}}$. In contrast, the pair–step correlation constant $\rho^{\dagger}$ of Assumption 3.2 is not set by hand, it is an emergent property of the random geometry, which we measure (statically over the boundary set $\mathcal{C}_K^{\star}$) rather than impose. The sensitivity study in Subsection 4.5 varies $\rho^{\dagger}$ by reseeding the random geometry and reporting the measured value.

GICA and all bandit baselines are run with $\lambda = 1.0$, $\delta = 0.01$, $\epsilon = 0.02$, $R = 0.1$, $S_0 = 2.0$, and $K = 10$, and are capped at 100,000 iterations. None of the runs reaches this cap on the reported instances. All methods share the same $(\lambda, \delta, \epsilon, R, S_0, K)$, so any difference reflects the sampling rule rather than the stopping configuration. Each step query returns a single noisy step-level observation at unit cost for GICA. For the path-arm baselines, querying a path corresponds to evaluating its constituent steps, so its cost equals the path length, matching the per-step verification budget across methods. Results are averaged over 10 independent seeds $\{0, 1, \ldots, 9\}$ per $(M, \text{algorithm})$ configuration, and error bars report one standard deviation.

## B.2    Details of the TTS Pipeline

### B.2.1    Bandit Hyperparameters

All bandit algorithms use a common confidence level $\delta = 0.05$, ridge regularizer $\lambda = 1.0$, tolerance $\epsilon = 0.1$, and shortlist size $K = 5$. The top-$K$ paths returned by each method are aggregated to a final answer by majority vote over a step-level verifier scores. The same decision rule is applied to the Best-of-$M$ upper bound so that the accuracy differences reflect the quality of the returned shortlist rather than the aggregation.

### B.2.2    Step Extraction and Features

Reasoning paths are segmented using a period followed by a newline marker, implemented as `.\n`. This boundary defines the unit at which a verifier query can be issued and yields variable per-path step counts $T_p$ across paths and problems. For each step $s$, we compute $e_s$ as the mean-pooled last-hidden-state embedding from a frozen sentence encoder. The same encoder and feature pipeline are used across all bandit methods to ensure a fair comparison. We represent each reasoning step $s_i$ using a fixed-dimensional feature vector derived from semantic embeddings. Specifically, each step is encoded into an embedding $e_i \in \mathbb{R}^d$ using a pretrained sentence encoder (`all-MiniLM-L6-v2`), with all embeddings $\ell_2$-normalized. For each candidate path $\pi_p$, we compute a centroid embedding $c_{\pi_p} = \frac{1}{T_p} \sum_{s \in \pi_p} e_s$ and similarly define a global centroid $c_{\text{global}} = \frac{1}{|\mathcal{S}|} \sum_{i=1}^{|\mathcal{S}|} e_i$, where $|\mathcal{S}|$ is the number of steps across all paths. Let $e_q$ denote the embedding of the input question. We construct step-level features as: $x_s = \left[ \cos(e_i, e_q), \cos(e_i, c_{\pi_p}), \cos(e_i, c_{\text{global}}), \frac{\text{pos}(i)}{|\pi_p|}, 1, 0 \right]$ $\ell_2$-normalized to $\|x_s\|_2 \leq L = 1$ where $\cos(\cdot, \cdot)$ denotes cosine similarity, and $\text{pos}(i)$ is the index of step $s_i$ within its path. The constant 1 is a bias term, and the final dimension is a placeholder for a dynamically updated boundary feature used by GICA. This design yields low-dimensional, bounded features that capture semantic relevance, path coherence, and structural position, enabling stable and efficient bandit optimization. Embedding quality affects GICA in two ways. First, the weaker embeddings lower the pair–step alignment $\rho^{\dagger}$ (Assumption 3.2), so verifier calls scale as $1/\rho^{\dagger}$ by Theorem 3.8 (Table 6). Second, since Theorem 3.8 concerns surrogate utilities, a poor feature map can reduce the agreement between the certified and exhaustive shortlists before the weights adapt. This is mitigated because every queried step is scored directly by the prefix-conditioned PRM.

### B.2.3 Verification Model

We adopt ThinkPRM-1.5B (Khalifa et al., 2026) and ThinkPRM-7B (Khalifa et al., 2026) as the process-level, reasoning-based verifiers in our main comparisons. Both models are reasoning-based generative PRMs, where they are scaled at test-time by providing more compute, so they can reason step-by-step through a long verification CoT to judge the correctness of each step of the solution (process level). Concretely, when step $s_{p_t, q_t}$ is selected at round $t$, the verifier is prompted with the input question $I_{\text{test}}$, the within path prefix $s_{p_t, 1:q_t - 1}$, and the candidate step $s_{p_t, q_t}$. Thus, the segmentation rule determines the unit of verification but does not truncate the reasoning context supplied to the verifier. The model first generates an internal verification trace and then produces a scalar correctness score, which we treat as the observation $y_t$ associated with that step. The two model sizes (ThinkPRM-1.5B and ThinkPRM-7B) are used to study the robustness of GICA to reasoning-based verifier models of different scales, while keeping the prompt template, segmentation rule, and decoding configuration fixed. The same verifier instance, prompt template, and decoding parameters are used across all bandit methods and across the exhaustive Best-of-$M$ baseline, so that any difference in downstream accuracy or inference runtime reflects the sampling rule rather than the verifier configuration. Each verifier call corresponds to a full verifier forward pass conditioned on the prefix and the candidate step. Its per-call cost is therefore identical across methods, and reductions in the number of verifier calls translate approximately linearly into reductions in inference runtime, as observed in Section 4. Finally, to remain consistent with the linear bandit model of Subsection 3.2, the scalar verifier outputs are treated as conditionally $R$-sub-Gaussian observations on the shared parameter $\theta^{\star}$. This is a tractable light-tailed approximation that we found to be faithful in practice, given the bounded range of ThinkPRM scores.

### B.2.4 More Details of Top-1 Decoding and Best-of-$M$

Top-1 decoding is the baseline that relies on single-shot solution generation using the LLM without any TTS and verification. For each test input $I_{\text{test}}$, the generator produces a single reasoning path through standard sampling at the same temperature used for the TTS configurations, and the final answer is read directly from that path without invoking any verifier. Best-of-$M$ is the exhaustive process-level verification reference point and serves as the upper bound on downstream accuracy attainable by any verifier-based selection rule that operates on the same candidate set $\Pi$. Given the $M = 100$ candidate paths $\Pi = \{\pi_1, \ldots, \pi_M\}$ sampled by the generator, the PRM is queried on every step of every candidate path, yielding a scalar score for each step $s_{p,q} \in \mathcal{S}$. Each path $\pi_p$ is then scored by aggregating its step-level verifier outputs, with the final solution path being chosen via majority vote over aggregated step-level scores. For a fair comparison across all bandit methods, the final answer is produced by the same majority-vote rule applied to aggregated step-level scores from top-$K$ shortlists of solution paths. Consequently, any accuracy gap between the bandit approaches is

attributable to the quality of the returned shortlist rather than to differences in how the shortlist is collapsed into a single answer. Because Best-of-$M$ issues the maximum possible number of verifier calls permitted under the chosen (generator, verifier) pair, it also provides the reference against which the inference runtime and verifier call savings of GICA are reported in Section 4.

## C   Extended Ablation Studies and Simulation Results

This appendix complements Section 4 so that on the real data benchmarks, we ablate the scale of the reasoning-based verifier, replacing ThinkPRM-1.5B with ThinkPRM-7B, to test whether the behavior of Section 4 is intrinsic to the sampling rule or an artifact of a specific verifier. We report only results not already in the main text.

### C.1   Ablation Studies on Real-World Datasets: Robustness to Verifier Scale

We repeat the full TTS pipeline with the larger ThinkPRM-7B verifier, holding the generators (DeepSeekMath-RL-7B, InternLM2-Math-Plus-7B), $M = 100$, $K = 5$, the step-segmentation rule, the feature encoder, and all bandit hyperparameters fixed at the values of Appendix B.2, so any change reflects verifier scale alone. Downstream accuracy is in Table 5 and per-query inference runtime and verifier calls in Figure 5.

Table 5: Exact match across datasets with ThinkPRM-7B as verifier. Second-highest scores are underlined.

| Method | Deepseek-MATH-RL-7B | | | InternLM2-MATH-PLUS-7B | | |
|---|---|---|---|---|---|---|
| | MATH-500 | MathOdyssey | AIME | MATH-500 | MathOdyssey | AIME |
| **Verification** | | | | | | |
| Top-1 decoding | 41.00 | 17.48 | 6.50 | 15.20 | 6.43 | 3.50 |
| Best-of-M (Exhaustive) | 54.50 | 31.87 | 11.75 | 54.76 | 22.87 | 10.25 |
| **Bandit Approaches** | | | | | | |
| CASE | 49.20 | 23.60 | 8.30 | 49.03 | 17.59 | 8.45 |
| m-LinGapE | 48.18 | 28.17 | 6.36 | 48.32 | 19.92 | 5.97 |
| LinGIFA | 48.70 | 27.99 | 7.91 | 48.79 | 20.25 | 8.85 |
| **Our Approach** | | | | | | |
| **GICA** | 51.00 | 28.97 | 7.81 | 49.16 | 20.31 | 7.75 |

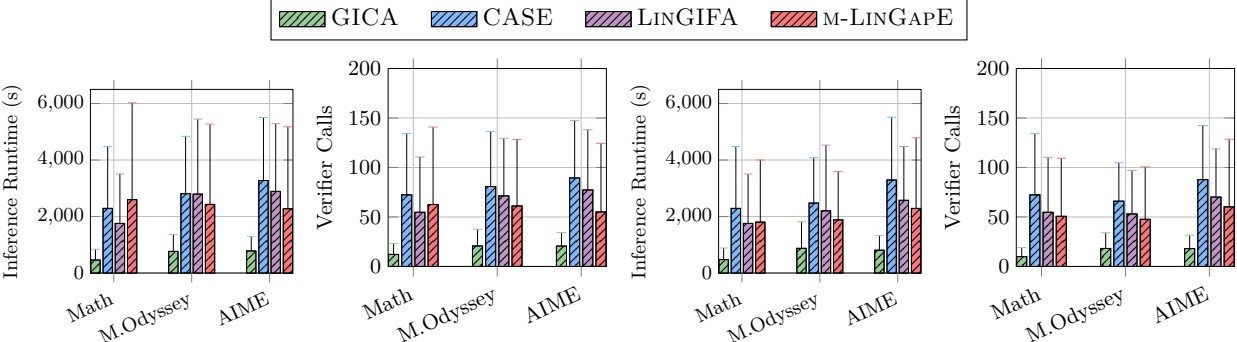

(a) Runtime (DeepSeek).   (b) Verifier calls (DeepSeek).   (c) Runtime (InternLM2).   (d) Verifier calls(InternLM2).

Figure 5: Sample efficiency of verification using GICA compared to other state-of-the-art linear-stochastic bandit algorithms across DeepSeek-7B and InternLM2-7B as generators, with ThinkPRM-7B as verifier.

Figure 5 shows that GICA is the most sample efficient compared to other approaches, i.e., GICA issues the fewest number of verifier calls among the adaptive methods across different generative models. Since each call has identical per-call cost (Appendix B), this maps approximately linearly onto reduced per-query inference runtime. We observe from Figure 5a that GICA offers up to **5.7** $\times$, **4.6**$\times$ speedup in terms of inference

runtime over CASE and LINGIFA respectively on MATH-500. We observe similar or higher speedups on MathOdyssey and AIME. Similarly, Figure 5c also reveals a similar pattern with InternLM2 LLM as generator, where GICA offers up to **4.8**× and **3.9**× speedups over CASE and LINGIFA respectively on MATH-500. Absolute inference runtimes are larger with the heavier verifier, but the relative ordering is unchanged, indicating the gain is intrinsic to the compositional sampling rule.

With regards to final task performance, GICA is closer to the exhaustive Best-of-$M$ upper bound on most benchmarks and also significantly outperforms Top-1 decoding on every benchmark. Among the bandit baselines it attains the highest accuracy on most of the benchmarks (MATH-500 and MathOdyssey) for both generators, on DeepSeekMath-RL-7B (MATH-500 51.00% vs. 49.20%; MathOdyssey 28.97% vs. 28.17%) and marginally on InternLM2-Math-Plus-7B (MATH-500 49.16% vs. 48.79%; MathOdyssey 20.31% vs. 20.25%). Two limitations are worth stating. First, on AIME GICA does not lead the baselines under either generator, trailing CASE and LINGIFA (7.81% vs. 8.30%/7.91% on DeepSeekMath-RL-7B; 7.75% vs. 8.45%/8.85% on InternLM2-Math-Plus-7B). AIME has fewer correct paths compared to other benchmarks due to the difficulty of the task. Hence, the discriminating signal concentrates in a few steps rather than a shared structure, weakening the benefit of compositional step sharing. Second, the gap to Best-of-$M$ widens under the ThinkPRM-7B verifier (e.g., InternLM2-Math-Plus-7B on MATH-500, 49.16% vs. 54.76%), reflecting a cost, accuracy tradeoff common to all adaptive methods, i.e., a more discriminative verifier makes exhaustive scoring more valuable, so pruning the space of reasoning paths to be verified forgoes some accuracy.

## C.2 Generation vs Verification Cost

To motivate the need for sample-efficient verification, we compare the verification and generation runtime costs across the datasets with DeepSeekMath-RL-7B as generator and ThinkPRM-7B as verifier for 100 reasoning paths per query in exhaustive setting. The outcome is as shown in Figure 6. We observe that verification runtime occupies quite a significant portion of total end-to-end latency, almost equal to or sometimes more than generation. For instance, in AIME, verification contributes to **50.4%** of end-to-end inference runtime/latency whereas generation contributes to **49.6%**. Similarly, on MATH-500, verification contributes to **53.8%** and generation contributes to **46.2%** of runtime. For MathOdyssey, verification contributes to **52.9%** of end-to-end runtime and generation contributes to **47.1%**. This demonstrates that verification latency is non-trivial, and careful exploration of the search space would make the runtime more tractable. We would like to highlight that the high cost (in terms of runtime) in verification is due to the use of reasoning-based verification models. Our setup is inspired by recent works which demonstrate that employing smaller generative LLMs with strong verification scaled at test time helps improve performance across a wide range of tasks (Khalifa et al., 2026).

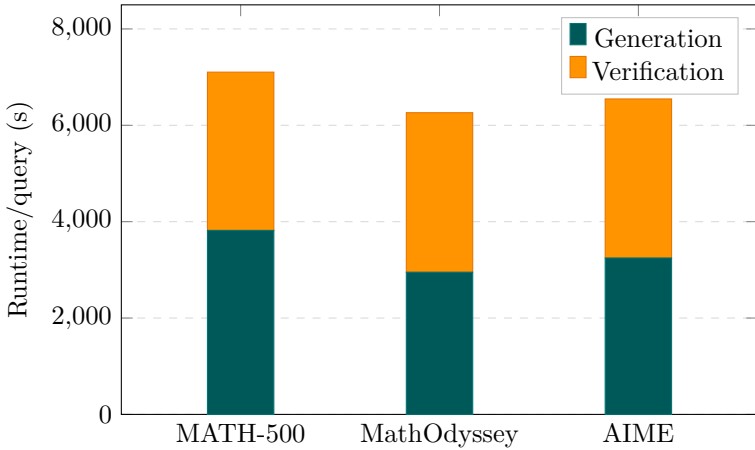

Figure 6: Generation vs verification runtime (average time/query) comparison with DeepseekMath-RL-7B as generator and ThinkPRM-7B as verifier. The percentage indicates what proportion of total runtime is the independent aspects (verification & generation).

## C.3 Shortlist Recovery in the Synthetic Setting

To isolate the surrogate from verifier noise, we study GICA in the synthetic setting of Appendix B.1, where the verifier is an oracle and the target is the top-$K$ set by average step score. Figure 7 reports the per-round top-$K$ identification error, the percentage of certified paths outside the true top-$K$ set, over ten seeds with $M = 1000$ and $K \in \{5, 10, 20\}$. For all three values of $K$ the error decreases to zero within a few tens of rounds and GICA recovers the true top-$K$ set exactly in every seed, confirming convergence of the surrogate to oracle layer under this hierarchical step to path setting.

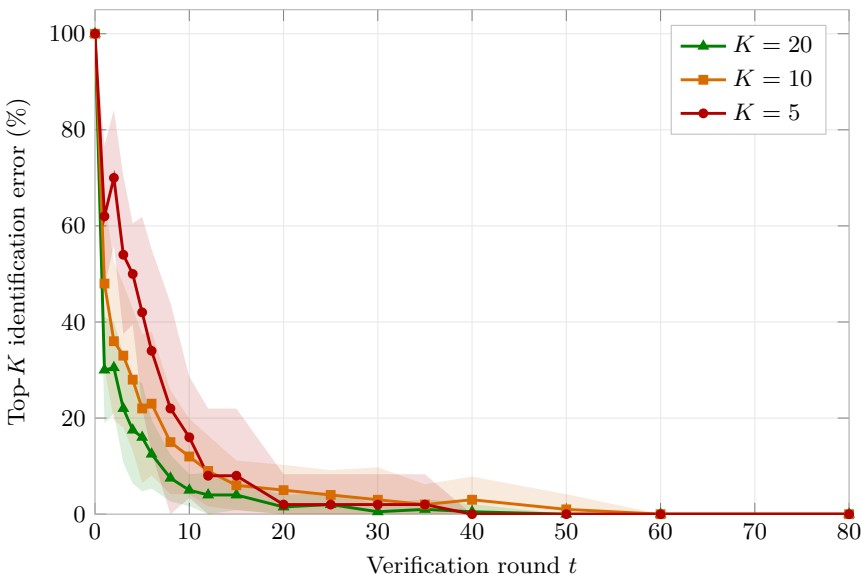

Figure 7: Per-round top-$K$ identification error of GICA in the synthetic setting ($M = 1000$, mean and one standard deviation over ten seeds).

## C.4 Sensitivity to K

We also vary the K = (1,3,5,10) in top-K paths with DeepseekMath-RL-7B as the generator and to be identified and observe the task performance. We observe that performance steadily increases and the gain becomes negligible after $K = 5$. It is also notable that at $K = 10$, the runtime of GICA bandit loop takes longer to converge due to need to certify more arms. This is also evident from synthetic setup in Figure 3c where runtime to certify $K = 10$ arms is longer than $K = 5$.

| Dataset | K=1 | K=3 | K=5 | K=10 |
|---|---|---|---|---|
| MATH-500 | 47.70 | 49.11 | 51.00 | 51.91 |
| MathOdyssey | 24.96 | 26.48 | 28.97 | 29.01 |
| AIME | 6.25 | 7.50 | 7.81 | 7.90 |

Table 6: Performance (measured by exact match) across three benchmarks with varying K for top-K identification.

## C.5 Accuracy–Verification-Cost Pareto Analysis

Figure 8 combines exact-match accuracy and average step-level verifier calls across all twelve evaluated dataset–generator–verifier configurations. Relative to exhaustive Best-of-$M$, GICA uses 33.3×–105.2× fewer verifier calls. In three ThinkPRM-1.5B configurations, the corresponding accuracy difference is at most 0.40 percentage points. The larger differences in some ThinkPRM-7B configurations identify regimes with

a stronger accuracy–cost trade-off. Solid curves connect non-dominated methods, whereas faded markers denote dominated points.

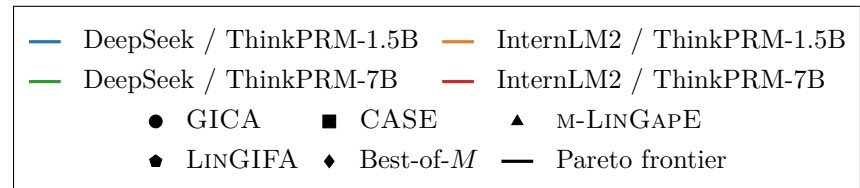

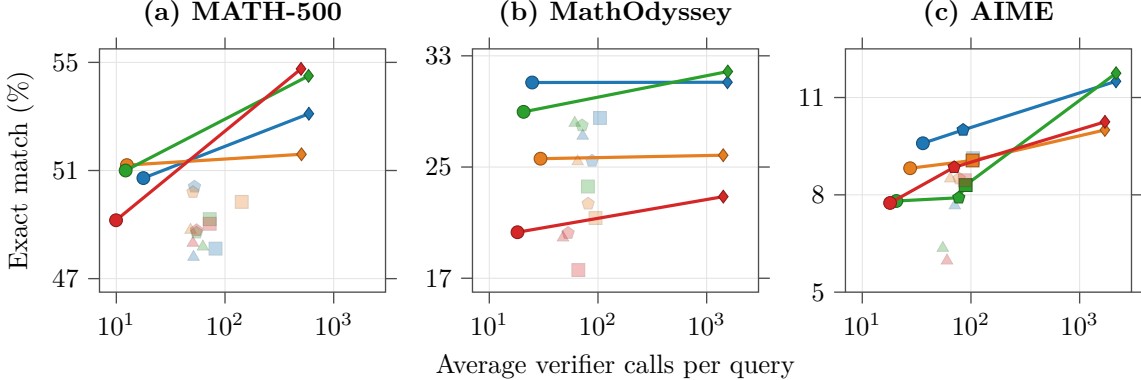

Figure 8: **Accuracy–verification-cost Pareto comparison.** Panels denote benchmarks, colors denote generator–verifier configurations, and marker shapes denote methods. The horizontal axis reports average verifier calls per query (log scale), and the vertical axis reports exact-match accuracy. Solid curves connect non-dominated points; faded markers denote dominated methods.

