# OpenReview forum: "GICA: The Gap-Index Compositional Arm Framework for Sample-Efficient Test-Time Scaling"
_TMLR — Under review for TMLR_

### Review · Reviewer_MGxB · 2026-07-04

**Summary Of Contributions:**

### Summary
- The paper tackles the growing cost of exhaustive process-level verification in test-time scaling with reasoning-based PRMs. It reframes verification as fixed-confidence top-K identification over compositional arms, where each reasoning path is modeled as an aggregate of its step features, with shared parameters letting a single step observation inform many paths at once. The authors propose GICA, a gap-index linear bandit algorithm, and provide correctness guarantees for the top-K shortlist, a sample-complexity bound independent of the number of candidate paths, and experiments on synthetic and math-reasoning tasks.

### Strength
- Modeling inter-path correlation through compositional arms, so step-level observations propagate across paths, is a clean departure from prior top-K bandit work.
- The candidate-independent sample-complexity bound is a theoretically meaningful result.
- Theory plus controlled synthetic experiments plus an end-to-end pipeline, with released code, supports reproducibility.

### Weakness
- The core motivation (why reduce verifier calls) and the framing choice (why fixed-confidence top-K) are under-justified. (See Requested changes section)
- What the algorithm certifies is two approximations removed from what we actually care about, accuracy, and that gap is never examined empirically.
- Accuracy does not exceed the Best-of-M ceiling and falls below baselines in places, so the whole value proposition rests on cost savings being significant.

**Audience:**

Yes

**Audience Explanation:**

Both TTS/process-verification practitioners and pure-exploration bandit researchers would find the reframing and the candidate-independent bound of interest.

**Broader Impact Concerns:**

Since skipping verification could let error paths through in safety-critical reasoning, a sentence noting this limitation for such settings would be worthwhile.

**Claims And Evidence:**

No

**Claims Explanation:**

Partially supported. The theory is clear and well-grounded, but two load-bearing premises are not.

- GICA generates all candidate paths and only trims verification, so generation cost is unchanged. The headline runtime speedup is measured on verification passes alone, with no evidence that verification dominates end-to-end cost. If generation is a large share of the total, the reported speedup would shrink in practice. A generation-vs-verification cost breakdown is missing.
- The algorithm certifies top-K under a linear surrogate, which is separated from downstream accuracy by two layers of approximation (verifier score \sim correctness, surrogate \sim verifier score). The rigor of the guarantee therefore does not carry over to accuracy, and the cases where GICA trails baselines look symptomatic of this. Nothing quantifies how well the certified shortlist matches the truly correct paths. Deployment also looks closer to a fixed-budget setting, yet the fixed-confidence choice goes unjustified.

The realism of the correlation and separability assumptions, especially how strong the correlation actually is, is not established in the main text.

**Requested Changes:**

- C1. Show that verification dominates total cost: report a generation-vs-verification breakdown and give end-to-end speedups, stating where the gains are limited.
- C2. Quantify the gap between what is certified and accuracy, for example agreement between the certified shortlist and the actually-correct paths.
- C3. Justify the framing in the main text: why fixed-confidence over fixed-budget, and the rationale for top-K and the value of K, with a sensitivity analysis over K.
- C4. Bring empirical validation of the correlation and separability assumptions into the main text.

Minor things
- Clarify the advantage over Best-of-M with an accuracy-vs-cost Pareto curve, supporting "same accuracy at much lower cost."
- Fix the inconsistent y-axis scales across subplots that obscure method comparison.
- Improve figure legibility, since labels and arrows overlap.
- "Sample-efficient" reads as generation savings; make explicit that savings are confined to verification (e.g., verification-efficient).
- Add a brief comparison against alternatives (smaller verifier, fewer paths, outcome-model fallback) to motivate why verifier-call reduction is the priority.

---

### Review · Reviewer_x3kr · 2026-07-04

**Summary Of Contributions:**

Summary:
The paper studies the high computational cost of process-level verification in test-time scaling (TTS) for large language models. Instead of exhaustively verifying every reasoning step of every candidate chain-of-thought, the authors formulate process-level verification as a fixed-confidence top-K identification problem over compositional arms. Building on this formulation, they propose GICA, a gap-index based bandit framework that exploits the compositional structure of reasoning paths to share information across intermediate reasoning steps and adaptively select the most informative verification queries. The paper provides a theoretical analysis with fixed-confidence sample complexity guarantees and evaluates the method on synthetic experiments and three mathematical reasoning benchmarks using multiple generator and verifier models. The results demonstrate that GICA achieves downstream accuracy close to exhaustive process-level verification while substantially reducing verifier calls and inference runtime.

Strengths:
1. The problem formulation is novel and well motivated, casting process-level verification as a compositional top-K identification problem rather than directly applying existing bandit methods.
2. The paper provides a solid theoretical treatment, including formal assumptions, algorithmic derivation, and sample-complexity analysis.
3. The paper provides comprehensive empirical evaluation, including comparisons against bandit baselines, runtime analysis, and experiments across multiple benchmarks and models.

Weaknesses:
1. The theoretical analysis relies on a shared linear utility model over step embeddings, which is supported by classical linear stochastic bandit literature. However, most of the cited supporting work dates back to early linear bandit formulations (for example, 2011-2017), whereas modern process reward models are themselves large nonlinear neural networks whose scoring behavior can be considerably more complex. While the linear surrogate is mathematically convenient and enables the theoretical analysis, it would strengthen the paper to discuss why this approximation remains appropriate for contemporary reasoning-based PRMs such as ThinkPRM, and under what conditions the assumption may become inaccurate.
2. The empirical evaluation is limited to mathematical reasoning benchmarks. Although this is a natural application for process reward models, recent test-time scaling literature (such as [1]) has increasingly evaluated methods across a broader collection of reasoning tasks, including planning, instruction following, code generation, and agentic reasoning. Since GICA is presented as a general framework for efficient process-level verification rather than a math-specific algorithm, a broader discussion or future evaluation in these settings would better support the approach's claimed generality.
[1] Lin, J., Zeng, X., Zhu, J., Wang, S., Shun, J., Wu, J., & Zhou, D. (2025). Plan and budget: Effective and efficient test-time scaling on reasoning large language models. In The Fourteenth International Conference on Learning Representations.

3. The proposed framework derives its advantage from sharing information across reasoning paths through semantically related intermediate steps. While this motivation is convincing theoretically, the paper provides limited empirical analysis demonstrating when or why this information sharing is most beneficial in practice. Additional analysis of how reasoning-path diversity or shared-step structure affects GICA's performance would strengthen the paper and provide better intuition for its practical behavior.

**Audience:**

Yes

**Audience Explanation:**

The paper addresses an active research direction at the intersection of test-time scaling, reasoning with large language models, and sequential decision making. Researchers working on efficient inference, process reward models, adaptive computation, bandit algorithms, and reasoning systems are likely to find the proposed formulation and algorithm of interest. Beyond the specific application to mathematical reasoning, the compositional bandit perspective may also motivate future work on efficient verification and adaptive inference in other reasoning domains.

**Broader Impact Concerns:**

I do not have significant broader impact concerns beyond those typically associated with improving the efficiency of large language model inference. The proposed method primarily reduces the computational cost of process-level verification and does not introduce new application domains or capabilities beyond existing test-time scaling frameworks. Therefore, I do not believe additional discussion of broader societal impacts is necessary beyond any standard limitations already included in the manuscript.

**Claims And Evidence:**

Yes

**Claims Explanation:**

The main claims are supported by both theoretical analysis and empirical evaluation. The paper clearly defines the proposed compositional bandit formulation, provides theoretical guarantees under explicitly stated assumptions, and validates the approach experimentally using synthetic studies and three mathematical reasoning benchmarks. The experiments compare against relevant bandit-based baselines and evaluate both downstream task accuracy and computational efficiency, showing consistent reductions in verifier calls and inference runtime while maintaining performance close to exhaustive process-level verification.

The theoretical assumptions, particularly the shared linear utility model over step embeddings, are idealized and may not fully capture modern LLM verifier behavior. However, these assumptions are clearly stated and are standard within the linear stochastic bandit literature. Therefore, they do not undermine the correctness of the theoretical claims made in the paper.

**Requested Changes:**

Critical: None. I did not identify any issues that would prevent publication assuming the technical results are correct.

Suggestions that would strengthen the paper:
1. Expand the discussion of the assumptions underlying the theoretical analysis. In particular, the proposed framework relies on a shared linear surrogate utility model over step embeddings, which is motivated by classical linear stochastic bandit literature. Since modern process reward models are themselves highly nonlinear language models, it would strengthen the paper to discuss why this approximation remains appropriate in contemporary PRM settings and under what conditions it may become less accurate.
2. Discuss the applicability of the framework beyond mathematical reasoning. The current evaluation focuses exclusively on mathematical reasoning benchmarks, whereas recent test-time scaling work has increasingly considered a broader range of reasoning tasks, such as code generation, planning, instruction following, and agentic reasoning. These tasks may generate reasoning traces with less information overlap, potentially violating the motivation and assumption discussed in the paper. While additional experiments are not necessary for acceptance, discussing the expected applicability and potential limitations of GICA in these settings would better support the paper's broader claims.
3. Provide additional empirical insight into the proposed compositional information-sharing mechanism. The primary novelty of GICA is its ability to propagate verifier feedback across reasoning paths through shared compositional structure. It would strengthen the paper to include additional discussion or analysis illustrating when this mechanism is most beneficial in practice, for example as a function of reasoning-path diversity or the amount of shared intermediate reasoning among candidate paths.
4. Discuss the sensitivity of the framework to the quality of the underlying step representations. Since information sharing depends on meaningful step embeddings, it would be helpful to discuss how embedding quality may influence GICA's effectiveness and in what situations the proposed approach is expected to be more or less beneficial.

---

### Review · Reviewer_fLoa · 2026-07-07

**Summary Of Contributions:**

This paper considers a post-hoc process-verification setting in which multiple complete reasoning paths are first generated, each response is segmented into reasoning steps, and process-level verifier scores are used to select a top-$K$ subset before majority voting. The paper proposes GICA, a compositional bandit framework that adaptively selects a subset of steps for verification rather than exhaustively scoring every step. The method reduces verifier calls and inference runtime compared with exhaustive process verification and the adapted linear-bandit baselines, at the cost of some downstream accuracy in several settings.

The paper addresses the relevant problem of reducing verifier cost. However, I have two major concerns.

1. **The practical relevance of the underlying pipeline and the baseline selection is insufficiently established.** The paper mainly compares GICA with classical top-$K$ linear-bandit methods adapted to the authors' formulation. These comparisons demonstrate that GICA is more efficient within the proposed bandit setting, but do not establish that the proposed pipeline is competitive with modern LLM test-time scaling strategies. Representative matched-compute comparisons with mainstream TTS approaches, such as large-sample majority voting, ORM-based reranking, an ORM-to-PRM cascaded pipeline, beam/tree search, and representative MCTS-style methods, are missing.

2. **The core cross-step information-sharing mechanism appears overly coarse for modern reasoning models.** GICA relies on low-dimensional semantic and structural features together with a shared linear surrogate to propagate verifier observations across reasoning steps. However, correctness in complex reasoning is highly dependent on fine-grained logical context, and semantically similar steps may have very different correctness. The paper does not provide sufficiently convincing evidence that this coarse surrogate reliably captures the relationships required for such information sharing, particularly for long reasoning traces.

Overall, while GICA shows clear efficiency gains over the selected bandit baselines, the current evaluation does not convincingly demonstrate its advantage as a modern test-time scaling method.

**Audience:**

Yes

**Audience Explanation:**

The proposed formulation may be of interest to a limited subset of TMLR readers, particularly those working at the intersection of adaptive sampling, bandit methods, and LLM verification. The reported efficiency results within the formulated setting may also be useful for this audience.

However, I am not convinced that the paper establishes that its post-hoc step-level verification setting is representative of modern test-time scaling practice, or that the proposed cross-step information-sharing mechanism is applicable to substantially more complex long-form reasoning. Therefore, my "Yes" mainly reflects potential interest in the specific formulation and empirical observations, rather than strong confidence in the broader relevance of the findings to current LLM test-time scaling.

**Broader Impact Concerns:**

N/A.

**Claims And Evidence:**

No

**Claims Explanation:**

The paper provides evidence that GICA reduces verifier calls and inference runtime compared with the selected bandit baselines. However, I do not think the current evidence is sufficient to support two central claims of the paper.

1. **The practical motivation of the proposed TTS setting is not convincingly established.** The paper motivates GICA by arguing that current test-time scaling with process-level verification is prohibitively expensive. However, the evaluated pipeline first generates all complete reasoning responses and then performs step-level verification over the generated responses. It is unclear whether this setting is representative of modern LLM test-time scaling practice. The experiments mainly compare GICA with classical bandit methods adapted to this specific formulation, rather than with representative TTS strategies actually used for LLM reasoning. Therefore, the current results show that GICA is more efficient within the authors' constructed bandit setting, but do not convincingly support the broader claim that it provides a practical solution to the cost of modern test-time scaling.

2. **The claim that verifier information can be effectively shared across reasoning steps is insufficiently supported.** The proposed method represents each step using a six-dimensional feature vector based mainly on semantic similarities and positional information, and uses this representation to propagate verifier feedback across steps. I find this representation overly coarse for complex reasoning. In modern long-thinking models, a response may contain a very long and fine-grained reasoning trace, while the correctness of an individual step can depend on subtle logical details rather than semantic similarity. This concern is further amplified by the use of newline delimiters to define reasoning steps. The current experiments do not convincingly demonstrate that such a low-dimensional representation can reliably capture meaningful relationships among steps in substantially longer and more complex reasoning traces.

Therefore, while the efficiency improvement over the selected bandit baselines is clear, I do not believe the current evidence sufficiently supports the paper's broader claims regarding practical test-time scaling and effective cross-step information sharing.

**Requested Changes:**

1. [Critical] Compare against representative modern TTS strategies under matched inference compute. The current experimental comparison is dominated by classical top-$K$ bandit methods adapted to the proposed formulation. This establishes the advantage of GICA within the authors' bandit setting, but does not establish that the proposed pipeline is competitive with modern LLM test-time scaling strategies. The paper should include representative LLM-side baselines under comparable total inference compute. At minimum, I would expect comparisons with larger-sample majority voting and ORM-based reranking, which avoid expensive step-level verification and can therefore evaluate substantially more complete responses under the same inference budget; an ORM-to-PRM cascaded pipeline, for example by first applying an ORM to all complete responses and then performing process-level verification only on the top $20\%$ candidates; and representative search-based methods such as beam/tree search and MCTS-style approaches respectively. This last comparison is particularly important because GICA only performs post-hoc selection over an already generated candidate set and therefore cannot improve the pass@N of that candidate set, whereas process-guided search can alter the generation trajectory and potentially improve the quality and coverage of the candidate solutions. The comparison should focus on final answer accuracy under a matched compute or runtime budget, rather than only the number of PRM calls.

2. [Critical] Provide convincing evidence that cross-step information sharing is reliable for complex reasoning. The main efficiency gain of GICA relies on propagating verifier information across reasoning steps using a low-dimensional semantic and structural representation. However, correctness in complex reasoning is often determined by fine-grained logical details, and semantically similar steps may have substantially different correctness. The paper should directly demonstrate that the proposed cross-step information-sharing mechanism is reliable on realistic long-form reasoning traces. In particular, the current evaluation is based on relatively short reasoning responses segmented by newline delimiters, while modern long-thinking models can generate substantially longer and more complex reasoning trajectories. Evaluation on stronger long-thinking generators, such as Qwen3 or DeepSeek-R1-style models, would be important to demonstrate that the proposed low-dimensional representation and information-sharing mechanism remain effective in this regime. Without such evidence, the central mechanism behind the reported efficiency gain remains insufficiently validated.

3. Evaluate beyond mathematical reasoning or narrow the scope of the claims. The paper is framed broadly as a test-time scaling method, but all real-task experiments are conducted on mathematical reasoning benchmarks. Since the proposed information-sharing mechanism is claimed to exploit general compositional structure across reasoning steps, evaluation in additional domains, such as code reasoning or agentic tasks, would substantially strengthen the claimed generality of the method. Alternatively, the paper should narrow its claims and framing to mathematical reasoning.